# Angle between DNA linker and nucleosome core particle regulates array compaction revealed by individual-particle cryo-electron tomography

Meng Zhang [1,2,3], César Díaz-Celis[3,4], Jianfang Liu [1], Jinhui Tao [5], Paul D. Ashby [1], Carlos Bustamante [2,3,4,6,7,8,9,10] ✉ & Gang Ren [1] ✉

The conformational dynamics of nucleosome arrays generate a diverse spectrum of microscopic states, posing challenges to their structural determination. Leveraging cryogenic electron tomography (cryo-ET), we determine the three-dimensional (3D) structures of individual mononucleosomes and arrays comprising di-, tri-, and tetranucleosomes. By slowing the rate of condensation through a reduction in ionic strength, we probe the intra-array structural transitions that precede inter-array interactions and liquid droplet formation. Under these conditions, the arrays exhibit irregular zig-zag conformations with loose packing. Increasing the ionic strength promoted intra-array compaction, yet we do not observe the previously reported regular 30-nanometer fibers. Interestingly, the presence of H1 do not induce array compaction; instead, one-third of the arrays display nucleosomes invaded by foreign DNA, suggesting an alternative role for H1 in chromatin network construction. We also find that the crucial parameter determining the structure adopted by chromatin arrays is the angle between the entry and exit of the DNA and the corresponding tangents to the nucleosomal disc. Our results provide insights into the initial stages of intra-array compaction, a critical precursor to condensation in the regulation of chromatin organization.

Throughout the cell cycle, chromatin alternates between a loosely packed state that favors gene accessibility and transcription and a tightly packed state that represses them[1]. This change is reflected in the reorganization observed in nucleosome arrays, transitioning from their open form as "10-nm" chromatin fibers, or "beads-on-a-string" to more compact structures. The "10-nm" fiber consists of tandem repeats[2] of nucleosome core particles (NCPs)[3], each comprising 147 bp of dsDNA wrapped around a histone octamer and separated by DNA linkers 10–60 bp in length. Several models have been proposed to explain how the "10-nm" fiber[4] condenses into various forms of "ordered 30-nm" structures, including the one-start solenoid[5], two-start helix[6,7], untwisted two-start helix[8], and crossed-linker[9]. Further folding and intertwining of these 30-nm fibers are considered key intermediates in the formation of higher-order chromatin

[1]The Molecular Foundry, Lawrence Berkeley National Laboratory, Berkeley, CA, USA. [2]Applied Science and Technology Graduate Group, University of California, Berkeley, CA, USA. [3]California Institute for Quantitative Biosciences, University of California, Berkeley, CA, USA. [4]Howard Hughes Medical Institute, University of California, Berkeley, CA, USA. [5]Physical Sciences Division, Pacific Northwest National Laboratory, Richland, WA, USA. [6]Department of Chemistry, University of California, Berkeley, CA, USA. [7]Department of Physics, University of California, Berkeley, CA, USA. [8]Department of Molecular and Cell Biology, University of California, Berkeley, CA, USA. [9]Molecular Biophysics and Integrative Bioimaging Division, Lawrence Berkeley National Laboratory, Berkeley, CA, USA. [10]Kavli Energy Nanoscience Institute, University of California, Berkeley, CA, USA. ✉e-mail: carlosb@berkeley.edu; gren@lbl.gov

structures[10,11]. However, the lack of consistent detection of these structures in vivo[12–16] has challenged this hierarchical model. Instead, a reversible thermodynamic process of liquid-liquid phase separation (LLPS) has been proposed as an alternative to explain chromatin compartmentalization and condensation behavior, supported by evidence from both in vitro and in vivo experiments[17,18].

Nucleosome array condensation have been shown to involve a multi-stage progression from dispersed, evenly distributed arrays to the development of spinodal condensates, appearance of small nuclei, which grow eventually into large spherical condensates[19]. Previous research indicates that prior to the formation of condensates, nucleosome arrays experience self-compaction[20,21] (Fig. 1a, and Supplementary Fig. 1a, b), implying that asynchronous intra-array and inter-array interactions govern distinct levels of nucleosomal condensation. While significant research efforts have been made to elucidate the initial stages of intra-array compaction preceding the attainment of inter-array interactions[21–23] (Fig. 1a), incomplete 3D structural characterization persists. The key barriers include: (i) the fast process of the intra-array compaction under physiological salt condition (Supplementary Fig. 1a, b), providing insufficient time to prepare the cryo-electron tomography (cryo-ET) grid before inter-array interaction takes over, and (ii) the intrinsic flexibility of arrays that prevents the use of averaging approaches by conventional cryo-EM single-particle analysis (SPA) and cryo-ET sub-tomogram analysis. Investigating the nonequilibrium intra-array compaction process is therefore necessary to understand the molecular processes that could

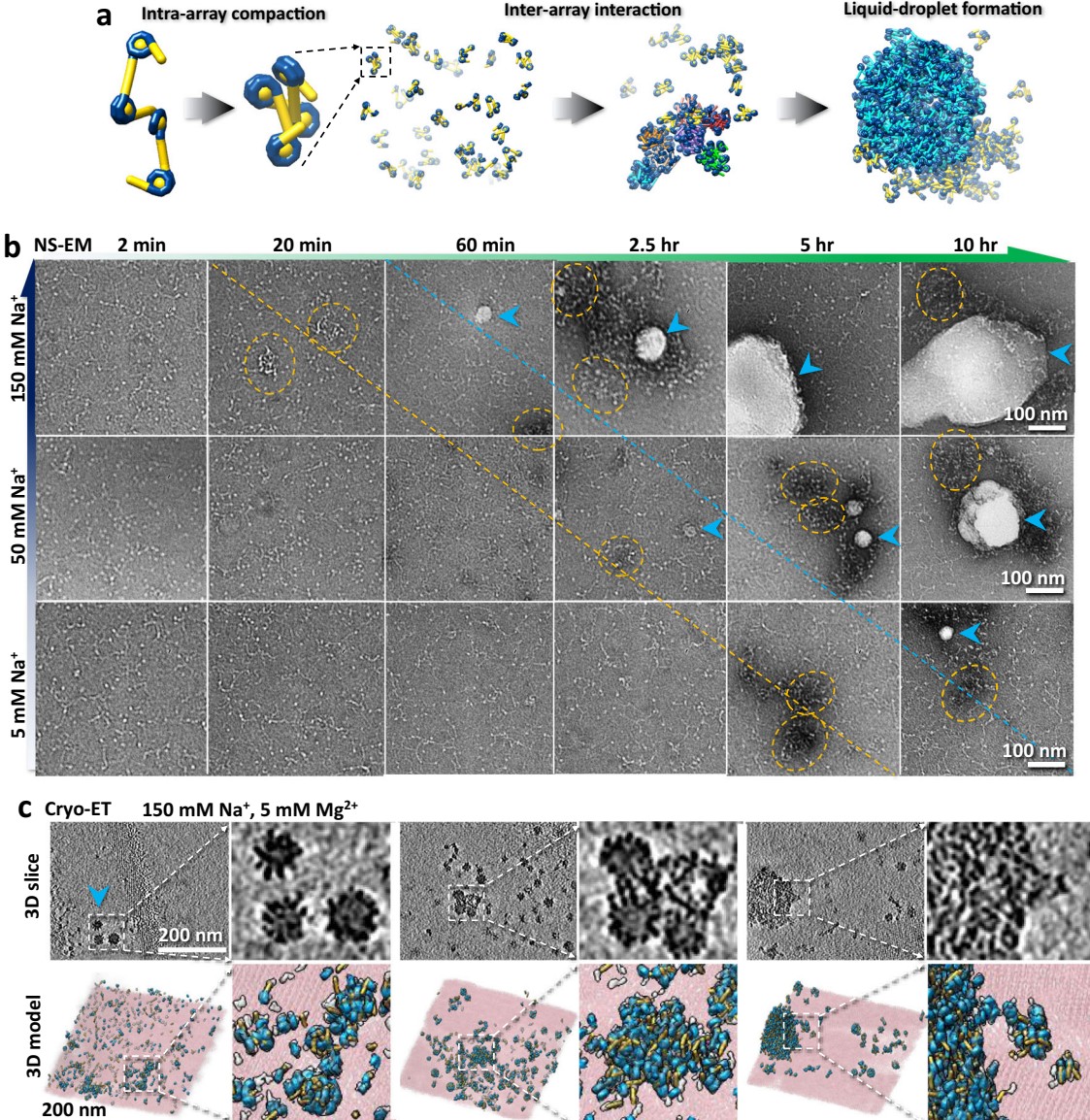

**Fig. 1 | Structure and morphology of phase separation. a**, A schematic model depicting the structural transition of nucleosomes, from a 10-nm "beads-on-a-chain" structure or loosely-packed polymer state to a fiber-like structure, spinodal condensate, and finally to a spherical condensate through the process of intra-array compaction, inter-array interaction and liquid-droplet formation. **b**, NS-EM images of the morphology of tetranucleosome incubated with 150 mM, 50 mM and 5 mM Na⁺, respectively, under room temperature for up to 10 h. The spinodal condensate particles are marked by orange circles, while the spherical condensates are indicated by cyan arrows. The boundary for generating spinodal condensate is depicted by orange dash line, while the boundary to generating the spherical condensate is described by cyan dash line. In this experiment, one grid was prepared for each condition, with ~5-10 grid squares examined and imaged, all showing a similar distribution of particles. **c**, Three representative area of cryo-ET 3D reconstruction of spherical condensate generated from tetranucleosome in physiological salt concentration as described[19]. Each 3D map showed by its central slice (top left) and zoomed-in portion of the small spherical condensates (top right), which is compared to the 3D density map superimposed with models (the NCP portions are colored by cyan, and DNA linker portions are colored by yellow).

transition from the 10-nm fiber structure to higher-order fiber structures and/or to LLPS droplets.

Here we use cryo-ET to determine the non-averaged 3D structure of individual nucleosomal array particles after 20 min incubation under low-salt condition (5 mM Na$^+$) and room temperature to capture the initial-state structure of tetranucleosome arrays toward to the emergence of inter-nucleosomal interactions and LLPS condensates. By characterizing these array structures and comparing them to that obtained under higher-salt condition (50 mM Na$^+$) with same temperature and duration, we uncovered the intra-array conformational changes prior to the emergence of inter-array interactions and phase separation. To determine how the number of nucleosomes in the array affects the distribution of the linker DNA orientation and length neighboring each NCP, we examined samples of mono-, di-, tri- and tetranucleosome arrays. Our approach enables us to visualize the spatial organization of the linker DNA flanking each NCP which, we find, a key factor that regulates chromatin conformation. We also investigate the effect on intra-array compaction of histone H1, known to be a catalyst in the process of phase separation[19]. Accordingly, we examined tetranucleosomes in the presence of histone H1 under low-salt condition (5 mM Na$^+$) and room temperature for 20 min. We found that H1 induces the DNA linkers to adopt a "closed" form and that H1 promotes DNA invasion into neighboring NCPs, a potential mechanism to increase chromatin networking and possible compaction. These observations provide insight into intra-array interactions associated with the preliminary stages before array condensation ensues.

## Results

### Slowing down the kinetics of phase separation by reducing Na$^+$ concentration

Prior investigations utilizing fluorescence microscopy and negative stain electron microscopy (NS-EM) have revealed that spherical condensates of 30 nM tetranucleosome arrays can form within 10 min of incubation under physiological salt concentrations (150 mM Na$^+$ and 5 mM Mg$^{2+}$) and room temperature (Supplementary Fig. 1a-b). This process occurs following the almost immediate formation of spinodal condensates. In the presence of H1, the spherical condensates are observed within 2 min of incubation[19]. This rapid reaction poses a challenge to characterize the initial intra-array conformational changes that take place before inter-array interactions commence.

After 20–60 min of incubation at room temperature under a 150 mM Na$^+$ concentration, NS-EM imaging revealed the presence of irregular aggregates previously characterized as spinodal condensates, and small isolated spherical condensate particles ranging from 30-50 nm in diameter[19]. These small spherical condensate particles could grow to micrometer size with prolonged incubation (Fig. 1b). However, reducing the salt concentration to 50 mM Na$^+$ delayed the appearance of similar condensates to >2.5 h of incubation (Fig. 1b). Further reduction to 5 mM Na$^+$ extended the generation of similar condensate particles to ~10 h (Fig. 1b). Similar results were observed by AFM (Supplementary Fig. 1c). Under cryo-ET analysis, these 30-50 nm spherical condensates, displayed the same features observed under physiological salt concentrations (Fig. 1c), corroborating previous cryo-EM study[19]. These experiments indicate that decreasing Na$^+$ concentration can be used to capture the changes in intra-nucleosomal array structure during the rapid process of intra-array compaction before the emergence of spherical condensates. Therefore, we analyzed structures obtained at 5 mM Na$^+$ and 50 mM Na$^+$ after 20 min of incubation.

### TEM image of individual mononucleosome particles

We assembled histone octamers on the 601-nucleosome positioning sequence (NPS), flanked by 200 bp and 100 bp of DNA linker at its entry and exit sites, respectively. To capture 3D snapshot structures of individual nucleosome array particles, it is necessary to directly image

the NCP and the linker DNA without averaging, which can be challenging due to the small diameter and flexibility of individual DNA fragments[24–26]. To overcome this challenge, we employed cryo-ET (Fig. 2a–d) and negative-stain electron tomography (NS-ET) (Supplementary Fig. 2), evaluating the capability of 2D imaging to determine nucleosome positioning along a flexible DNA string. The asymmetric construction enabled us to distinguish the orientation of the nucleosome, a method previously utilized in studies of nucleosome structure[3,7,27–29] and phase separation[17,19,30].

Survey cryo-EM (Fig. 2a–c) and NS-EM (Supplementary Fig. 2a-i) micrographs and zoomed-in images of the particles, revealed that in a low salt buffer (5 mM Na$^+$), mononucleosomes exhibited: (i) a discoidal-shaped NCP with dimensions of ~10 × 10 × 4 nm (Fig. 2b); (ii) DNA linkers of ~2 nm wide (Fig. 2c) with a helical periodicity of ~3.5 nm observed in NS images, corresponding to the double-helix major groove (Supplementary Fig. 2d–f); (iii) nearly two turns of DNA wrapped around the histone disc (orange arrows in Fig. 2c); and (iv) a low-density pore of ~1 nm present near the center of the nucleosomal disk (yellow arrow in Fig. 2c, and Supplementary Fig. 2g-i). These features of the NCPs are consistent with the published nucleosome crystal structure (PDB entry 1AOI[3]).

### 3D reconstruction of individual mononucleosome particle

Using a method that combined individual-particle electron tomography (IPET)[31], image denoising[32], and missing-wedge compensation through low-tilt tomographic reconstruction (LoTToR)[33], we performed 3D reconstruction of an individual mononucleosome by cryo-ET (Fig. 2d) and NS-ET (Supplementary Fig. 2j-l) without any averaging. The resolution of the cryo-ET 3D reconstruction reached 43 Å, as estimated and validated by various criteria and analyses (Supplementary Fig. 3a-c). This reconstruction revealed the discoidal shape of NCPs with two asymmetric DNA linkers (Fig. 2e-g), and almost two turns of DNA around the histone core with clear entry and exit points for the linker DNA. We determined the structure of each particle by rigid-body docking of the NCP crystal structure and flexible fitting of the DNA linkers, connecting them through molecular dynamics (MD) flexible fitting[34] (Supplementary Movie 1). Only the cryo-ET structures were used for further characterization, as the NS-ET structures are somewhat flattened by the thin-layer stain (Supplementary Fig. 2k, l).

For the quantitative characterization of nucleosome morphology, we employed the following parameters: (i) the extent of DNA unwrapping from the histone core (Fig. 2e, between black dashed-lines); (ii) the relative orientation of the two DNA linkers, measured as the $\theta$ angle between two vectors formed by the initial 20 bp DNA segments of the two linkers emerging on the NCP surface (Fig. 2f, red arrows); (iii) the degree of separation (or crossing) of the two DNA linkers in front and side views of the NCP discoidal plane, $\theta_F$ and $\theta_S$, respectively (Fig. 2g); (iv) the orientation of each DNA linker relative to its NCP, defined as (a) the angle $\alpha$ in the $x$-$y$ plane between the DNA vector and a tangent to the NCP disc at the point of contact between the linker and the disc (Fig. 2g, left), and (b) the bending angle $\beta$ between the DNA vector of the emerging linker and the NCP dyad axis in the y-z plane (Fig. 2g, right). These parameters were used to quantitatively infer nucleosome dynamics.

### 3D structural dynamics of mononucleosomes under 5 mM Na$^+$

To investigate the conformational dynamics of nucleosomes using their 3D structures, we reconstruct 47 mononucleosome density maps using cryo-ET (Fig. 2h–k, Supplementary Fig. 3d–f, and Movie 1), followed by the modeling and characterization protocols described above. Aligning the NCP portions of the 47 models (Fig. 2l) unveiled the high flexibility of the two DNA linkers. Analysis showed that the length of the DNA wrapped around the histone octamer varied (Fig. 2m), consistent with previous cryo-EM single-particle averaging analysis[29] and reflects the "breathing" property inherent to NCPs[35].

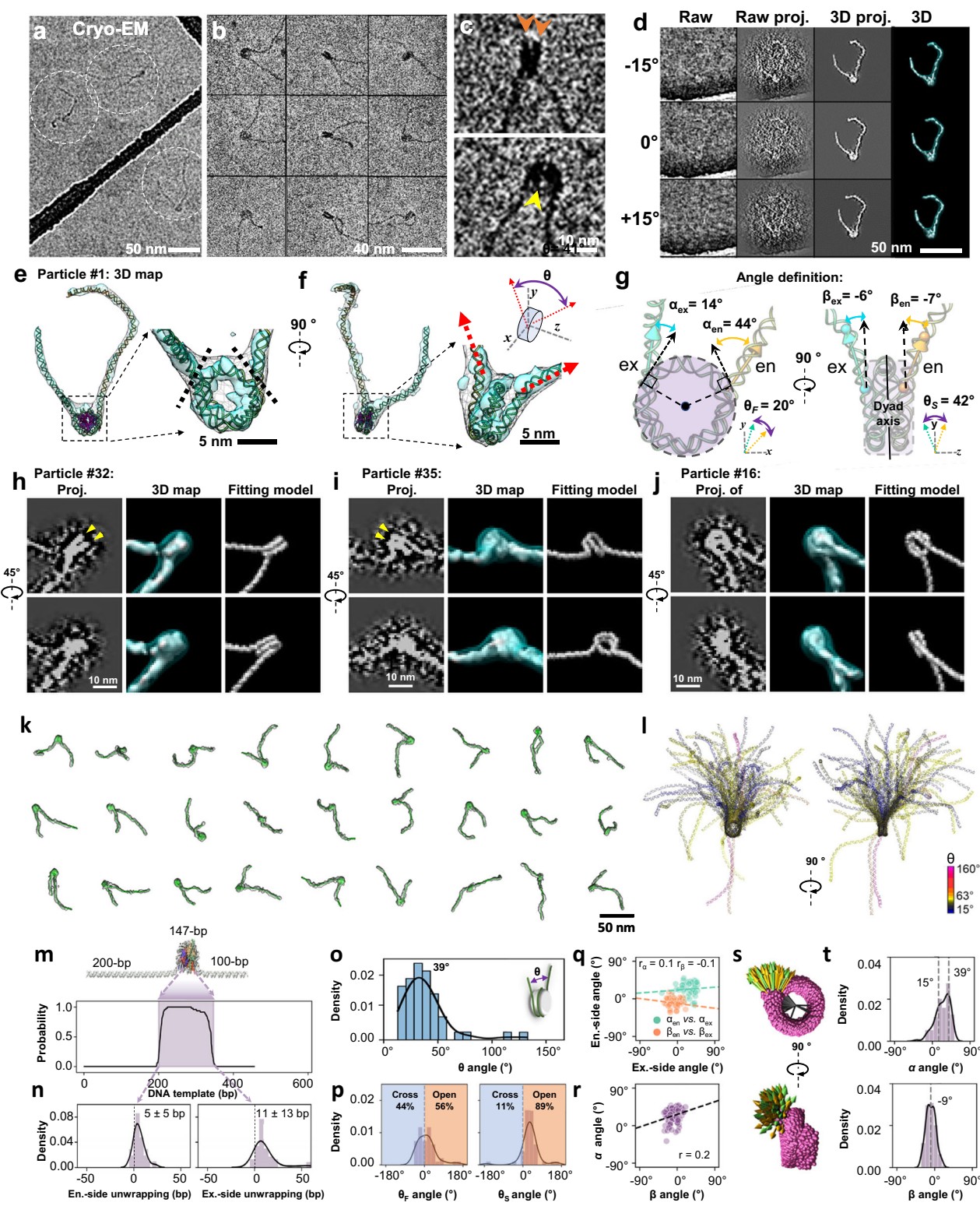

Compared with the crystal structure (1AOI[3]), the nucleosomal DNA ends on the upstream/entry and downstream/exit side of the 601 NPS were unwrapped on average by $5 \pm 5$ bp and $11 \pm 13$ bp, respectively (Fig. 2n), consistent with MD simulations that display DNA unwrapping in 5 or 10 bp steps[36]. The more pronounced unwrapping/rewrapping dynamics of the downstream region of the 601 NPS agrees with single-molecule fluorescence resonance energy transfer (FRET) experiments[37], where the downstream region of the 601 outer wrap

DNA sequence is less flexible and has lower affinity for the histone octamer than the more flexible upstream region.

The broad range of the DNA linker $\theta$ angles, spanning from -0° to 140°, reflects substantial dynamic space coverage (Fig. 2o). The DNA linkers cross more frequently in the NCP front view (44%) compared to the side view (11%), as quantified by the components $\theta_F$ and $\theta_S$ (Fig. 2p). The weak correlation between the $\alpha$ angles of the entry and exit linkers ($\alpha_{en}$ and $\alpha_{ex}$) (Fig. 2q, green points), and between the $\beta$ angles

**Fig. 2 | Cryo-EM 3D structure and dynamics of mononucleosome. a**, Cryo-EM images, and, **b**, nine representative particles of mononucleosome in 5 mM Na$^+$. **c**, Zoomed-in images of two representative particles. Two DNA gyres and the NCP low-density central hole are indicated by orange and yellow arrows, respectively. **d**, 3D reconstruction process of an individual nucleosome particle, showing by raw image (ground-truth), the mask-free projection of the initial 3D map and the masked projection and final 3D map (column 1 through 4, respectively) at three representative tilting angles. In this experiment, 4 cryo-EM grids were prepared, 10 cryo-ET data sets were acquired, and all isolated particles (47 particles) were targeted for 3D reconstructions as showed in the supplementary Fig. 14-60. **e,f**, Perpendicular views of the final 3D map displayed by after (gray) and before (cyan) 45 Å Gaussian lowpass filtering, which superimposed with the flexibly fitted model. **g**, Schematic showing the angles. The orientation of the linker DNA relative to the NCP is defined as the wrapping angle α and the bending angle β, for both entry (cyan) and exit (yellow) side of the NCP. The convention of defining the negative direction as bending away from the NCP is applicable to $\alpha_{en}$, $\alpha_{ex}$, $\beta_{en}$, $\beta_{ex}$, $\theta_F$, and $\theta_s$. **h–j**, Zoomed-in view of the NCP regions of three mononucleosomes, shown with the projection, the final map, and the fitted model, respectively. DNA on the histone is highlighted by yellow arrows. In this experiment, a total of 47 particles were reconstructed as shown in supplementary Fig. 3d-f and 14-60. for statistics. **k**, 30 representative density maps, shown with fitted models. **l**, Super-imposed 47 models aligned based on NCP portion. Models color-encoded by the θ angle. **m**, Histogram of DNA-histone contact positioning. **n**, Histograms of DNA entry and exit positions to/from NCP. The positive values represent the events of unwrapping against the standard model. **o,p**, Histograms of the θ angle and its two planar projections $\theta_F$ and $\theta_S$. **q**, Scatter plot and correlation between the entry and exit DNA arms. **r**, Scatter plot and correlation between the wrapping angle α and bending angle β. **s**, Superimposed vectors of both entry (yellow) and exit (green) DNA linkers on NCP (magenta). **t**, Histogram of the wrapping angle α and the bending angle β distribution measured from all DNA linkers.

($\beta_{en}$ and $\beta_{ex}$) (Fig. 2q, orange points) reflect the independent motion of the two DNA linkers. Additionally, the weak correlation observed between α and β angles within the same DNA linker (Fig. 2r) indicates no strongly preferred "swing trajectory" of the linkers on the NCP surface (Fig. 2s). The α angles exhibit a left-skewed distribution within a range of -± 50°, with major and minor peaks at 39° ± 5° and 14° ± 16°, respectively (Fig. 2t). The β analysis shows a nearly normal distribution within a range of -30°, with a mean of 9° ± 11° (Fig. 2t). A slightly restricted dynamic movement of the DNA arms perpendicular to the NCP discoidal plane can be deduced from the smaller range of the β angles, compared to the α angles.

To characterize the structural diversity of the mononucleosomes, the 47 structures were pairwise aligned by using the minimization of their root-mean-square deviations (RMSD) and then ordered according to a hierarchical classification[38] (Supplementary Fig. 4a). Mononucleosome structures were morphed from the most populated state to the rarest state under a purely geometrical constraint by target molecular dynamic (TMD) simulation. This morphing illustrates the possible interconversion among conformational states adopted by the arrays under non-equilibrium conditions after 20 min incubation and under 5 mM Na$^+$ (Supplementary Movie 1). These results demonstrate the advantage of employing IPET to investigate disordered structures and quantitatively analyze their dynamics and fluctuations.

## 3D structural dynamics of nucleosome arrays under 5 mM Na$^+$

To elucidate the structural rearrangements nucleosomes, undergo within an array, we reconstructed a total of 33, 45, and 31 density maps of di- (Fig. 3a-d, Supplementary Fig. 5a), tri- (Fig. 3e–h, Supplementary Fig. 5b), and tetranucleosomes (Fig. 3i-l, Supplementary Fig. 5c), respectively, under 5 mM Na$^+$. These reconstructions were followed by modeling (Supplementary Movie 2) and quantitative analysis. Our approach enabled us to determine change in structure and dynamics as the number of nucleosomes within the array increased. Analogous to mononucleosomes, these arrays are bordered by 200 bp and 100 bp DNA linkers on opposite ends, with a 40 bp DNA linker separating consecutive 601 NPS. The tomograms revealed nucleosome arrays embedded in -70–90 nm thick ice without evident conformational constraints (Supplementary Fig. 5d), displaying an overall asymmetric zig-zag architecture (Fig. 3a–l). These conformations are reminiscent of previously reported nucleosome fiber structures observed in 2D views of oligo-nucleosome samples[39], slice views of metazoan, pico-plankton, and HeLa cell nuclei[40,41], yet distinct from symmetric structures like the twisted double-helix revealed by cryo-EM SPA[7] and crystallography[6,8].

Similar to mononucleosomes, the average DNA unwrapping length at the NCP entry site in arrays was -5 bp shorter than at the exit site (Supplementary Fig. 6a and b, rows 2-4). This finding suggests that the array breathing motion is also influenced by the asymmetrical stiffness of the 601 sequence[29,36,37]. In arrays, we observed a strong positive correlation between the $\theta_F$ component and NCP number (Supplementary Fig. 7a, row 2-4). $\theta_F < 0$ indicates a 'closed/cross-arm' conformation when viewed from the front of an NCP (Supplementary Fig. 7b, c). The adding of a nucleosome to form a dinucleosome significantly increased the prevalence of 'closed-arm' conformations observed (83%) compared to mononucleosomes (44%). However, this prevalence gradually decreased when arrays contained three (56%) or four (49%) nucleosomes (Supplementary Fig. 7a), suggesting that the reduction observed for $n > 2$ may reflect increased excluded volume effects. Dinucleosomes exhibited a noticeably restricted dynamic range and twisting between two NCP discoidal planes (Supplementary Fig. 7d, panel I). Consistent with the aforementioned results, the α angles of arrays shifted their population towards negative values (minor peak of the distribution Fig. 3c and Supplementary Fig. 8a–d, rows 1 and 2), reaching a maximum for dinucleosomes. Although different from mononucleosomes, tri- and tetranucleosomes showed very similar distributions for α and β angles (Fig. 3c and Supplementary Fig. 8a–d, rows 3 and 4). Changes in wrapping angles, along with linker lengths, were utilized to evaluate the accuracy of nucleosome array modeling (Supplementary Fig. 8e–g).

The distances among the centers of mass of NCPs reflect the array degree of compaction. We found that the numbers of NCPs insignificantly impacts this parameter (Fig. 3d, h, l and Supplementary Fig. 9b, row 1-3). Analysis of the dihedral angle defined by the discoidal planes of $i$ vs. $i + 1$ NCPs in the arrays, confirmed the previous observation that only dinucleosomes exhibit a preferred orientation, with NCPs displaying a more perpendicular arrangement relative to each other (Supplementary Fig. 9d, e, row 1-3). These distinct dinucleosome structural features could be relevant to their recognition by remodelers, as this perpendicular conformation is also observed in the crystal structure of a pseudo dinucleosome bound to Isw1a[42].

Movies of di- and trinucleosomes morphing from the most populated to the least or rare state were created under a purely geometrical constraint by TMD simulation, as used for mononucleosomes (Supplementary Fig. 4b, c and Supplementary Movie 2). These movies illustrate possible conformational fluctuations of the arrays after 20 min incubation and under 5 mM Na$^+$. The structures revealed that, while dinucleosomes often exhibit a crossed arm conformation, tri- and tetranucleosomes showed much less arm crossing and suggest that this trend likely extends to arrays with a larger number of NCPs.

## Intra-array compaction under 50 mM Na$^+$

A tetranucleosome is considered the fundamental unit of chromatin[43] and has been described to adopt higher-order structures such as helical fibers[6,7] and to form phase-separated condensates[19]. To investigate how tetranucleosomes initiate their intramolecular compaction, we imaged and reconstructed them under 50 mM Na$^+$ conditions. (Supplementary Fig. 10a). We acquired a total of 34 tetranucleosome structures (Fig. 3m, Supplementary Movie 3), which exhibit an

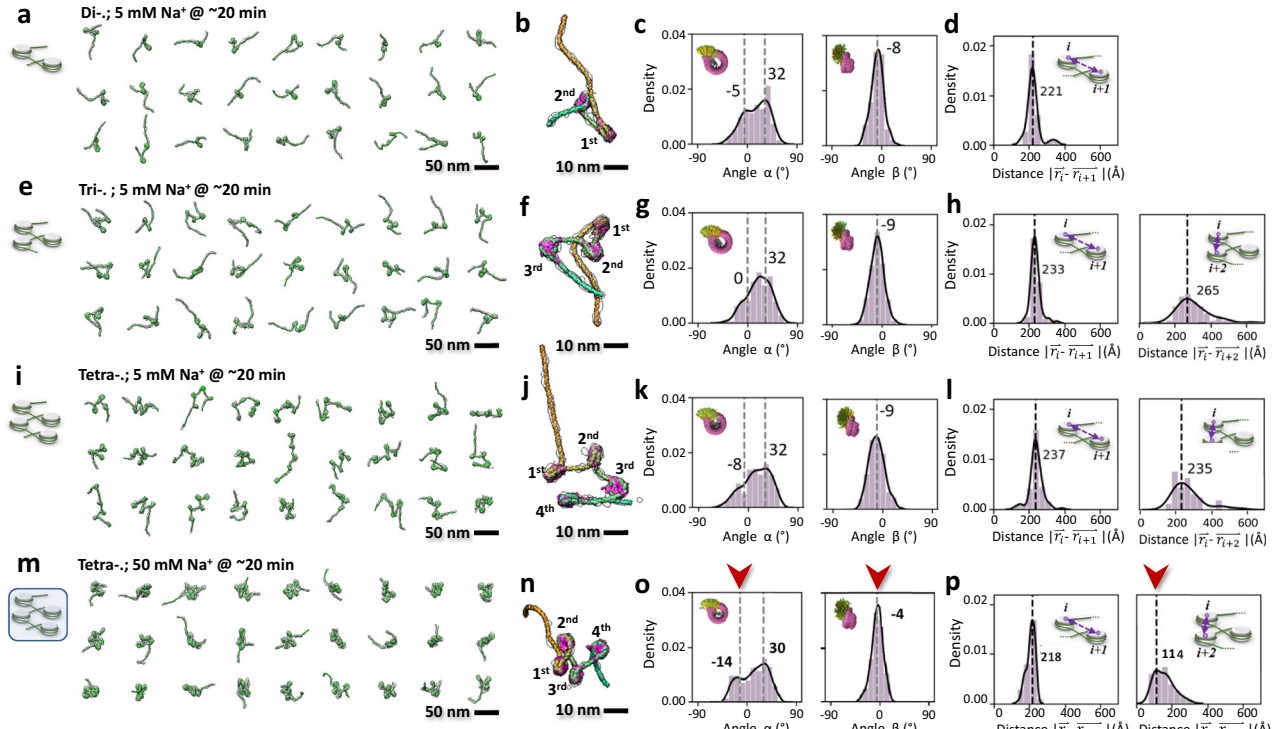

**Fig. 3 | Cryo-ET 3D structures illustrating the dynamics of di-, tri- and tetranucleosome array particles. a**, 27 representative cryo-ET density maps from individual particle reconstructions of dinucleosomes in 20 mM HEPES buffer with 5 mM Na⁺. Each of the density maps is super-imposed with its flexibly fitted model. **b**, Zoomed-in views of a representative map with its fitting model. DNA color-encoded by their bp index (from yellow to cyan) and histone colored in pink. **c**, Histograms of the wrapping angle α and the bending angles β. **d**, Histogram of

the core-to-core distance measured between the $i$ and $i+1$ NCP. **e-h** The structure and dynamics of trinucleosomes under same incubation conditions. **i-l** The structure and dynamics of tetranucleosome under same incubation conditions. **m-p** The structure and analysis of the same tetranucleosome under higher salt condition (50 mM Na⁺) after the same incubation time. The red arrow indicates the major change of the peaks.

irregular zig-zag arrangement but display a more compacted shape compared to those observed at 5 mM Na⁺. Additionally, the unwrapping at the exit site slightly reduces (from $14 \pm 12$ bp to $9 \pm 8$ bp), indicating a more stable histone-DNA interaction (Supplementary Fig. 6a, b, row 4-5). In 50 mM Na⁺, the distribution of $\theta_F$ and $\theta_S$ showed notable peak shifts toward negative values, corresponding to a greater proportion of "closed-arm" conformations (Supplementary Fig. 7a, row 4-5). This effect led to DNA linkers lying closer to their anchored NCP, reflected in a decrease in α and an increase in β (Fig. 3o and Supplementary Fig. 8c, d, row 4-5). These changes are likely a direct result of a stronger electrostatic screening, which reduces repulsive interactions between negatively charged DNA linkers, facilitating their closer proximity on the same NCP[44].

A marginal reduction (1.9 nm) in the distance between consecutive NCPs was observed when increasing the salt concentration from 5 mM to 50 mM Na⁺, probably reflecting the rigid nature of the 40 bp DNA linker whose persistence length varies between 150 bp and 120 bp in these ionic conditions[45]. A more substantial reduction in distance (15.1 nm) was observed between every-other NCPs, as well as between the first and fourth NCPs (11.9 nm) (Fig. 3p and Supplementary Fig. 9b, row 3-4). These changes indicate a significant compaction of the array conformations (Fig. 3m). In 50 mM Na⁺, the dihedral angle between NCPs shifts towards a more perpendicular orientation for consecutive NCPs and to a more parallel orientation between every-other NCPs, compared to observations at 5 mM Na⁺ (Supplementary Fig. 9e, row 3-4). Although a better electrostatic screening should result from the higher ionic strength, so that the two DNA linkers could approach each other, facilitating NCP alignment, direct contact or stacking of the NCPs was not observed in 50 mM Na⁺.

## H1-induced DNA invasion into neighboring NCPs

Given that the linker histone H1 accelerates tetranucleosomes phase separation[19] and that its binding affects the linker trajectory of the array[46], we aimed to determine whether H1 could induce compaction in loosely-packed nucleosome arrays under 5 mM Na⁺, akin to the effect observed at higher salt concentrations (Fig. 3m). Tetranucleosomes were incubated with H1 in 5 mM Na⁺ using a 1:4 molar ratio for 20 min at room temperature (Fig. 4a). Interestingly, visual inspection indicated that arrays in the presence of H1 do not attain the same level of compaction at 5 mM Na⁺ as observed in its absence at higher salt concentration (50 mM Na⁺) (Fig. 4a, Supplementary Fig. 10). However, the simultaneous presence of both factors, H1 and higher salt conditions, under the same incubation time and temperature, induces array condensation (Fig. 4b, c, and Supplementary Fig. 11), preventing the structural determination of each individual array. Thus, we can only compare structural changes observed with and without H1 at 5 mM Na⁺.

We obtained a total of 33 cryo-ET density maps with their models from the tetranucleosome sample incubated with H1 in 5 mM Na⁺ for 20 min under room temperature (Fig. 4d, Supplementary Movie 3). Closer examination showed that 40% of the particles exhibited a conformation in which the 200 bp DNA linker arm extruding from one end of the array interacts with another NCP in the array (Fig. 4e). The probability of this interaction occurring with the second NCP was notably high (30%), in contrast to the third and fourth NCPs (3% each, Fig. 4e, panel II). No interaction was detected between the 100 bp DNA linker arm and NCPs.

To assess whether, in the presence of H1, the 200 bp DNA linker interacts with the NCP surface or invades it by displacing a segment

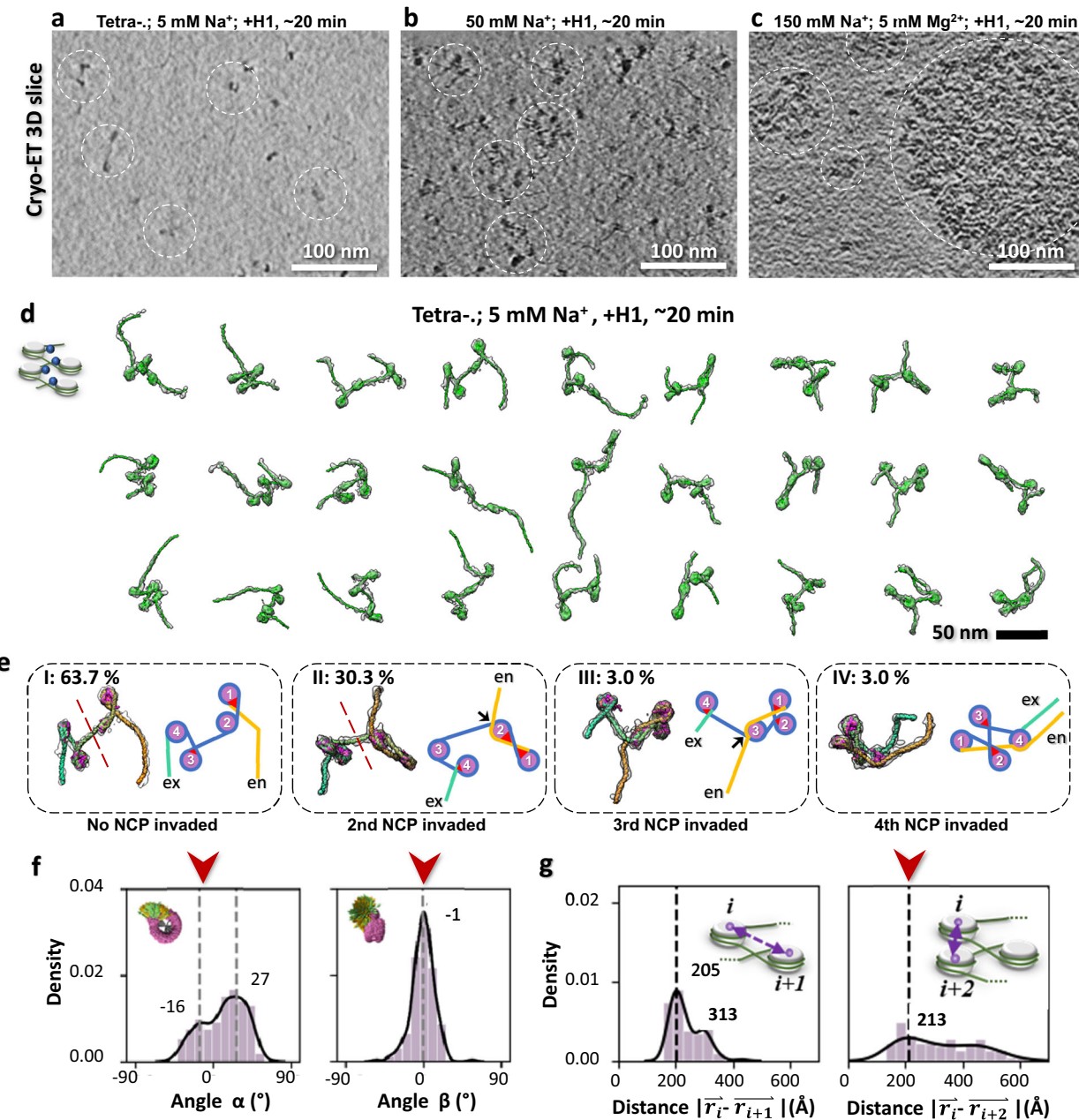

**Fig. 4 | Morphology and 3D structures of tetranucleosome arrays in the presence of H1. a-c**, Cryo-ET central slices s show the morphology of tetranucleosome arrays incubated with 5 mM Na⁺, 50 mM Na⁺ and physiological salt, respectively, for 20 min, in the presence of linker histone H1. In these experiments, a total of 3 cryo-EM grids for each condition were prepared, and 4-5 grids areas on each grid were imaged by cryo-ET, in which 4 tilt-series have been used for 3D reconstructions. **d**, 27 representative cryo-ET 3D density maps and models of tetranucleosome with 5 mM Na⁺ in the presence of H1. **e**, Zoomed-in views of four representative tetranucleosome array maps and models (left side of each panel) displaying four typical (I, II, III, and IV) NCP unwrapping/rewrapping conformations

in response to the presence of H1. The schematics (right side of each panel) illustrate various 200-bp DNA arm trajectories during its invasion into one of the intermediate NCPs. The entry-, intermediate-, and exit-DNA portion are colored in orange, blue, and green, respectively. Histones are colored in purple. A red dashed line indicates that the second NCP unwrapping results in two spatially separated dinucleosomes. Red triangles mark the conventional H1 binding sites on NCP, while black arrows point to the possible H1 binding sites introduced by the distal DNA arm invasion. **f**, Histograms of α and β angle distributions and, **g**, core-to-core distance distributions measured from tetranucleosome NCPs in the presence of H1.

of the wrapped 601 NPS, we examined the DNA unwrapping of NCPs. If DNA linker arm invasion occurs, the invaded NCP should exhibit significant unwrapping on one side, resulting in a linker length >40 bp. Our results confirm this inference, showing significantly increased exit-site unwrapping, which manifests as the emergence of a second peak at ~45 bp in the unwrapping length distribution (Supplementary Fig. 6a-b, row 6). This sub-population of 45 bp-unwrapping was mainly contributed by the second NCP,

which displays a longer 40 bp linker in the 3D reconstructed maps (Supplementary Fig. 6c). Since a 45 bp unwrapping is substantial, we investigated whether the H2A-H2B heterodimer becomes fully exposed or protected by the invading arm[47,48]. Our analysis of aligned and averaged density maps of the unwrapped second NCP (Supplementary Fig. 6d) showed no missing DNA density, indicating that the invading DNA arm has indeed rewrapped around the histone surface.

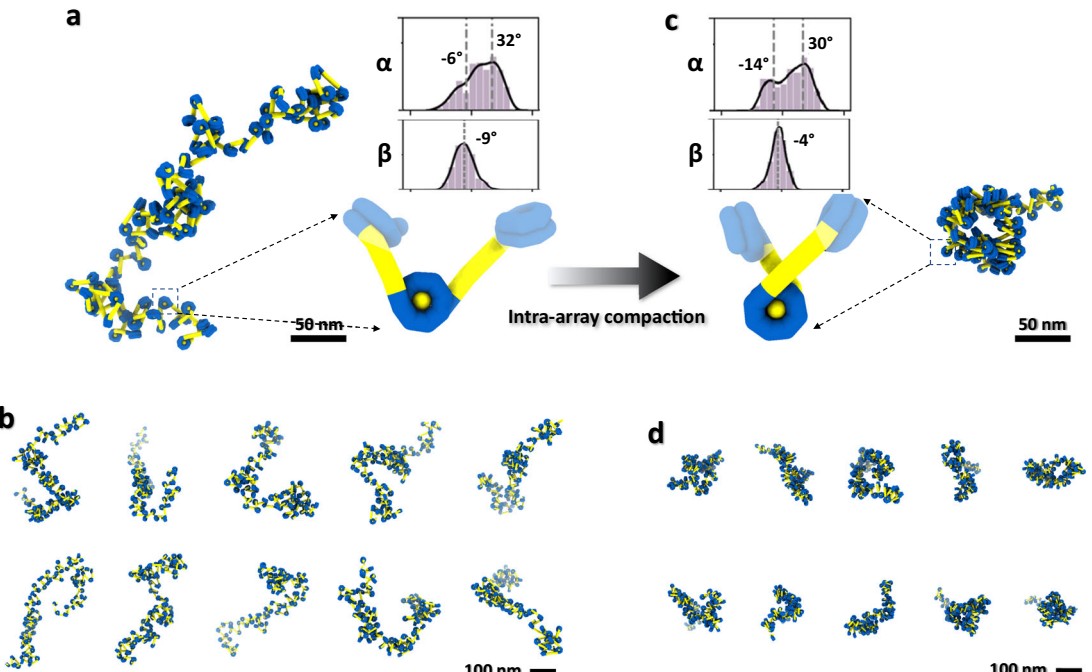

**Fig. 5 | Simulating chromatin morphology and conformational regulation mechanism through linker DNA angles. a**, In silico assembly of a chromatin fiber by connecting 100 NCPs (top panel) using the α, β angle distribution (top right panel) and the NCP unwrapping distribution derived from experimental statistics of tetranucleosome arrays in 5 mM Na⁺. Zoomed-in view of the spatial orientation of a representative NCP (in cyan) with its linker DNA (in yellow). **b**, Ten additional simulated chromatin fiber (100 NCPs) generated from the same distribution of experimental data. **c**, In silico assembly of a chromatin fiber (connecting 100 NCPs) based on the experimental distributions of angles and the NCP unwrapping levels measured from the tetranucleosome sample with 50 mM Na⁺ after 20-min incubation. Zoomed-in view of the NCP (in cyan) with its linker DNA (in yellow). **d**, Ten more chromatin fibers generated based on the same distribution.

Due to the structural rearrangement of the arrays caused by the invasion of the 200 bp arm in the presence of H1, both the θ angle and its subcomponents relative to the discoidal plane ($\theta_F$ and $\theta_S$) exhibit a distinct bimodal distribution, displaying a major and a minor peak (Supplementary Fig. 7a, row 6). The position of the major peak corresponds to the "closed-arm" NCPs, similar to those observed in 50 mM Na⁺, and that of the minor peak corresponds to a "wide-open arm" NCP population (arms angle approaching 180°), resulting in the 45 bp unwrapping induced by the 200 bp arm invasion (Supplementary Fig. 7a, row 6, red and green arrows). Classification analysis (Supplementary Fig. 12, orange arrows) confirmed that "wide-open arm" NCP populations mainly arise from the second NCP (Fig. 4e, II). In line with the dominant population of "closed-arm" NCPs, both α and β angles indicate that DNA linkers turn towards their own NCP in the presence of H1 (Fig. 4f and Supplementary Fig. 8c–d). This configuration results in the formation of a "triangular" shape between the NCP and its two linkers that accommodates the binding of H1 (Fig. 4e, red triangles).

Apart from its effect on the second NCP, the remaining changes resulting from the binding of H1 resembled those induced by high ionic strength. This observation suggests that H1 under 5 mM Na⁺ conditions can selectively regulate the compaction of only a part of the arrays. We hypothesize that this phenomenon arises from the interplay between H1 binding and the internal repulsion within the array. H1 binding necessitates the "closed-arm" NCP conformation. If all four NCPs assume this conformation, the array would adopt a compact structure like those observed in 50 mM Na⁺ or in crystal structures of arrays[8]. However, due to incomplete electrostatic screening, repulsion among NCPs remains potent. Accordingly, a conformation where the second NCP partially unwraps (forming two dinucleosome regions, Fig.4e, panel I and II, red-dashed line) represents a way to satisfy both scenarios simultaneously: adopting a "closed-arm" conformation for most NCPs (3 out of 4), while maintaining a separation distance

between every other NCPs, thus minimizing the free energy of the system. Our NCP distance calculation supports this hypothesis, showing that the mean distances among NCPs (23.8 nm, 32.1 nm, and 41.6 nm between $i$ and $i+1$, $i$ and $i+2$, and $i$ and $i+3$ NCPs, respectively, Fig. 4g and Supplementary Fig. 9b row 5) are comparable to array distances observed at 5 mM Na⁺ without H1.

## Spatial distribution of the linker DNA regulates chromatin morphology

The statistical parameters obtained in this study (Supplementary Data 1) provide a comprehensive characterization of the intra-array compaction of individual nucleosome arrays (tetranucleosomes) prior to the ensuing of inter-array interactions[19]. It is interesting to extrapolate our findings from short arrays to longer ones, offering insights into the possible adoption of a loosely-packed state in the chromatin fiber. Arrays consisting of 100 NCPs (hectanucleosomes) were computationally generated using the structural parameters identified in this study. Analysis of the simulated hectanucleosomes based on the α and β angles measured at 5 mM Na⁺ revealed curvy, irregular fibers (Fig. 5a, b) with an average length and width of 561 ± 44 nm and 40 ± 5 nm, respectively (Supplementary Fig. 13a, row 1). Based on the angle measured at 50 mM Na⁺, both the chromatin length and width decreased under these latter conditions to 395 ± 35 nm and 32 ± 4 nm, respectively (Fig. 5c, d and Supplementary Fig. 13a, row 2). Further simulations, which prevented DNA unwrapping while retaining the DNA linker angles relative to the NCP, resulted in only minimal changes in fibers dimensions by 2-11% (Supplementary Fig. 13a, row 3-4), highlighting the significance of DNA linker angles as the main factors determining the morphology of longer arrays (Fig. 5a and c).

Additionally, by extending our simulations to encompass a larger structure (250,000 NCPs, analogous to a mini chromosome with 50 million base pairs of DNA), we explored the tertiary structure of these simulated fibers. This larger chromatin structure displays a relatively

dispersed spatial organization, featuring both low-density and high-density regions (Supplementary Fig. 13b, c), closely resembling the chromatin domain morphology observed in studies using fluorescence 3D imaging[49] and immuno-gold labeling[50] of interphase nuclei.

## Discussion

Despite the nuclear environment role in facilitating chromatin condensation[51,52], the loosely-packed "10 nm-fiber" structure is predominantly observed within specific nuclear domains and extracts[41]. This phenomenon results from a complex interplay of factors, including nuclear proteins, histone acetylation, ionic strength, pH, and temperature[17,19,53], which collectively influence the timing and spatial distribution of chromatin compaction and decompaction. In our in vitro experiments, low ionic strength conditions (5 mM and 50 mM Na⁺) were deliberately used to induce the loosely-packed 10 nm array fibers state and to inhibit inter-nucleosome array interactions. Utilizing the cryo-ET individual-molecule approach, we have determined the structure of mononucleosomes as well as di-, tri-, and tetranucleosome arrays under these conditions, without applying structural averaging (Supplementary Data 1). Our study revealed that the critical determinants of the structural changes observed are the angles between the DNA linkers and the NCPs. In particular, the angle α between the DNA vector of the linker and the tangent to the NCP disk at their point of contact plays a pivotal role, mediating the level of intra-array compaction and resulting in various tetranucleosome conformations. Factors such as nuclear proteins, salt concentration, histone acetylation, and temperature, likely affect the magnitude of this angle, thereby affecting chromatin structure.

Interestingly, upon binding, histone protein H1 induces significant changes in the angle between the DNA linkers flanking an NCP[46], which can cooperatively modulate nucleosome array compaction[8,28]. In the presence of H1, three of the four nucleosomes in the tetranucleosome arrays adopt a closed-arm conformation, similar to that observed at higher ionic strength (50 mM Na⁺). In contrast, they exhibit significant unwrapping of the second NCP, which is observed to be a frequent target for invasion by the distal DNA linker. Although the invasion phenomenon, as described here, is an "end effect", it is possible that regions of DNA flanked by longer linkers could be involved in similar invasion events. This invasion favors the creation of additional DNA junctions, which could serve as extra binding sites for H1 (Fig. 4e, II and III, black arrow) and to increase NCP connectivity. At 50 mM Na⁺, H1 already accelerates inter-array interactions (Supplementary Fig. 11), and at higher ionic conditions (150 mM Na⁺, 5 Mg²⁺), it strongly catalyzes the formation of spherical liquid droplets (Supplementary Fig. 11), consistent with its role in promoting nucleosome LLPS and strengthening the inter-array network[54]. In the condensation regime, long DNA arms become entangled, complicating structural analysis of individual particles by cryo-ET.

The "30-nm" chromatin structure has been extensively debated, with cryo-EM studies reporting both a relatively homogeneous distribution of NCPs[12,26] and the presence of fiber-like structures within cells[55,56]. This structural diversity likely reflects the intrinsic plasticity of the chromatin fiber, which in the cell could also be subjected to modifications such as acetylation and methylation[17]. Additionally, the discrepancy may result from whether the study was performed in vitro[7,26,56] or in vivo[15,16,40,56,57]. Specifically, the existence of long-range regular 30-nm fibers observed in vitro still lacks supporting evidence in vivo. In this study, we observed statistical changes in both α and β angles on tetranucleosomes during the array compaction. To understand how these linker DNA angles regulate the overall morphology of chromatin, such as inducing the "30-nm" fiber feature, we conducted simulations of longer array containing 100 NCPs using the parameters derived from the tetranucleosome arrays in 5 mM and 50 mM Na⁺, respectively (Fig. 5). The significant increase in the density of the simulated chromatin arrays, accompanied by the emergence of highly-irregular "30-nm" fibers (Supplementary Fig. 13a), indicates that the DNA linker angles play a pivotal role in shaping chromatin architecture. A further test was conducted by simulating a mini-chromosome threading 250,000 NCPs using the same angle parameters. The distribution of NCPs within a thin slab (25 nm thickness) in the dense region (Supplementary Fig. 13b, c, right panel) displays a morphology and density similar to those observed in in-situ cryo-ETs, such as the tomographic slices of HeLa cells (10-20 nm thickness)[40], chicken erythrocyte nuclei (10-50 nm thickness)[55], picoplankton cell lysates (30 nm thickness)[41], and in 2D views of swollen HeLa S3 chromosomes at low Mg²⁺ concentration[12]. Other parameters, such as physiological salt, temperature, involvement of binding proteins, and nucleosomal phasing, have not yet been incorporated into the simulation. However, the likeness between the structures observed experimentally and those generated here, highlights the importance of the entry and exit linker angles in regulating the overall morphology of chromatin prior to phase transition.

Together, these insights into the conformational changes of nucleosomal arrays advance our understanding of how their large-scale, global morphology is controlled by changes in the individual units and provide a more comprehensive view of chromatin structural transition and regulation.

## Methods

### Expression and purification of histones

Vectors containing the genes of *Xenopus laevis* histones H2A, H2B, H3, and H4 under the control of a T7 promoter were expressed in *E. coli* BL21(DE3) and purified as previously described[58]. For each histone, a single *E. coli* BL21(DE3) colony was grown at 37 °C in LB media supplemented with 100 μg/uL ampicillin and 25 μg/uL chloramphenicol to an OD₆₀₀ of ~0.6, and histone expressions were induced by adding 1 mM Isopropyl b-D-1-thiogalactopyranoside (IPTG) to the cell culture. After 3 h, the cell culture was centrifuged at 7000 × g for 30 min, and the cell pellet was suspended in ice-cold wash buffer (50 mM Tris-HCl pH 7.5; 100 mM NaCl; 1 mM EDTA; 5 mM 2-mercaptoethanol (BME); 1% Triton X-100 [w/v]; and protease inhibitors, Roche). This procedure was repeated twice to wash away LB medium. The cell pellets were suspended in 4 volumes of wash buffer, flash freezing with liquid nitrogen, and stored at -80 °C for later use. To purify the inclusion bodies, suspended cell pellets were thawed and sonicated on ice seven times with 20-s bursts at 7.0 W with a Misonix 2000 sonicator. The lysate was centrifuged at 30,000 x g for 1 h at 4 °C, and the pellets were rinsed by suspension with wash buffer. This procedure was repeated three times. To remove Triton X-100, pellets were rinsed by suspension using washing buffer without the detergent. Suspended pellet was centrifuged at 30,000 x g for 1 h at 4 °C and this procedure was repeated three times. Inclusion bodies were then solubilized in 20 mM Tris-HCl pH 7.5; 8 M Urea; 1 mM EDTA; 10 mM DTT, and purified by anion and cation exchange. Purification was checked by 15% SDS-PAGE and fractions containing pure histones were pooled and dialyzed against 1 L of 10 mM Tris pH 8.0. Histones were centrifuged to remove aggregates, concentrated (~10 mg/mL) using a 10 K Amicon Ultra-15, lyophilized, and stored at -80 °C for later use.

### Histone octamer purification

The reconstitution of the *X. laevis* histone octamer was performed as following. Lyophilized histones were solubilized with unfolding buffer (20 mM Tris-HCl pH 7.5; 7 M guanidine hydrochloride; 10 mM DTT), and combined at a H2A:H2B:H3:H4 ratio of 1.2:1.2:1:1. The volume was adjusted to a total histone concentration of 1 mg/mL and it was incubated with mixing for 3 h at room temperature. Solubilized histones were dialyzed four times for 12 h each against 1 L of refolding buffer (10 mM Tris-HCl pH 8.0; 2 M NaCl; 1 mM EDTA; 5 mM DTT) using a 3.5 kDa dialysis membrane at 4 °C. Refolded histone solution was centrifuged at 100,000 x g for 30 min to remove aggregates,

concentrated by centrifugation to ~0.5 mL using 10 kDa Ultra-15 (Milipore), and loaded onto Superdex 200 Increase 10/300 GL (Cytiva) previously equilibrated with refolding buffer. Fractions were checked by 15% SDS-PAGE and the gel was stained with AcquaStain protein staining (Bulldog Bio). Fractions containing three bands (corresponding to H3, H2A/H2B, which, because of their similar size they are not resolved in the gel, and H4) and in equimolar quantities (based on the intensity of the three bands) were pooled, concentrated to ~8-10 mg/mL using 30 kDa Ultra-15 (Milipore), aliquoted, and flash-frozen with liquid nitrogen, and stored at -80 °C for subsequent use. The synthesis of the H2A-H2B dimer followed the same procedure utilized for octamer reconstitution, with H2A and H2B combined at a ratio of 1:1.

## Synthesis of DNA templates

The mono-, di, tri-, and tetranucleosome DNA templates consists of one, two, three, and four repeats of the 601-nucleosome positioning sequence (NPS), respectively. Each template is flanked to the left by 200 bp DNA and to the right by 100 bp DNA, and the 601 NPS in di-, tri-, and tetranucleosome templates are separated by a 40 bp linker length. Mononucleosome DNA templates were amplified by PCR from the pGEM-3Z/601 plasmid (addgene) and cloned back into the PGEM 601 vector using primers containing the restriction recognition site for the *BsaI* enzyme (NEB). Di-, tri-, and tetranucleosome DNA templates were synthetized by ligation of two, three, or four PCR products containing one 601 NPS each. The PCR products were amplified using the pGEM-3Z/601 plasmid (Addgene) and primers containing the restriction recognition sites for enzymes *BsaI* and *BbsI* (NEB). In our design, the di-, tri-, and tetranucleosome DNA are flanked by *BsaI* restriction sites, and the *BbsI* sites were used to ligate the two, three, or four PCR products, respectively. To generate the full templates, the PCR products were digested first with *BbsI* and ligated at an equimolar ratio using *E. coli* DNA ligase (NEB). The longest product of each ligation, corresponding to the di-, tri, or tetranucleosome DNA, were purified using 0.8% agarose gels, digested using the *BsaI* enzyme, and cloned into *BsaI*-restricted pGEM-3Z/601 plasmid using T4 DNA ligase (NEB). Ligation product was transformed into *E. coli* DH5α for plasmid extraction by miniprep. Each ligation product was checked by DNA sequencing. Plasmid containing the mono-, di-, tri-, and tetranucleosome templates were grown in *dam⁻/dcm⁻ E. coli* (NEB), purified by maxiprep, and excised by restriction with *BsaI*. Mononucleosome DNA template was purified from the vector backbone by 5% preparative acrylamide electrophoresis using a Model 491 Prep Cell (Bio-Rad) electroelution system. Di-, tri-, and tetranucleosome templates were purified by 4% preparative acrylamide electrophoresis.

## Nucleosome assembly and purification

*X. laevis* histone octamer and mono-, di-, tri, and tetranucleosome DNA templates were combined at a ratio of 1:1.2, respectively, in high-salt buffers (10 mM Tris-Cl pH 8.0; 2 M NaCl; 1 mM EDTA; 0.5 mM DTT; and 1 mM PMSF) at a final DNA concentration of 100 ng/uL. In the case of the arrays, H2A-H2B heterodimer dimer was also incorporated at a molar ratio of 0.2 compared to the octamer. Assembly solutions were dialyzed using a 3.5 kDa dialysis membrane at 4 °C against 500 mL of high-salt buffer for 1 h at 4 °C, followed by a 36-h lineal gradient dialysis against 2 L of low-salt buffer (10 mM Tris-Cl pH 8.0; 1 mM EDTA; 0.5 mM DTT; and 1 mM PMSF) with continuous stirring. A final dialysis of 3 h was performed against 500 mL of low salt buffer. Nucleosome reconstitutions were checked by 4% acrylamide and 0.2X TBE buffer (Tris-borate-EDTA) native electrophoresis. Mononucleosomes were purified from hexasomes and bare DNA by 4% preparative acrylamide (59:1 acrylamide:bisacrylamide) electrophoresis using the 491 Prep Cell (Bio-Rad). Di-, tri-, and tetranucleosomes were purified by 10-40% lineal sucrose gradient (20 mM HEPES-NaOH pH 7.5; 1 mM EDTA; 1 mM DTT) and centrifuged at 150,000 x g for 16 h at 4 °C, using an ultracentrifuge Beckman Optima MAX-XP with the rotor MLS-50 (Beckmann), as previously described[19].

## Specimen preparation

**Cryo-EM samples.** The cryo-EM specimens were prepared following the procedure described before[31]. Briefly, an aliquot (~3 μl) of incubated nucleosome array sample at ~40 nM was placed onto the 200 mesh Quantifoil copper grid (Q210CR-06, Electron Microscopy Sciences) that had been glow-discharged (PELCO easiGlow™ Glow Discharge Cleaning System) for 15 s. After incubating for ~10 s, the grid was flash-frozen in liquid ethane at ~90% humidity and 4 °C with a Leica EM GP rapid-plunging device (Leica, Buffalo Grove, IL, USA) after being blotted with filter paper with a controlled blotting time (2 s). The flash-frozen grids were transferred into liquid nitrogen for storage.

**NS-EM samples.** The NS-EM specimens of nucleosome array sample were prepared as described in a previously published protocol[59]. In brief, the samples were diluted to ~20 nM with sample buffer. An aliquot (~4 μL) of diluted sample was placed on an ultra-thin carbon-coated 200-mesh copper grid (CF200-Cu-UL, Electron Microscopy Sciences, Hatfield, PA, USA) that had been glow-discharged for ~15 s. After ~1 min incubation, the excess solution on the grid was blotted with filter paper. The grid was then washed with water and stained with 1% (w/v) uranyl formate (UF) before air-drying with nitrogen. For the time series incubation experiment, tetranucleosome array samples were diluted to 30 nM in 20 mM HEPES buffer under three different salt conditions (5 mM NaCl, 50 mM NaCl, and 150 mM NaCl). At the specified time point (2 min, 20 min, 60 min, 2.5 h, 5 h, and 10 h), 4 μL of the sample was used to prepare the NS-EM sample described above for each of the incubation solutions.

**AFM samples.** The AFM samples were prepared using tetranucleosome array samples. A muscovite mica disc (diameter 9.9 mm, Ted Pella, Inc.) was freshly cleaved and used as a supporting surface. Twenty microliters of poly-L-lysine solution (0.1% w/v, Mw 150,000-300,000, Ted Pella, Inc) was placed on the mica surface for 5 min and was then thoroughly rinsed with water and dried by a stream of nitrogen gas. After diluting nucleosome array to 30 nM in 20 mM HEPES buffer with 5 mM NaCl, 4 μL of the sample was deposited onto a poly-l-lysine pre-treated mica surface for a minute. The samples were prepared after 5 min, 75 min, 4 h, and 12 h incubation. The non-adsorbed nucleosome molecules were removed by washing with 50 microliter of background solution for 6 times. The samples were placed into an AFM equipped with a liquid cell. After the thermal relaxation for 10 min, the AFM images were collected at different locations of sample.

## TEM data acquisition

Cryo-EM specimens were screened by Titan Krios G2/G3 TEMs (Thermo Fisher Scientific, US) transmission electron microscope operated at 300 kV high tension with a Gatan energy filter. The untitled cryo-EM micrographs were acquired with a Volta phase plate near the focal plane using a Gatan K2 Summit direct electron detection camera under a magnification of ~81 kx (each pixel of the micrographs corresponds to ~0.9 Å in specimens). The particles with a box size of 800 × 800 pixel will be used to display the morphology. Cryo-EM tilt image series of the samples were collected from -60° to +60° at 3° increments on a Titan Krios G2/G3 TEM equipped with a Gatan energy filter and a K2/K3 Summit direct electron detection camera operated under 300 kV high tension. During data acquisition, the SerialEM 4.1[60] software was used to automatically track the specimen and maintain defocus at ~2.5 μm. The acquired tilt image series at magnification of ~81/50 kx (each pixel corresponds to 1.46/1.67 Å for K2/K3 camera) represents a total dose of ~102 - 183 e-/Å². For each tilt angle, a total number of ~8 frames were collected under the exposure of 0.25 s per frame.

NS-EM specimens were screened by using Zeiss Libra 120 Plus and 200MC TEMs (Carl Zeiss NTS, Germany) operated at 120 and 200 kV high tension with a 10-20 eV energy filter. The negative staining micrographs were acquired under defocus at -0.6 μm using a Gatan UltraScan 4 K × 4 K CCD under a magnification of 125 kx and 160 kx (each pixel of the micrographs corresponds to -0.94 Å and -0.74 Å in specimens respectively). Tilt image series of mononucleosome samples were collected from -60° to +60° in 3° increments using a Zeiss Libra 120 Plus TEM (Carl Zeiss NTS, Germany) equipped with an in-column energy filter and a Gatan UltraScan 4 K X 4 K CCD. During data acquisition, the Gatan tomography module (Gatan Inc., Pleasanton, CA, USA) operated in Advanced Tomography mode was used to track the specimen and maintain defocus at -2.0 μm. The acquired tilt image series at magnification of 50 kx (each pixel corresponds to 0.24 nm) represents a dose of ~40 e⁻/Å² per tilt image.

## AFM data collection

All in situ AFM images were collected in tapping mode at room temperature (23 °C) with a NanoScope 8 atomic force microscope (J scanner, Bruker) and with hybrid probes consisting of silicon tips on silicon nitride cantilevers (SNL-10 triangular lever, $k = 0.35$ N/m, tip radius <10 nm, resonance frequency 65 kHz in air, Bruker). The drive amplitude was about 20 nm in fluid, and the signal-to-noise ratio was maintained above 10. The scanning speed was 1 Hz. The amplitude set point was carefully tuned to minimize the average tapping force (~50 pN) during in situ imaging. The images were analyzed with the Nanoscope Analysis 1.40 image processing software package from Bruker.

## Image preprocessing

The motion of the cryo-ET frames was corrected by MotionCor2[61]. To reduce the cryo-ET image noise, we followed a machine learning method (NOISE2NOISE method) as described[32]. The defocus value of both cryo-ET and NS-ET tilt series was measured by using GCTF[62]. During data collection, a carbon area perpendicular to the tilt axis was included to aid in the detection of contrast transfer function (CTF) when particles were scarce. The phase and amplitude of the determined contrast transfer function (CTF) were corrected by TOMOCTF[63] based on strip-based periodogram averaging method (deltaD = 1000, w1 = 0.7, w2 = 0.25). The tilt series were initially aligned by using IMOD[64]. By using boxer from EMAN1 (ver. 1.90) software[65], a box of 256 × 256 pixels was used to select the particles of nucleosome from the -113 micrographs imaged under 125 k× magnification, while a box of 200 × 200 pixels was used to select the segments of the upstream (or entry) and downstream (or exit) DNAs from the -150 micrographs imaged under 160 k× magnification. All NS-EM particles were masked using a round mask generated from SPIDER software after a Gaussian high-pass filtering. The reference-free class averages of particles were obtained using refine2d (EMAN1 software) based on 30,540 particles of DNA segment and 13,029 particles of NCPs.

## Individual particle electron tomography (IPET) 3D reconstruction

By labeling a marker on the particle center, all nucleosome arrays within the scope of the large tomogram are manually selected without bias. The tilt series of each of the targeted particles were semi-automatically tracked and then windowed in square windows of ~ 1000 ×1000-pixel size using IPET software[31], before being binned four to five times to reduce computation time in subsequent reconstructions. Following the pipeline of IPET reconstruction[31], a local tilt series images containing a single nucleosome particle was extracted from the IMOD-aligned full-size tilt series to perform focused ET reconstruction (FETR), which can increase the tilt series alignment accuracy by reducing the effects from the large image distortion[31]. Briefly, an ab initio 3D density map was directly back-projected in Fourier space and served as an initial model. The refinement was then iteratively invoked to translationally align each tilted particle image to the computed projection. During the refinement, automatically generated Gaussian low-pass filters, soft-boundary circular masks and loose particle-shaped masks were sequentially applied to the tilt images and references to increase the alignment accuracy[31]. An improved model was then reconstructed based upon the refined alignment at the end of each refinement iteration. The 3D map was then reconstructed by back-projection of the filtered and masked particle tilt series. The final back projection was performed in Fourier space without weighting. To reduce the artifact caused by the limited tilt angle range, the final 3D map was submitted for a published missing-wedge correction method by LoTToR[33], in which the low resolution mask used was generated by the Model-Based Iterative Reconstruction (MBIR) method[66]. All final density maps were low-pass filtered to 4.5 nm using EMAN[65] 'lp' function with a Gaussian filter and displayed using UCSF Chimera (ver. 1.16)[67].

## Estimation of the reconstruction resolution

The resolution for the reconstructions were estimated by two methods. (i) Data-to-Data based analysis: the Fourier Shell Correlation (FSC) was calculated between two independently reconstructed 3D maps, in which each map was based on one-half of the tilt-series (split by even and odd tilt index) after particle alignment refinement of the IPET[31]. The frequencies at which the FSC curve first falls to values of 0.5 and 0.147 were used to represent the reconstruction resolution. Notably, the resolution estimated by this method could be severely under-estimated since the reconstruction from one half of the tilt-series significantly reduced the quality of the map compared to the final reconstruction. (ii) Data-to-Model based analysis: the FSC curve between the final IPET reconstruction and the density map converted from the corresponding fitting model was calculated. The frequencies at which the FSC curve fell below 0.5 was used to estimate the resolution. The density map of the fitting model was generated by pdb2mrc in EMAN1 software[65].

## Modeling the structure of nucleosome arrays

To build a structural model for the reconstructed map of each individual nucleosome array particle, the pathways for the two flanking DNA arms were initially traced by sampling a group of 3D points located at the high-density loci of the map followed by sorting their order into a points list. Then, we fit the crystal structure of Xenopus laevis NCP into the discoidal-shaped high-density region. In this step, our fitting was based on two assumptions: (1) nucleosomes are positioned on 601 regions and (2) unwrapping of one side of a nucleosome corresponds to an increase in the strength of wrapping on the other side[37,68] (i.e. only one side of NCP unwraps at a time). Therefore, in the case of an unwrapped NCP density map, it is only necessary to take into account two potential fitting solutions (Supplementary Fig. 8e). Given this framework, the identification of any DNA segment corresponding to a 40-bp, 100-bp, or 200-bp length can help to determine the fully wrapped side of an NCP, which then serve as a reference point for fitting the remaining NCPs (see evaluation in below section).

In the meantime, the DNA linker length serves as a mean to differentiate between crossovers (octasome) and detachment of DNA ends (hexasome) when θ angle approaches 180°. An evident contrast between the two scenarios is that ~80 bp of DNA unwrapping in the hexasome leads to an additional 27 nm of DNA extending from either distal end of the NCP arms. In certain instances, the loading of extra nucleosome core particles (NCPs) onto the distal arms can complicate the differentiation between the 100 bp and 200 bp arms. Yet, it's only the 200 bp arms that offer sufficient DNA length to accommodate the off-positioned 147 bp NCP. In this scenario, we noted the distal arm satisfying the condition of x bp + 147 bp + y bp, where x + y equals 53 bp.

 

Next, all fitted DNA trajectory (including entry-exit arms, intermediate linkers, and NCPs wraps) were converted to a list of 3D points. These points series from different models were merged together after removing adjacent clashes and then fitted with a smooth quadratic Bezier curve followed by conversion into DNA model by using GraphiteLifeExplorer[69]. The total length of the DNA used to thread the model matched with our designed DNA construct, *i.e.* 456, 632, 818, 1008 for mono-, di-, tri- and tetranucleosome, respectively. Notice that for some of the maps, a small portion of DNA densities (~10%) was missing near the middle portion of DNA arms (which may be due the orientation of DNA segments aligned near-perfectly perpendicular to the beam direction, resulting the lowest image contrast of the DNA). Fortunately, those missing portions were small and did not prevent the DNA model fitting, which can be circumvented by interpolating surrounding density at those loci. To further refine the model, the Molecular Dynamics Flexible Fitting (MDFF, ver 4.0)[34] was applied to energy minimize the model under a force gradient created by the electron density map.

From our cryo-ET individual particle investigation of nucleosome arrays conformation, we encountered increasing difficulties as the nucleosomes began to stack together, particularly when the linkers, NCPs and distal arms came into close proximity. This challenge is evident in the reduced quality of tetranucleosomes under higher ionic strength conditions, such as 50 mM Na⁺ comparing to mononucleosomes at 5 mM Na⁺. Histone protein H1, a well-known factor, exhibits a synergistic effect with high ionic strength, promoting the stacking conformation[7] and accelerating nucleosome phase separation[19]. when NCPs are densely stacked together, it becomes a limitation of our method, making it challenging to distinguish the orientation of the internal NCPs and linkers. Further work aimed at enhancing the resolution is required to solve these denser structures.

## Evaluating the fitting models

To evaluate the self-consistency of the above fitted models, the correlation between the measured DNA linker length between two consecutive NCPs (named as $L(n)$) and the estimated DNA unwrapping level between the same NCPs were calculated (Supplementary Fig. 8f). The $L(n)$ was acquired from the experimental data, in which distance between the DNA exit position of the nth NCP and the entry position of the $(n+1)^{th}$ NCP from the density map were measured $(L(n) = d(n) + d(n+1))$. The DNA unwrapping level was estimated from the fitted model, in which the unwrapping angles from the exit side of nth NCP and entry side of the $(n+1)^{th}$ NCP were added together $(\varphi(n) = \varphi_{ex}(n) + \varphi_{en}(n+1))$. The Pearson correlation coefficient, $r = -0.9$ was calculated from the linear regression fitting of the unwrapping length $L(n)$ against the angle $\varphi(n)$ using the statsmodels package in Python (Supplementary Fig. 8g). The dynamic nature of histone assembly and disassembly on DNA templates can lead to the NCP localizing at positions other than the canonical 601 sites. This phenomenon occurs more frequently when there is a free 200 bp DNA arm, a higher histone: DNA loading ratio (e.g., in tetranucleosomes), and under low salt conditions that increase non-specific binding. The off-template NCPs (binding to the 100 or 200-bp distal arms region) are removed from the following statistical analysis.

## Defining the entry and exit linker DNA origins on the NCP

By using the fitted nucleosome array models, the entry and exit DNA arm origins on the NCP can be estimated with the following procedure. The DNA portion of the fitted nucleosome array model was converted into a list of 3D points (series m) by averaging the coordinates of C1 atoms of each base-pair. After aligning the nucleosome crystal structure to the fitted array model at its nth NCP region, a points list (series n) along the wrapped DNA from the nucleosome crystal structure was also generated by the same method. By comparing the two points lists, the overlapped region on the fitted DNA model can be identified if any

points from the series n were found within its 8 Å radius of series m. The 8 Å criterion was chosen based on that, when two DNA helical centers separated away from each other for over one third of the DNA diameter (~24 Å), the separation of two aligned DNAs can be distinguished. The base pair indexes at the two distal ends of this overlapped region were used to define the entry and exit DNA linker arm origin along the array. The total number of the base-pairs between the entry and exit position were used to calculate the length of wrapped DNA on the histone surface.

## Measuring the NCP wrapping dynamics on histone surface

By measuring the position and length of the wrapped DNA on each histone surface, the DNA unwrapping footprint along the array can be quantified as follows. For a fitted array model, a binary score was assigned to each DNA base pair along the DNA sequence depending on its contact with the histone octamer ("1" for contact if the atoms of the base pair fall in a 4 Å distance with any histone octamer atoms and "0" for non-contact). By averaging the scores of the $i^{th}$ base pair within same type of fitted array models, the mean score distributions along the DNA template (from upstream to downstream) for a mono-, di-, tri-, and tetranucleosome were calculated separately. The score distributions around the regions contains the designed 146-bp Widom 601 position sequence[70] reflects the probability of finding DNA unwrapping events. The difference of the score distribution for the entry and exit side of each NCP loading position reflects the asymmetry property of the DNA sequence in binding to histone octamers and the dynamic relationship between DNA unwrapping and equilibrium assembling, such as transition states among the tetrasome, hexsome and full octasome[71].

## Defining the entry and exit linker DNA vectors on the NCP

Due to the fact that the 20-bp DNA segment (~6.8 nm) is relatively stiff compared to the persistence length of DNA ~ 50 nm, the base pairs residing at the entry side DNA arm origin and its 20-bp upstream were used to define the start and end-point of a vector, respectively (based on their helical center coordinates). This vector was used to represent the entry DNA arm pointing direction. Similarity, the base pairs residing at the exit side DNA arm origin and its 20-bp downstream were used to define the exit DNA arm vector.

## Measuring the angle θ, wrapping angle α and bending angle β of the DNA arm vectors

The angles of the DNA linker arms that extended from the discoidal-shaped NCP surface were measured by the following steps. (i) Defining the X-, Y-, and Z-axes of each NCP model. The Z-axis was defined along the helical axis of the wrapping DNA measured from its rotational symmetry. The Y-axis was defined by the dyad axis of the NCP. The cross-point of these two axes was used as the center of NCP and defined the X-axis, where it simultaneously passed the NCP center and was perpendicular to both Z- and Y-axis. (ii) Defining the relative DNA arm angle θ. θ measured the angle between the previously defined entry and exit DNA linker arm vectors on the same NCP. Because DNA arm conformational states ("open and close") must be defined relative to the NCP, the projections of the θ angle on the NCP discoidal plane (X-Y plane) and its perpendicular plane (Y-Z plane) were measured as $\theta_F$ and $\theta_S$, respectively. (iii) Defining the in-plane wrapping angle α of an NCP arm vector. α calculated the angle formed by two vectors within the X-Y plane. One vector is the projection of the NCP linker arm vector on the X-Y plane, and the other vector is defined by the tangential direction of the discoidal-shaped projection of NCP on X-Y plane, which crosses the origin of the corresponding NCP linker arm. (iv) Defining the out-of-plane bending angle β of an NCP arm vector. β calculated the angle formed by the NCP arm vector and the X-Y plane. The measured angle distribution was fitted with either one gaussian or two gaussians with the sklearn.mixture.Gaussian Mixture package.

## Measuring the intra-array NCP core-core distances and plane-plane angles

To quantitatively define the spatial relationship among the NCPs from different types of nucleosome arrays, the core-core distances and plane-plane angles between each pair of NCPs were measured. The core-core distance, $D(n, n+m)$, measured the distance from the center of the nth to the $(n+m)^{th}$ NCPs. The core-core angle, $\varphi(n, n+m)$, measured the dihedral angle between the discoidal planes (X-Y planes) of the nth and $(n+m)^{th}$ NCPs. The histograms were fitted by a Kernel Density Estimation (KDE) function[72]. The dihedral angle distributions were compared with a sine function, which represents a distribution of orientations of two random planes (the angle $\theta$ between their normal directions), which in turn are equivalent to the probability of finding a point on the unit hemisphere contained in a differential ring-shape area:

$$P(\theta)d\theta = \frac{2\pi[R\sin(\theta)]Rd\theta}{2\pi R^2} = \sin(\theta)d\theta \qquad (1)$$

## Insilco-construction of the chromatin-like higher order structure

The distribution of the wrapping and bending angles relative to the NCP were used to build longer in silico nucleosome array fibers. Each longer array fiber containing 100 NCPs was generated by randomly connecting nucleosome model units. To be specific, due to the large number of atoms within the NCP (>12k) and the fact that the length of 40-bp DNA (-14 nm) is much smaller than the persistence length of DNA (-50 nm), NCP units were coarse grained where the histone octamer and linker DNA arms were treated as spheres and straight lines, respectively. Based on the distribution of the measured mean and standard deviation of the bending and wrapping angles (i.e. $\alpha$ and $\beta$) for the tetranucleosome array at low salt and high salt condition, two vectors in a 20-bp length followed the same angle distributions relative to the NCP were randomly generated and used to represent the entry and exit DNA linker arms by using VMD software[73]. By repeating this process, a pool of NCP units with various linker arm conformations but following the same spatial distribution to the experimental data were prepared. To assemble the NCPs units into a fiber, randomly selected NCPs from the pool were sequentially connected together based on the following procedures: (i) The exit side 20-bp linker for the ith NCP was linearly connected to the entry side 20-bp linker for the i+1th NCP, (ii) The 40-bp linker causing -70° left-handed DNA rotation has been considered[74], in which the $(i+1)^{th}$ NCP unit was rotated along the entry side linker vector after connecting to its previous NCP. (iii) if the $(i+1)^{th}$ NCP was identified to clash with any of the previous NCPs, an alternative NCP will be randomly selected from the pool and resembled onto the ith NCP by repeating step i through iii. In order to simulate more realistic dynamics of the nucleosome fibers, the unwrapping events were also considered into the system. The origin of the entry and exit DNA linker vectors on the helical NCP track were also varied based on the experimentally measured distribution when constructing the pool of the NCP units.

## Insilco-construction of the mini-chromosome

To simulate the tertiary organization of the chromatin fibers, a pool of 1000 of the above constructed hectanucleosome fibers was prepared by using low- and high-salt angle and unwrapping parameters. Considering that a compact chromosome typically consists of around 50 million base pairs of DNA, it has the theoretical capacity to accommodate -250,000 NCPs (assuming that each 200 base pairs occupy a single nucleosome). To facilitate the extension of chromatin fibers, we implemented a repetitive random sampling approach. The initial hectanucleosome fibers were interconnected with the second set by

aligning their distal NCP linker vectors without structural clashing. This threading procedure was iteratively executed for a total of 2500 iterations. The iterative process allowed for the gradual expansion of the chromatin fibers without imposing any spatial constraints. The final representation involved cropping out the central cubic volume measuring 200 nm in length from the region with the highest density in both the low-salt and high-salt models.

## Classification of the nucleosome array conformations

To identify some of the low energy conformational state of the nucleosome array, which should be observed more often than others based on Boltzmann distribution, nucleosome arrays were classified based on their structural similarity. For each type of the array, 30 to 40 conformations were obtained and subjected to a pair-wise alignment through minimization of the RMSD between each pair of models. Based on the values of the constructed distance matrix evaluated by RMSD, the array conformations were sorted and classified with an optimization algorithm that minimized the tree spanning (method option = 'single') using the hierarchical clustering in Scikit-Learn of Python package[75]. The final result was displayed in a dendrogram produced by the scipy.cluster.hierarchy package.

## Visualization of the nucleosome array structural diversity

To facilitate the visualization of the conformational variety of nucleosome array structures via motion rather than a static snapshot, we morphed the conformations of nucleosome array structures from one to another. This was done by following a hierarchical clustering order, calculated based on the pairwise RMSD values among all structures. The morphing animations began with the structures located at the lowest branch of the dendrogram (representing consensus conformations) and terminated at the highest branch, corresponding to rare conformations. Because the chimera morphing function introduces artificial DNA kinks for nucleosome arrays, the morphing was carried out using Targeted Molecular Dynamics (TMD) simulations. The TMD was performed in NAMD2 (ver. 2.14) engine[34], where the SIRAH coarse-grained structure[76] was steered from one conformation toward to another with the SIRAH forcefield as implemented in AMBER[76] within implicit solvent (5 mM ion concentration). The System was maintained at 298 K temperature using the Langevin dynamics with a damping coefficient of 50/ps. The elastic constants for TMD forces used were in a range of 200-600 kcal/mol/Å scaled based on the distances of the corresponding CG model atom pairs. After 50,000 steps of energy minimization (corresponding to a total time of 1 ns, with each step of 20 fs) and 1,000,000 to 4,000,000 steps of steering (corresponding to a total time of 20 ns to 80 ns) within the implicit solvent, the simulation was terminated when the real-time RMSD fell below 3 Å. Among these morphing structure pairs, -10% of them were eliminated due to the DNA intertwining with each other. The morphing movie was also displayed with the same order as above from the most populated structure states to the rarest states. We noted that the trajectories stemming from TMD are nonequilibrium trajectories that do not reflect the true unbiased dynamics of the system.

## Reporting summary

Further information on research design is available in the Nature Portfolio Reporting Summary linked to this article.

# Data availability

The 3D density maps of a total of 223 cryo-ET IPET reconstructed particles of nucleosome arrays produced in this study have been deposited in the EMDB under the following accession codes: EMD-28284, to EMD-28286 EMD-28331 (for mono-), EMD-28334 to EMD-28366 (for di-), EMD-28367 to EMD-28372, EMD-28527, EMD-28528, EMD-28379 to EMD-28404, and EMD-28406 to EMD-28416

(for tri-), EMD-28417 to EMD-28447 (for tetra-), EMD-28448 to EMD-28450, EMD-28452 to EMD-28464, EMD-28524 to EMD-28526, and EMD-28468 to EMD-28482 (for tetra- 50 mM Na$^+$), and EMD-28483 to EMD-28486, EMD-28488 to EMD-28493, EMD-28495 to EMD-28497, EMD-28499 to EMD-28513, and EMD-28516 to EMD-28507 (for tetra-H1). The associated cryo-ET raw tilt series, final maps, fitting models, EMDB reports, and figure-related cryo-EM raw micrographs have been uploaded to a Figshare repository at: https://doi.org/10.6084/m9.figshare.25658517. Supplementary Figs. 14–238 include seven representatives of raw tilted images, projections of the intermediates of 3D reconstructions, final reconstructions and their fitting models showed at the corresponding tilted angles. The measured angles on each NCP within these particles are detailed in Supplementary Data 1 and all measured data used for producing the curves in figures are available in the Source Data file. Statistical results are provided in Supplementary Data 2. Source data are provided with this paper.

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

## Acknowledgements

We thank Dr. Karen Bustillo at The National Center for Electron Microscopy (NCEM), Molecular Foundry, Lawrence Berkeley National Lab (LBNL) for supporting the NS-EM imaging; Dr. Dan Toso at Cal-Cryo-EM center of QB3-Berkeley for supporting the cryo-EM/cryo-ET imaging; and Ms. Amy Ren for editing. The work at the molecular foundry, LBNL, was supported by the Office of Science, Office of Basic Energy Sciences of the United States Department of Energy (contract no. DE-AC02-05CH11231). US National Institutes of Health grants R01HL115153, R01GM104427, R01MH077303, R01DK042667 (GR, JL, MZ) and R01GM032543 (C.B). The AFM measurement was supported by the U.S. Department of Energy (DOE), Office of Science, Basic Energy Sciences (BES), under Award #DE-SC0022305 (JT). PNNL is operated by Battelle for the Department of Energy under contract No. DE-AC05-76RLO1830.

## Author contributions

C.D., M.Z., G.R. and C.B. designed research; C.D. prepared mono-nucleosomes and nucleosome arrays; M.Z. carried out all cryo-EM experiments, including data acquisition, image processing and built the cryo-ET models; JT and PA for the AFM imaging and analysis; M.Z. and G. R. analyzed the data and interpreted the data. M.Z. prepared the figures and movies; M.Z. and G.R. wrote the paper and revised by all authors.

## Competing interests

The authors declare no competing interests.
