## [Peer Review File · Nature Communications]

Reviewers' Comments:

Reviewer #1:

Remarks to the Author:

The article by Zhang et al., characterises the single-molecule behaviour of mono, di, tri, and tetranucleosomes at low salt concentrations with sub-nucleosome resolution using an interdisciplinary approach that combines state-of-the-art cryoET with different computational modelling and simulation approaches.

While I find the results highly interesting and the cryo-ET data particularly beautiful, I was not always convinced that the claims presented are fully backed up by the data and analysis. This is perhaps related to the manuscript being, in my opinion, very hard to follow. I have the following concerns about the manuscript in its current form.

1. On line 331, the authors say “~40% of the particles displayed a conformation in which the 200-bp DNA arm invades one of the NCPs by replacing part of its originally wrapped DNA at the exit side”. I was very excited by this result but, on closer inspection, I could not fully follow how the authors determined that a DNA arm was actually invading a foreign nucleosome, rather than just being located nearby given the resolution of the cryoET experiments. While at low salt, the electrostatic repulsion among DNA arms is large, also is the electrostatic attraction between lysines in the histone tails and the DNA phosphates. Can the authors distinguish between DNA--foreign histone tail interactions and DNA invasion of a foreign exposed nucleosome core? Similarly, on line 371, the authors state that “The second peak appearing at 313A likely reflects co-existing distal arm invading conformations”. On line 390, the authors mention: “appearance of conformations in which the upstream DNA arm of the first nucleosome invades and wraps around the second nucleosome (Fig.4e, panel II) may reflect the fact that, at low ionic strength, NCPs are kept separated at “preferred” distances leading in some cases to partial unwrapping of the DNA, which in turns permits the wrapping of “foreign DNA””. Again, I do not see how these conclusions were obtained. The result that DNA invades a foreign nucleosome does make sense, in light of the plasticity of nucleosomes and heterogeneity of chromatin structure in vivo and in some in vitro experiments, and might represent an additional parameter that chromatin can exploit to modulate its structure. Thus, it would be excellent if the authors could explain clearly how this important conclusion was drawn from the data.
2. On line 385, the authors mention that “the presence of H1 locks the two DNA linkers/arms of an NCP, yielding a similar outcome as high salt concentrations by drawing them closer together”. I do not understand what the authors mean by “locks the two DNA linkers/arms” and I also do not see how this was concluded from the data.
3. The discussion of all the different angles analysed (e.g. alpha, beta, theta, theta_I, theta_II) is very hard to follow. I think it would be much easier for the readers, especially due to the interdisciplinary appeal of the work, if the authors made the effort to digest the information a bit more, e.g. describing their geometrical findings in terms of how the nucleosome entry/exit angles open or close up, or how the inter-nucleosome distances and orientations change for the different systems studied, instead of repeatedly detailing the changes in alpha, beta, theta, etc.
4. The authors perform Targeted MD coarse-grained simulations of mononucleosomes with the SIRAH force field. Then, the authors state that these simulations “displaying the thermally-induced dynamics of the particles in solution” and “These results demonstrate the capability of the use of individual particle cryo-ET methods to study nucleosome array dynamics in the following studies.” However, targeted MD is a method that biases a conventional MD simulation to enforce a transition between two equilibrium states (e.g. state I and II) by using moving harmonic restraints. Hence, the trajectories stemming from targeted MD are nonequilibrium trajectories that do not reflect the true unbiased dynamics of the system. In other words, the dynamic information that is extracted from TMD is meaningless. Regarding these simulations, very limited information is given about how the simulations were performed, and in their present form cannot be reproduced. Why the SIRAH model was chosen also needs to be discussed. If an unbiased MD approach had been used, the dynamics reported would only be as good as the underlying model used; thus, explaining how the SIRAH model has been validated for nucleosome arrays is important.

5. The authors develop space-filling models (spheres and tubes) of 100-nucleosome arrays using the geometric descriptors obtained from their cryoET experiments. Then they use two approaches, which they describe as “seamless assembling” and “random docking” to build high-density structures with such 100-nucleosome arrays. It is interesting to try to extrapolate the single-molecule behaviour of 100-nucleosome arrays to the organisation that they might have within a biomolecular condensate. However, phase separation is an emergent property of chromatin, and there is no reason to expect that the single-molecule structural properties of chromatin arrays will remain unchanged within condensates. On the contrary, the expectation should be that chromatin structure will change when it goes from the “gas phase” to the “liquid phase”. Nucleosomes within a single chromatin array can only increase the enthalpy of the system by establishing interactions with other nucleosomes in the same array, which favours a given type of chromatin arrangement. In contrast, nucleosomes within a chromatin condensate contribute to increasing the enthalpy of the system by forming interactions with multiple arrays—indeed multivalency is critical to yield the percolated network of the condensed liquid—and hence, phase separation likely favours a different type of chromatin arrangement. Therefore, while this space-filling task allows one to imagine how 100-nucleosome arrays might look inside a condensate, I do not think this exercise is sufficient for drawing conclusions about the structure of chromatin inside condensates. For instance, line 424 states “the 30-nm fibers generated in this simulations were not often observed” in the experiments “one possible explanation for this discrepancy is the low probability of capturing fibers lying exactly on the plane of the thin slice cell section”, when the in-silico structures might be completely artificial. On line 435, the authors say that “we repeated the above simulations by explicitly preventing DNA unwrapping” “but did not induce an obvious change of the condensate’s density in either salt conditions” “suggesting that breathing does not increase enzyme accessibility to the interior of the condensates”—which again is a conclusion that cannot be unequivocally derived from their data.

Minor corrections:

Some of the references point to the wrong papers.

Reviewer #2:

Remarks to the Author:

In this MS, Zhang et al use primarily cryo-ET and also MD simulations to study the structure of mono-, di-, tri-, and tetranucleosomes in solution. By analysis of spatially well-separated reconstituted nucleosomes and oligonucleosomes, the MS maps the angular disposition of the DNA arms. The samples studied include: mono-, di-, tri-, tetranucleosomes in 5 mM NaCl; tetranucleosomes in 50 mM NaCl; tetranucleosomes with histone H1 in 5 mM NaCl. It was unclear which samples were considered controls, making the independent variable(s) difficult to identify. Measurements were made along axes parallel to the discoidal plane (theta parallel / alpha) and perpendicular (theta perpendicular / beta). Overall, the DNA arms sample a large range of angles. This angular sampling changes (but remains broad) in 50 mM NaCl resulting in more compact tetranucleosomes.

Overall, I struggled to follow the narrative of this MS because the samples (see above) do not have adequate controls and because the presentation (text and figures) was confusing throughout.

My comments are mostly about the cryo-ET and will hopefully be useful.

DETAILED COMMENTS

The motivation for the study is not very convincing. How does learning about the structure of different nucleosome arrays in an environment that does not promote LLPS (i.e. low ionic strength) inform about chromatin condensation via LLPS?

The previous paper showed that tetranucleosomes form condensates. Do mono-, di- and trinucleosomes also form condensates? I understand the need to present the mononucleosome data as a way to show that the analysis workflow works, but I am not sure what’s the point of showing dinucleosomes and trinucleosomes – especially if it is unknown whether they form

condensates.

Some tetranucleosomes look like they have 5 nucleosome core particles (e.g. particle #132, #155, #156, #158). These were not addressed.

Some particles had unresolved DNA arms (e.g. particle #132, #136, #137, #138). In these cases, how do you assign which is the entry or exit arm?

One way the model fitting was evaluated was to measure the correlation between the measured DNA linker DNA between two particles from the density and the unwrapping angle from the fitted model (under methods: see lines 698 – 710). While this shows that the visualized model does fit into the density, it does not show that this is the only model that can fit into this density. This is particularly important for more ambiguous densities such as the condensed tetranucleosome array in 50 mM Na⁺.

How thick are the cryo-EM samples?

Are the DNA arms or nucleosomes interacting with the air:water interface?

The title should be revised to more accurately say what the data represent. As written, it implies that the structures visualized by cryo-ET (the dominant portion of the MS) have captured conformational changes during a state change ("prior" to phase separation). A more accurate title would be: "3-D structural diversity of mono, di-, tri-, and tetranucleosomes in low-salt conditions". While this variant title is rather dry, it is far more accurate than the current one.

How are mono, di-, tri-, and tetranucleosomes picked from the tilt series to be used for analysis? Was it random or was there a criteria (if so, what criteria)? Were they picked by hand or by some software?

The way the experiments were done requires more explanation:

Performing Volta phase plate imaging with a high overfocus is not commonly done (this MS used 2.5 μm; typically -0.5 μm defocus is used), and the resulting tilt series have a much lower contrast than expected (Figure 1D). What is the rationale behind using a high overfocus? If there was a typo here and the actual defocus was -2.5 μm, why was such a large underfocus used?

The amplitude and phase of the CTF was corrected using multiple CTF correction programs. What is the rationale behind using multiple programs? Please add more detail on how the CTF correction was done.

The reported parameters sometimes do not make sense e.g. reporting the average of a skewed distribution (c.a. Fig. 2e). Shouldn't the mode of the values be presented instead?

The manuscript tries too hard to draw parallels with chromatin function(transcription) and cellular phenotypes (condensation in vivo / in situ). For example:

Lines 85-88

Implies that the chromatin in this study is transcriptionally active. Is there any experimental evidence to justify this description? The same appears in lines 91-92.

This statement also implies that somehow the structural changes going from 5mM Na⁺ to 50 mM Na⁺ phenocopies changes that affect "gene expression". The cell's interior does not, to my knowledge, have a 10-fold variation in sodium concentration, nor do cells change their ionic strength so much through the cell cycle. Also, there are numerous other intracellular factors that affect chromatin condensation that are not tested in this MS.

Lines 95-98

The mention of "external stimuli" is too vague. Please give more explicit examples of how changes in oligonucleosome structure induced by increasing the Na⁺ concentration or adding histone H1 would inform on a cell's response to external stimuli.

Line 95

The Na⁺ concentrations (up to 50 mM) are not physiological, which the text acknowledges in line 286 when referring to the predecessor study. This deviation from physiological-ness is at odds with the implication that this study somehow mimics transcriptionally active chromatin.

Line 125-126

Why would the 2D imaging results become “the basis for the following 3D reconstructions”?

Line 158

The section heading implies that the data will reveal “dynamics”. Indeed, in the way the rest of the manuscript is written, the implication is that the cryo-ET data show dynamics. It would be more accurate to call this section “3D structural variability of mononucleosomes”. For cryo-ET data to show dynamics, a reaction-initialization device and plunge-freezing like the one from Joachim Frank’s lab would be needed.

Lines 189-192 and 251-252

Could ice thickness contribute to restricted beta angles?

Lines 235-244

The discussion about the role of electrostatics on DNA arms in the context of oligonucleosomes is confusing. The impact of the DNA arm(s) connecting the dinucleosomes (or nucleosome N with N-1 and N with N+1) is ignored. Consider doing a simulation to back up these claims.

Lines 295-298

Why are tetranucleosomes in 50 mM Na⁺ considered a condition prior to phase transition? Does phase transition occur when one gradually increases the salt concentration in the same sample? I thought a better condition would be tetranucleosomes in 150 mM Na⁺ at the 0 min time point.

Lines 333-336

This experiment tells us (a) whether H1 can condense arrays at 5 mM Na⁺ (b) the characteristics of tetranucleosome arrays in the presence of H1 at 5 mM Na⁺. But how does incubating tetranucleosomes with H1 in 5 mM Na⁺ help investigate how H1 accelerates the spinodal-to-spherical conversion at 150 mM Na⁺?

Lines 339-340

Looking at the distribution in Extended Data Figure 4, and the standard deviation values, I’m not sure the values of -17 ± 19 -bp versus -14 ± 12 -bp represent any meaningful difference. These values are said to represent a “slightly increased unwrapping level and dynamics”, which is confusing because the unwrapping values are negative, which should be interpreted as rewinding.

Lines 340-343

The text here says that -45 bp constitutes unwrapping, but I had thought that only positive values would constitute unwrapping while negative values constitute rewinding.

Lines 585-586

4°C is not physiological for most of the organisms in which chromatin is studied. If the goal is to relate this study’s experiments to transcription or to cells, the experiments should be repeated at 30°C or 37°C. This is also not a temperature that in vitro transcription assays would be done at. Alternatively, as I suggested above, remove all the implications about function/cells.

Line 109

Please explain the choice of a 447-bp-long?

Line 175

The theta angles approach 180°. At these extremes, the two DNA arms could either be crossed or alternatively, the “core” DNA could be half detached, resulting in only one gyre of DNA. To distinguish the two possibilities, the DNA gyres would have to be resolved (~2.5 nm resolution). However, the data was low-pass filtered to 4.5 nm (line 661). Also, the resolution Please explain this discrepancy.

Line 187 and Fig. 2j

I see a right skew, not a left one.

Line 194

It's more accurate to say either "speculate about the dynamics" or "simulate the dynamics" instead of "display the dynamics".

Line 196-197

It was very hard to find "supplementary figure 3" in the 250-page supplementary figure file. I suggest reordering or giving readers a hint that it's on page 53.

Line 285

The "previous study" should be ref 28.

Line 398

"a longer linker" – Isn't the linker length constant in the constructs used?

Lines 424-430

The cryolamella referenced is ~160 nm thick and the cryosections referenced are ~70 nm thick, so the argument that too thick of a section (30 nm) would hide 30-nm fibers is incorrect. Also, note that Scheffer's chicken erythrocyte paper was done with 50-nm thick cryosections and the 30-nm fibers are easily visible.

Lines 447-451

The statistical mechanical arguments here need a caveat that the possible interaction between the chromatin complexes and the air:water interface (see above) may bias the distribution from what would be found in bulk solution.

CONFUSING WORDING/PHRASING:

"mono-, di-, tri-, and tetranucleosome arrays"

Mononucleosomes are not arrays

"Intramolecular" and "intermolecular" seem to be used interchangeably throughout.

Line 50

"reorganization of nucleosome arrays" – This line refers to the state of chromatin inside of cells. I'm afraid it's confusing to use "arrays" to describe cellular chromatin.

"in response to external stimuli"

This term is appropriate for cell and animal studies. I think "changes in buffer conditions" is more correct here.

Because the nucleosome has pseudo-twofold symmetry, the terminology used to describe entry and exit DNA needs to be clearer. The mononucleosome used in this MS is asymmetric because one DNA arm is much longer than the other. Instead of calling it the entry and exit DNA arms, it would be a lot clearer to say long and short DNA arms so readers don't have to remember that entry = 200 bp and exit = 100 bp.

Line 78-79 "Histone H1 catalyzes more than 10-fold the spinodal-to-spherical conversion"

Similarly, in lines 332-333: "Considering the Histone H1 catalyzes the spinodal-to-spherical conversion for more than 10-time faster than that absents H1" → "Considering that histone H1 catalyzes the spinodal-spherical conversion rate more than 10-times"

Line 84

"large portion of chromatins" > "large portions of the chromatin"

Line 129-130 "Reference-free 3D reconstruction"

This is a bit excessive. I am unaware of people in the tomography field doing non-reference-free 3D reconstruction.

Line 132 "missing-wedge correction"

I suggest "missing wedge compensation" instead because "correct" means there are no errors. While tomography experts know better, the broader audience will think that we've eliminated all matters with the missing wedge, which is not true.

Lines 115-118

The structural features visible are described in a sentence that includes both cryo-EM and NS-EM. Most notably, this sentence mentions "major groove" in the same phrase as cryo-EM. While the figure callout that follows (Extended Data) is for the NS-EM experiment, a lot of readers are going to think that major grooves were visible in the cryo-EM data. It's better to describe what's seen in cryo-EM vs. what's seen in NS-EM in separate sentences.

Lines 124-125

This sentence conflates what's visible in cryo-EM vs NS-EM. It hurts the credibility of the cryo-EM/ET data (the bulk of the experiments in this MS), which don't have any DNA groove features.

Lines 200-202 "These results demonstrate the capability of the use of individual particle cryo-ET methods to study nucleosome array dynamics in the following studies."

It's more accurate to say that cryo-ET initializes a molecular dynamics study. To truly study dynamics by imaging, one would have to image the same unique particle as a function of time, like in liquid-cell EM or timelapse light microscopy.

Line 257

What does it mean for DNA arms to be "lifting off from the NCP discoidal plane"?

Line 291 "Tetranucleosome has been used as a phenotype with its minimal structural unit and its capability of forming higher-order chromatin structures..." I don't understand this sentence.

Line 317

What kind of "electrostatic effect"?

Lines 332-333

"Considering the Histone H1 catalyzes the spinodal-to-spherical conversion for more than 10-time faster than that absents H1" > "Considering that histone H1 catalyzes the spinodal-spherical conversion rate more than 10-times"

Line 513

The title should read "reconstitution and purification"

Line 578

Add in the actual citation for "Mol Cell paper reference".

Lines 590-591 "optimized negative-staining protocol (OpNS)"

To my knowledge, everyone optimizes their negative staining experiments (and all other experiments) prior to publication, but they don't call it optimized negative-staining.

Line 601

Presumably, the intended text was "2.5 μm underfocus". If this is the case (or if they used overfocus), please explain why such a large defocus was used with a Volta phase plate. The inventors of the Volta phase plate suggest imaging closer to focus (though not at 0 μm defocus). See my related comment above. This issue comes up again at line 608-609.

Lines 630-631

"Phase and amplitude were corrected by SPIDER, GCTF, and TOMOCTF". Is that necessary? And how was that performed? All 3 software packages correcting at once or sequentially? Is this part of the NOISE2NOISE? See my related comment above

Lines 648-649

"which can eliminate the effects from the large image distortion". Local tilt series image alignment cannot eliminate any distortion effects, it merely reduces the impact of large image distortions (?). Does "large image distortion" mean distortion of a large image (like the original uncropped tilt series) or an image distortion that's large?

FIGURES

Figure 1a

What Na⁺ concentration was used in this experiment? What kind of nucleosome arrays were used in this experiment?

Figure 1c

The density features indicated in orange are not turns of DNA (~90 bp). I think "gyres" is more often used in the field.

Figure 1d

I am not sure what the advantage of showing tilt series images is. The particles can barely be seen in these projection images.

Figure 1 f/g

Do the cyan and grey isosurfaces correspond to the different contour levels?

Figure 1f/h

Images are cut off.

Figure 1g-h

Please give a more detailed description of the difference between theta and theta-parallel? And why are the values for the beta-angles negative?

Figure 1h

alpha en and alpha ex are introduced, but the rest of the text and plots just have alpha without subscripts. Please clarify.

Figure 2d and lines 166-168

It's stated that on average, mononucleosomes are unwrapped by 5 bp and 11 bp on the entry and exit side respectively. But in this figure, the values "-5" and "-11" appear instead, alongside grey dotted lines that appear to mark the 0 bp unwrapping. Is the "average unwrapping" calculated based only on the positive unwrapping values? Because the meaning of the sign (positive/negative) is not well defined, it's hard to know if the distributions imply that most nucleosomes are partially unwrapped or overwrapped.

Figure 2e

This is not a normal distribution, so please consider reporting the mode, not the average value.

Figure 3c

For multimodal distributions, please show the individual Gaussian curves' fits that allow the extraction of the different peak values. While it is obvious that there are two peaks for the distribution of dinucleosome alpha-angles, it is not for the trinucleosome and tetranucleosome alpha-angle distributions. Could there be more than 2 peaks for those? This issue also appears multiple times in graphs of other figures e.g. Extended Data Figure 7b.

Extended Data Figure 1a

What fluorescence is being detected?

Extended data 1b, lower panel

The condensate is referred to as "Spherical". Being that this is a negatively stained sample, the material has most likely flattened against the carbon support and indeed in the results, they are

described as “relatively flattened”. Please use a more accurate description of the shape of this condensate.

Extended Data Figure 2

Is this data showing 3D reconstructions of cryo-ET or NS-ET?

If it is cryo-ET, panel A should show a representative tilt series of cryo-ET, not NS-ET.

Minor note: Or should readers understand that NS-ET data was acquired while the sample holder is operated in cryogenic conditions? Not mentioned in methods.

Extended Data Figure 5a

There appears to be two peaks for the theta-angle distribution of tetranucleosomes with H1, but only one peak value is reported. This issue also appears multiple times in graphs of other figures e.g. Extended Data Figure 7b.

Extended Data Figure 9

“causing intra-array NCP units’ differentiation” – I do not understand what this means.

I cannot distinguish individual nucleosomes. There needs to be a 3-D isosurface rendering like Fig. 1f to show that the theta angles can be reasonably measured.

For the distribution plots (like Fig 3c), it would be clearer to use a schematic to show the range of angles as two lines with an double-headed arced arrow like Fig. 5b. Also, it would be helpful to have such a schematic for the other distributions like Fig. 3d.

For the tilt series images throughout the MS, why do some show -45, 0 , 45 and others -30, 0, 30 degrees?

Lu Gan

Reviewers' comments:

Reviewer #1 (Remarks to the Author):

The article by Zhang et al., characterises the single-molecule behaviour of mono, di, tri, and tetranucleosomes at low salt concentrations with sub-nucleosome resolution using an interdisciplinary approach that combines state-of-the-art cryoET with different computational modelling and simulation approaches.

While I find the results highly interesting and the cryo-ET data particularly beautiful, I was not always convinced that the claims presented are fully backed up by the data and analysis. This is perhaps related to the manuscript being, in my opinion, very hard to follow. I have the following concerns about the manuscript in its current form.

Comment 1.1: On line 331, the authors say “~40% of the particles displayed a conformation in which the 200-bp DNA arm invades one of the NCPs by replacing part of its originally wrapped DNA at the exit side”. I was very excited by this result but, on closer inspection, I could not fully follow how the authors determined that a DNA arm was actually invading a foreign nucleosome, rather than just being located nearby given the resolution of the cryoET experiments. While at low salt, the electrostatic repulsion among DNA arms is large, also is the electrostatic attraction between lysines in the histone tails and the DNA phosphates. Can the authors distinguish between DNA–foreign histone tail interactions and DNA invasion of a foreign exposed nucleosome core? Similarly, on line 371, the authors state that “The second peak appearing at 313Å likely reflects co-existing distal arm invading conformations”. On line 390, the authors mention: “appearance of conformations in which the upstream DNA arm of the first nucleosome invades and wraps around the second nucleosome (Fig.4e, panel II) may reflect the fact that, at low ionic strength, NCPs are kept separated at “preferred” distances leading in some cases to partial unwrapping of the DNA, which in turns permits the wrapping of “foreign DNA””. Again, I do not see how these conclusions were obtained. The result that DNA invades a foreign nucleosome does make sense, in light of the plasticity of nucleosomes and heterogeneity of chromatin structure in vivo and in some in vitro experiments, and might represent an additional parameter that chromatin can exploit to modulate its structure. Thus, it would be excellent if the authors could explain clearly how this important conclusion was drawn from the data.

Reply 1.1: We express our gratitude for the valuable feedback provided by the reviewers. Our conclusion regarding DNA invasion remains unchanged. In order to bolster the solidity of this conclusion and rule out the possibility, as mentioned by the reviewer, that the DNA segments are coincidentally located in close proximity to an NCP, we have provided more evidence in our new manuscript with the following reasoning: 1. An DNA invasion replace the native wrapped DNA of an NCP, therefore, one side of the NCP's DNA linker must be observed with a significantly longer length than its designed 40 bp. 2. The process of foreign DNA rewinding onto the nucleosome core particle (NCP) must result in the refilling of the previously missing DNA density on the NCP. In order to demonstrate this point, we have employed the technique of density map averaging.

In the new version, we have now rewritten the H1 section and make new figures (Extended Data Fig. 4c and 4d):

“Interestingly, ~40% of the particles displayed a conformation where the 200 bp DNA linker arm interacted with one of the NCPs (Fig. 4e). The probability of this interaction occurring with the second NCP was notably high (30%), while the probabilities for it to occur with the third and fourth

NCPs were lower at 3% each (Fig. 4e, panel II). No interaction was detected between the 100 bp DNA arm and any of the NCPs.

To determine if the 200 bp DNA linker simply interacts with the NCP surface or invades it by displacing a segment of its wrapped 601 NPS, we assessed the DNA unwrapping of NCPs. This is because an invaded NCP should display a significant unwrapping on its one side, yielding a longer linker length than expected. The presence of H1 did not affect the DNA unwrapping at the entry site (6 ± 8 bp vs. 6 ± 8 bp (without H1)), but it slightly increases DNA dynamics and unwrapping on the exit site (17 ± 19 bp vs. 14 ± 12 bp (without H1)) and shows a second peak at ~ 45 bp (Extended Data Fig. 4a-b, row 6). This subset of exit-site 45 bp-unwrapping events was mainly observed in the second NCP. A longer 40 bp linker can be directly visualized in maps (Extended Data Fig. 4c), suggesting that a segment of DNA completely unwrapped from the H2A-H2B heterodimer, leading to a spontaneous site exposure^{45,46}. To confirm that the foreign 200 bp DNA linker wrapped onto this partially exposed histone core, we boxed, aligned and averaged together the density maps of the these unwrapped second NCP (Extended Data Fig. 4d). The absence of any missing DNA density indicates that exposed histone surface has been rewrapped by DNA, confirming that foreign DNA invasion has occurred.”

Comment 1.2: On line 385, the authors mention that “the presence of H1 locks the two DNA linkers/arms of an NCP, yielding a similar outcome as high salt concentrations by drawing them closer together”. I do not understand what the authors mean by “locks the two DNA linkers/arms” and I also do not see how this was concluded from the data.

Reply 1.2: To ensure clarity of the intended meaning for the reader, the following sentence has been rephrased:

“On the one hand, when H1 binds to linker DNA junctions near to the NCP dyad, it restricts the dynamics of the arms²⁶. These results are similar to 50 mM Na⁺ where the arms are brought closer to each other in a “closed-arm” conformation.”

Comment 1.3: The discussion of all the different angles analysed (e.g. alpha, beta, theta, theta_I, theta_II) is very hard to follow. I think it would be much easier for the readers, especially due to the interdisciplinary appeal of the work, if the authors made the effort to digest the information a bit more, e.g. describing their geometrical findings in terms of how the nucleosome entry/exit angles open or close up, or how the inter-nucleosome distances and orientations change for the different systems studied, instead of repeatedly detailing the changes in alpha, beta, theta, etc.

Reply 1.3: In accordance with the suggestion put forth by the reviewers, we have reorganized the manuscript that make sure either before or after the description of the data, there is an explanation or interpretation of their meaning. For example:

“The two components θ_{\parallel} and θ_{\perp} showed their peak populations at $\sim 9^{\circ}$ and $\sim 32^{\circ}$ (with a mean of $10^{\circ} \pm 44^{\circ}$ and $32^{\circ} \pm 45^{\circ}$), respectively (Fig. 2i). “The crossing of the two DNA linkers is more likely to be observed in the front view of the NCP plane compared to the side view.”

“The average length of DNA unwrapping at the entry site of NCPs in di-, tri-, and tetranucleosomes was found to be 4 ± 5 bp, 7 ± 7 bp, and 4 ± 8 bp, respectively (Extended Data Fig. 4, row 2-4). At the exit site, the average DNA unwrapping was 8 ± 10 bp, 10 ± 11 bp, and 14 ± 12 -bp, for di-, tri-, and tetranucleosomes, respectively. These values are similar to those observed in mononucleosomes (5 ± 5 bp and 11 ± 13 bp), which suggests that the breathing motion in arrays is also influenced by the asymmetrical stiffness of the 601 sequence^{27,34,35}.”

“In nucleosome arrays, we found that the number of NCPs is negatively correlated with the peak position of the θ angle, with a value of 55° , 47° , and 41° for di-, tri-, and tetranucleosomes, respectively. This trend is more pronounced in the θ_{\parallel} component (-40° (di-), -21° (tri-), and -7° (tetranucleosome)) than in the θ_{\perp} component (-38° (di-), -37° (tri-), and -35° (tetranucleosome)) (Extended Data Fig. 5a, row 2-4). When $\theta_{\parallel} < 0$, it represents a ‘closed/cross-arm’ conformation along the NCP plane (Extended Data Fig. 5b-c). Our result indicates that when one nucleosome is added to form a dinucleosome, there is a significant increase in the percentage of ‘closed-arm’ conformations ($\sim 80\%$) compared to mononucleosomes ($\sim 50\%$). However, this change gradually diminishes when three or four nucleosomes present in the arrays (Extended Data Fig. 5a), which suggests a unique property of dinucleosome arrays.”

“While the β angle displayed no observable change in different nucleosome arrays (all peaks were at -9° - -8°), the α angle displayed a left-skewed distribution with a major peak at $\sim 32^{\circ}$, similar to the value observed for mononucleosomes (39°), and a minor peak between -8° and 0° , which varies from mononucleosomes (15°) (Fig. 3c and Extended Data Fig. 6a-d, rows 1 and 2). This minor peak is more apparent in dinucleosomes, indicating a higher likelihood of observing DNA linkers closer to the NCP surface, which is consistent with its high percentage of “closed-arm” conformations. Conversely, tri- and tetranucleosomes exhibit significant similarities in both α and β angles (Fig. 3c and Extended Data Fig. 6a-d, rows 3 and 4).”

“An examination of the distance separating consecutive NCPs (i vs. $i+1$) within arrays revealed that the distance between their centers of mass is not influenced by the number of NCPs (221 \AA , 233 \AA , and 237 \AA nm for di-, tri-, and tetranucleosomes, respectively), as shown by the mean of the major peak distribution (Fig. 3d and Extended Data Fig. 7b, row 1-3). The agreement of these values with the theoretical calculation using two NCP hemispheres and a DNA rod ($2 \times 50 \text{ \AA} + 3.4 \text{ \AA/bp} \times 40 \text{ bp} = 236 \text{ \AA}$) supports the presence of a ridge linker within the arrays.”

And etc.

This revised version of manuscript is anticipated to offer readers improved guidance and comprehension of the presented material.

Comment 1.4: The authors perform Targeted MD coarse-grained simulations of mononucleosomes with the SIRAH force field. Then, the authors state that these simulations “displaying the thermally-induced dynamics of the particles in solution” and “These results demonstrate the capability of the use of individual particle cryo-ET methods to study nucleosome array dynamics in the following studies.” However, targeted MD is a method that biases a conventional MD simulation to enforce a transition between two equilibrium states (e.g. state I and II) by using moving harmonic restraints. Hence, the trajectories stemming from targeted MD are nonequilibrium trajectories that do not reflect the true unbiased dynamics of the system. In other words, the dynamic information that is extracted from TMD is meaningless. Regarding these simulations, very limited information is given about how the simulations were performed, and in their present form cannot be reproduced. Why the SIRAH model was chosen also needs to be discussed. If an unbiased MD approach had been used, the dynamics reported would only be as good as the underlying model used; thus, explaining how the SIRAH model has been validated for nucleosome arrays is important.

Reply 1.4: We would like to thank the reviewer for this comment. It is important to clarify that the purpose of conducting morphing simulations is solely for visual display, aiming to vividly showcase the diverse particle conformations transitioning from the most populated state to the rare state. As indicated by the reviewer, we refrain from drawing any conclusions or making any statements based on this short paragraph of simulation. The morphing was accomplished by TMD, which excels the Chimera for its morphing function that induces kinks and DNA strand separation during our initial testing. Since nucleosome arrays involve a large number of atoms, we utilized SIRAH, a coarse-grain force field, to accelerate the morphing.

We respectfully disagree with the reviewer's assertion that these simulations are meaningless. Unlike conventional structure averaging studies, only a few structures are necessary to present to the reader. Despite our best efforts to describe the array morphology by measuring various parameters statistically, direct visualization remains crucial to the reader to comprehend the flexible nature of these structures, particularly because these open array structures have not been previously reported. With regard to the rigor of our claim, we agree to remove the term “thermally induced dynamics, in solution” and rephrase the sentence accordingly.

*“Through interpolation of the intermediates between each two consecutive ordered structures by target molecular dynamics (TMD) simulations⁴⁰, mononucleosomes are morphed from the most populated state to the rarest state, displaying the **pseudo-dynamics** of the particles.”*

And in the method section:

“Visualization of the nucleosome array **pseudo-dynamics”**

*The pseudo-dynamics of nucleosome array was simulated by morphing through array conformations in an order followed by the hierarchical clustering. The morphing begins with the structures at the lowest branch of the dendrogram and stops at the highest branch. **As the number of atoms in the nucleosome arrays reaches up to 120,000, a coarse-grained approach using the Amber SIRAH force field definition⁸⁴ is employed prior to conducting Targeted Molecular Dynamics (TMD) simulations. The TMD was performed in NAMD2 engine⁴⁰, where the coarse-grained structure was steered from one conformation toward to another.**”*

Comment 1.5: The authors develop space-filling models (spheres and tubes) of 100-nucleosome arrays using the geometric descriptors obtained from their cryoET experiments. Then they use two approaches, which they describe as “seamless assembling” and “random docking” to build high-density structures with such 100-nucleosome arrays. It is interesting to try to extrapolate the single-molecule behaviour of 100-nucleosome arrays to the organisation that they might have within a biomolecular condensate. However, phase separation is an emergent property of chromatin, and there is no reason to expect that the single-molecule structural properties of chromatin arrays will remain unchanged within condensates. On the contrary, the expectation should be that chromatin structure will change when it goes from the “gas phase” to the “liquid phase”. Nucleosomes within a single chromatin array can only increase the enthalpy of the system by establishing interactions with other nucleosomes in the same array, which favours a given type of chromatin arrangement. In contrast, nucleosomes within a chromatin condensate contribute to increasing the enthalpy of the system by forming interactions with multiple arrays—indeed multivalency is critical to yield the percolated network of the condensed liquid—and hence, phase separation likely favours a different type of chromatin arrangement. Therefore, while this space-filling task allows one to imagine how 100-nucleosome arrays might look inside a condensate, I do not think this exercise is sufficient for drawing conclusions about the structure of chromatin inside condensates. For instance, line 424 states “the 30-nm fibers generated in this simulations were not often observed” in the experiments “one possible explanation for this discrepancy is the low probability of capturing fibers lying exactly on the plane of the thin slice cell section”, when the in-silico structures might be completely artificial. On line 435, the authors say that “we repeated the above simulations by explicitly preventing DNA unwrapping” “but did not induce an obvious change of the condensate’s density in either salt conditions” “suggesting that breathing does not increase enzyme accessibility to the interior of the condensates”—which again is a conclusion that cannot be unequivocally derived from their data.

Reply 1.5: We acknowledge the valuable feedback from the reviewers and appreciate their support for our idea of extrapolating single-molecule behavior to chromatin fiber organization. However, a major issue highlighted by the reviewers pertains to the scope of our study, as parameters from the non-phase separated state cannot be used to construct phase-separated structures. We concur with the reviewers on this aspect. Our research aims to investigate the stage preceding phase separation, a stage intra-array compaction is initiated but in the absence of inter-array interactions. However, considering a more realistic cellular environment, where arrays are 1. connected into a single string and 2 confined within a limited nucleus space, both of which constrain the degree of freedom of the arrays, leading to the formation of some “structure patterns” even they are not phase separated. The simulation of box filling task is to explore this state and try to resolve some argument in the literature. We acknowledge that this point we are not explained adequately. To clarify the scope and purpose of our study, we have made the following major modifications:

1. Clearly define the scope of our study at begin of the simulation:

“The statistical parameters obtained provide a comprehensive characterization of the dynamic behavior of a single nucleosome array unit (tetranucleosome) in low salt environments (5 to 50 mM Na+), an ionic condition induces the intra-array compaction but precludes inter-array interactions. Therefore, it is interesting to extrapolate the single-molecule behavior from our result towards longer arrays and gain insight into the initial compaction process of the chromatin fiber from a loosely-packed state. Arrays containing 100 NCPs (hectanucleosomes) were computationally generated based on our obtained statistics.”

2. To explain the reason why we assemble the fibers, we have added section

“The structure of chromatin has been debated for a decade. Cryo-EM studies have reported both homogeneous distribution of NCP¹² as well as the presence of fiber-like structure within cells^{15,48}. This may be due to the uneven condensation of chromatin⁵⁰, which undergoes spatial and temporal transitions between a phase-separated and non-separated state. However, the existence of long range of ordered/regular 30-nm fibers obtain in vitro still lacks cellular supporting evidence. Hence, it is worthwhile to investigate whether assembling of our simulated irregular fibers shares similarities with structures found in vivo.

3. Following reviewer’s suggestion, we removed the “seamless assembling” simulation result, where nucleosomes are in close contact, falling into a condensate regime as mentioned by the reviewer.

“Since chromatin is confined within a limited volume of the nucleus⁴⁸, the generated fibers were inserted into a 200 nm cube with a “random docking” approach (see methods for details).”

4. Following reviewer’s suggestion, we removed our extrapolated conclusion “suggesting that breathing does not increase enzyme accessibility to the interior of the condensates” and only make one conclusion as following:

*“We repeated the above simulations by explicitly preventing DNA unwrapping. The resulting simulations only varying the length and width of the fibers by 11-13% (**Extended Data Fig. 10a-b, row 3-4**). Therefore, the angles of DNA linkers with respect to the NCP are more critical factors in reshaping the longer array morphology (**Fig. 5a bottom panel**).”*

Comment 1.6: Minor corrections: Some of the references point to the wrong papers.

Reply 1.6: Following the reviewer’s feedback, the wrong reference has been corrected.

Reviewer #2 (Remarks to the Author):

In this MS, Zhang et al use primarily cryo-ET and also MD simulations to study the structure of mono-, di-, tri-, and tetranucleosomes in solution. By analysis of spatially well-separated reconstituted nucleosomes and oligonucleosomes, the MS maps the angular disposition of the DNA arms. The samples studied include: mono-, di-, tri-, tetranucleosomes in 5 mM NaCl; tetranucleosomes in 50 mM NaCl; tetranucleosomes with histone H1 in 5 mM NaCl. It was unclear which samples were considered controls, making the independent variable(s) difficult to identify. Measurements were made along axes parallel to the discoidal plane (theta parallel / alpha) and perpendicular (theta perpendicular / beta). Overall, the DNA arms sample a large range of angles. This angular sampling changes (but remains broad) in 50 mM NaCl resulting in more compact tetranucleosomes.

Overall, I struggled to follow the narrative of this MS because the samples (see above) do not have adequate controls and because the presentation (text and figures) was confusing throughout.

My comments are mostly about the cryo-ET and will hopefully be useful.

Comment 2.1: DETAILED COMMENTS

1.The motivation for the study is not very convincing. How does learning about the structure of different nucleosome arrays in an environment that does not promote LLPS (i.e. low ionic strength) inform about chromatin condensation via LLPS?

2.Lines 295-298: Why are tetranucleosomes in 50 mM Na⁺ considered a condition prior to phase transition? Does phase transition occur when one gradually increases the salt concentration in the same sample? I thought a better condition would be tetranucleosomes in 150 mM Na⁺ at the 0 min time point.

3.The previous paper showed that tetranucleosomes form condensates. Do mono-, di- and trinucleosomes also form condensates? I understand the need to present the mononucleosome data as a way to show that the analysis workflow works, but I am not sure what's the point of showing dinucleosomes and trinucleosomes – especially if it is unknown whether they form condensates.

4.Lines 333-336: This experiment tells us (a) whether H1 can condense arrays at 5 mM Na⁺ (b) the characteristics of tetranucleosome arrays in the presence of H1 at 5 mM Na⁺. But how does incubating tetranucleosomes with H1 in 5 mM Na⁺ help investigate how H1 accelerates the spinodal-to-spherical conversion at 150 mM Na⁺?

Reply 2.1: We acknowledge the valuable feedback from the reviewer, which highlighted the primary concern regarding the scope of our paper. This has led to subsequent inquiries as to why we are investigating phase separation under low salt conditions, which group serves as the control, why we are studying dinucleosomes and trinucleosomes, and how H1 induce phase separation at 5mM Na⁺. All of these inquiries stem from our investigation of the particular intra-array compaction stage along the pathway of phase separation development. However, the use of the term "phase separation" may give rise to the expectation among readers that our study relates to the formation of condensate structures. We are confident that by providing a more explicit delineation of the scope of our study, we can address these related questions simultaneously.

The motivation for the study is not very convincing. How does learning about the structure of different nucleosome arrays in an environment that does not promote LLPS (i.e. low ionic strength) inform about chromatin condensation via LLPS?

1. The process of nucleosome array phase separation is not a straightforward binary process (dissolved arrays vs. liquid droplets). As described in our previous study (10.1016/j.molcel.2022.06.032), it involves several distinct stages, beginning with evenly distributed arrays and progressing to spinodal condensates, formation of small nuclei, and ultimately, large matured spherical condensates. Indeed, we have learned that even preceding to the formation of the spinodal condensates, there is a distinct step that arrays compact themselves within the molecules. This suggests that intra-array and inter-array interactions are non-synchronous processes that mediate different levels of nucleosomal condensation. **We consider this process to be a preliminary step of nucleosome array phase separation.** The investigation into their flexible structural features is imperative, as it is complementary to the ordered-fiber structures and phase separated condensates, neither of which fully accounts for the diverse spectrum of structures present in vivo.

Lines 295-298: Why are tetranucleosomes in 50 mM Na⁺ considered a condition prior to phase transition? Does phase transition occur when one gradually increases the salt concentration in the same sample? I thought a better condition would be tetranucleosomes in 150 mM Na⁺ at the 0 min time point.

2.The use of a low salt condition has been a conventionally employed method to induce open chromatin fibers and has also been previously tested to decelerate the phase separation of tetranucleosome arrays (10.1016/j.molcel.2022.06.032). We concur with the reviewer that

utilizing tetranucleosomes in 150 mM Na⁺ at the 0 minute time point would represent an ideal condition. Nevertheless, it is not feasible due to the sample deposition time, blotting time, and plunge-freezing time required. In a previous study, we employed a similar low salt strategy to achieve a comparable condition (10.1016/j.molcel.2022.06.032). The selection of 5-50 mM Na⁺ ionic strength in this study was made specifically to capture the early intra-array compaction state. The effect of 50 mM Na⁺ as a boundary condition was tested to generate highly sparse condensates consisting of oligo-tetranucleosome arrays. (while dashed circle).

Tetranucleosome at 50 mM Na⁺ z-dimensional slice

The previous paper showed that tetranucleosomes form condensates. Do mono-, di- and trinucleosomes also form condensates? I understand the need to present the mononucleosome data as a way to show that the analysis workflow works, but I am not sure what's the point of showing dinucleosomes and trinucleosomes – especially if it is unknown whether they form condensates.

3. Following the characterization of the structural dynamics of a mononucleosome with two flanking arms under 5 mM Na⁺ as a reference, we loaded increasing number of NCPs to the template to determine whether the interplay among NCPs are sufficient to neutralize the repulsive DNA and initiate array compaction. Although no compaction was observed, we discover that the structural dynamics of the longer arrays become stabilized starting from tri- and tetranucleosomes. This supports the inference that the trinucleosome is the minimum folding unit and the tetranucleosome is the fundamental building block of chromatin fiber.

Lines 333-336: This experiment tells us (a) whether H1 can condense arrays at 5 mM Na⁺ (b) the characteristics of tetranucleosome arrays in the presence of H1 at 5 mM Na⁺. But how does incubating tetranucleosomes with H1 in 5 mM Na⁺ help investigate how H1 accelerates the spinodal-to-spherical conversion at 150 mM Na⁺?

4. Thanks for the feedback from reviewer. The tetranucleosome at 5 mM Na⁺ was used as control. H1 and higher ionic strength (50 mM Na⁺) were served as two independent factors introduced to the control system to explore their effect on the initial intra-array compaction dynamics. In this section, our objective is to determine whether H1 and higher ionic strength have a comparable impact on inducing array compaction or condensation. In the sentence: “*Considering the **Histone H1 catalyzes the spinodal-to-spherical conversion** for more than 10-time faster than that absents H1²⁹,*

it is interesting to investigate how H1 alone contributes to speeding-up the conversion during the regulation of the structural dynamics of the arrays”, The phrase "**Histone H1 catalyzes the spinodal-to-spherical conversion**" was used to introduce H1, rather than being our goal in this study. To avoid potential confusion, we have modified the sentence into:

“Since linker histone H1 plays a catalytic role in the nucleosome array phase separation¹⁸ and its binding depend on the linker trajectory of the array⁴⁷, we determine if H1 can condense the loosely-packed nucleosome arrays and how this process is achieved.”

In this study, we elucidated various 3D structures of irregular arrays exhibiting distinguished modes of compaction in response to the presence of H1 and higher ionic strength. While the ordered array 3D structures were frequently reported in vitro, their counterparts in vivo remain difficult to identify. The irregular array 3D structures obtained in our vitro study provide this piece of missing information and can serve as a basic model to build upon for a deeper insight into the higher-order open chromatin structure.

We are agreeing this information are critical to readers and now has been in cooperate to our manuscript introduction section second and third paragraph as following:

“Nucleosome array phase separation involves a multi-stage progression, which starts with evenly distributed arrays and progress towards the formation of spinodal condensates, small nuclei, and ultimately, large matured spherical condensates¹⁸. It has been proposed that at an earlier stage, nucleosome arrays undergo self-compaction¹⁹ before the formation of spinodal condensates (Fig. 1a Extended Data Fig. 1a-b). This suggests that intra-array and inter-array interactions are non-synchronous processes, with the former preceding the latter, thereby governing different levels of nucleosomal condensation. Therefore, we consider the nucleosome array self-compaction to be a preliminary step in their phase separation. Despite considerable research on the dynamics of intra-nucleosome array compaction²⁰⁻²², the lack of a 3D structural-based characterization persists because the large dynamics of the arrays at this stage prevent their structural averaging. The investigation into their flexible structural features is imperative, as it is complementary to the ordered-fiber structures and phase-separated condensates, neither of which fully accounts for the diverse spectrum of chromatin structures present in vivo.

To capture the dynamics of the early steps of nucleosome array compaction, we used cryo-electron tomography (cryo-ET) to take snapshots of the 3D structure of individual mononucleosomes and nucleosomal array particles. By analyzing short nucleosome arrays containing two, three, and four NCPs, we were able to study the impact of each nucleosome on intra-nucleosomal interactions and their role in the formation of higher order structures. Our approach enables us to visualize the highly dynamic linker DNA extending from the NCPs. We found that starting from tri- and tetranucleosomes, the array structural dynamics stabilized, supporting the idea that they constitute the minimum folding unit of chromatin^{7,23,24}. Both increasing the ionic strength and adding linker histone H1 causes the DNA linkers to adopt a

“closed” form, but they result in different modes of array compaction. These observations provide insight into intra-array interactions associated with the preliminary stage of array condensation.”

Comment 2.2: Some tetranucleosomes look like they have 5 nucleosome core particles (e.g. particle #132, #155, #156, #158). These were not addressed.

Reply 2.2: In the original MS, we do have addressed this issue in the section of Evaluate the fitting model: *“The off-template NCPs (binding to the 100 or 200-bp distal arms region) are removed from the following statistical analysis.”*

But thanks for review pointing it out. We have now added more explanation to this section:

“... using the statsmodels package in Python. The dynamic nature of histone assembly and disassembly on DNA templates can lead to the NCP localizing at positions other than the canonical 601 sites. This phenomenon occurs more frequently when there is a free 200 bp DNA arm, a higher histone: DNA loading ratio (e.g., in tetranucleosomes), and under low salt conditions that increase non-specific binding. The off-template NCPs (binding to the 100 or 200-bp distal arms region) are removed from the following statistical analysis.”

Comment 2.3: Some particles had unresolved DNA arms (e.g. particle #132, #136, #137, #138). In these cases, how do you assign which is the entry or exit arm? One way the model fitting was evaluated was to measure the correlation between the measured DNA linker DNA between two particles from the density and the unwrapping angle from the fitted model (under methods: see lines 698 – 710). While this shows that the visualized model does fit into the density, it does not show that this is the only model that can fit into this density. This is particularly important for more ambiguous densities such as the condensed tetranucleosome array in 50 mM Na+.

Reply 2.3: Thanks for reviewer raise concern about the detailed method how we fit the model. Our modeling fitting and distal arm determination was based on two assumptions that nucleosomes are positioned on 601 regions, and secondly, that the unwrapping of one side of a nucleosome corresponds to an increase in the strength of wrapping on the other side. This was supported by findings from several single-molecule optical tweezer experiments. Given this framework, identification of one of the 40 bp linker, or 100 bp entry arm or 200 bp exit arm within an array can serve as a reference for fitting the remaining nucleosome core particles. Although the additional nucleosome core particles (NCPs) loaded onto the two distal arms (namely #132, #136, #137, #138) resulted in difficulty distinguishing them, the 200 bp arm adheres to the rule of $a + 147$ (for the extra octasome) + b ($a+b=53$), while the 100 bp arm adheres to the rule of $0 + 100$ (for the hexasome). We have incorporated a more elaborate description of this approach in our methodology section.

“Then, we fit the crystal structure of *Xenopus laevis* NCP into the discoidal-shaped high-density region. In this step, our fitting was based on two assumptions: (1) nucleosomes are positioned on 601 regions and (2) unwrapping of one side of a nucleosome corresponds to an increase in the strength of wrapping on the other side^{43,71} (i.e. only one side of NCP unwraps at a time). Therefore, in the case of an unwrapped NCP density map, it is only necessary to take into account two potential fitting solutions (**Extended Data Fig. 6, e**). Given this framework, the identification of any DNA segment corresponding to a 40-bp, 100-bp, or 200-bp length can help to determine the fully wrapped side of an NCP, which then serve as a reference point for fitting the remaining NCPs (see evaluation in below section). For the distal arm orientation determination, though additional nucleosome core particles (NCPs) may load onto the two distal arms resulted in difficulty distinguishing them in few cases, the 200-bp arm adheres to the rule of $a + 147$ (for the

extra octasome) + b, where $a + b = 53$, while the 100 bp arm adheres to the rule of $0 + 100$ (for the extra hexasome).

We also add a schematic and statistics to the section.

“To evaluate the self-consistency of the above fitted models, the correlation between the measured DNA linker length between two consecutive NCPs (named as $L(n)$) and the estimated DNA unwrapping level between the same NCPs were calculated (**Extended Data Fig. 6, f**). The $L(n)$ was acquired from the experimental data, in which distance between the DNA exit position of the n th NCP and the entry position of the $(n+1)$ th NCP from the density map were measured ($L(n) = d(n) + d(n + 1)$). The DNA unwrapping level was estimated from the fitted model, in which the unwrapping angles from the exit side of n th NCP and entry side of the $(n+1)$ th NCP were added together ($\varphi(n) = \varphi_{ex}(n) + \varphi_{en}(n + 1)$). The Pearson correlation coefficient, $r=-0.9$ was calculated from the linear regression fitting of the unwrapping length $L(n)$ against the angle $\varphi(n)$ using the statsmodels package in Python (**Extended Data Fig. 6, g**).”

Comment 2.4: How thick are the cryo-EM samples? Are the DNA arms or nucleosomes interacting with the air:water interface? How are mono, di-, tri-, and tetranucleosomes picked from the tilt series to be used for analysis? Was it random or was there a criteria (if so, what criteria)? Were they picked by hand or by some software? Lines 189-192 and 251-252. Could ice thickness contribute to restricted beta angles? Lines 447-451: The statistical mechanical arguments here need a caveat that the possible interaction between the chromatin complexes and the air:water interface (see above) may bias the distribution from what would be found in bulk solution.

Reply 2.4: The thickness of the ice of our sample falls within a standard range of approximately 80 to 100 nm. The samples were embedded in the center of the ice, rather than on its surface. No discernible conformational restrictions imposed by the ice surface were detected. This can be also be confirmed from the top view (x-y direction) and 90-degree rotation side view (x-z direction) from our supplementary 223 particle list. Thanks reviewer for raising concerns of the ice thickness, we have addressed them both in the main text and figures to enhance the integrity of our manuscript.

“Similar to mononucleosomes, these short nucleosome arrays are flanked by 200 bp and 100 bp DNA linkers at two ends, with a 40 bp DNA linker between two consecutive 601 NPS. **These arrays were embedded in the ice center (~70-90 nm thickness) with no apparent conformational restrictions (Extended Data Fig. 3d) and displayed an overall asymmetric zig-zag architecture (Fig. 3a and b).**”

Also, explanation regarding to particle picking was added to methods section:

“By labeling a marker on the particle center, all nucleosome arrays within the scope of the large tomogram are manually selected without bias. The tilt series of each of the targeted particles were semi-automatically tracked and then windowed in square windows of $\sim 1,000 \times 1,000$ -pixel size”

Comment 2.5: The title should be revised to more accurately say what the data represent. As written, it implies that the structures visualized by cryo-ET (the dominant portion of the MS) have captured conformational changes during a state change (“prior” to phase separation). A more accurate title would be: “3-D structural diversity of mono, di-, tri-, and tetranucleosomes in low-salt conditions”. While this variant title is rather dry, it is far more accurate than the current one.

Line 513: The title should read “reconstitution and purification”

Reply 2.5: Thanks for the feedback from reviewer. The state preceding phase separation is the precise condition under our investigation. Specifically, the array underwent initial internal compaction to facilitate subsequent inter-array interactions. We recognize the potential for misinterpretation by our readers and appreciate reviewers’ concerns. As a compromise, we have opted to revise our title to more accurately reflect the scope of our research. **“Structural Insights into Nucleosome Array Compaction by Cryo-Electron Tomography”**

Comment 2.6: The way the experiments were done requires more explanation: Performing Volta phase plate imaging with a high overfocus is not commonly done (this MS used $2.5 \mu\text{m}$; typically $-0.5 \mu\text{m}$ defocus is used), and the resulting tilt series have a much lower contrast than expected (Figure 1D). What is the rationale behind using a high overfocus? If there was a typo here and the actual defocus was $-2.5 \mu\text{m}$, why was such a large underfocus used? The amplitude and phase of the CTF was corrected using multiple CTF correction programs. What is the rationale behind using multiple programs? Please add more detail on how the CTF correction was done.

Line 601: Presumably, the intended text was “ $2.5 \mu\text{m}$ underfocus”. If this is the case (or if they used overfocus), please explain why such a large defocus was used with a Volta phase plate. The inventors of the Volta phase plate suggest imaging closer to focus (though not at $0 \mu\text{m}$ defocus). See my related comment above. This issue comes up again at line 608-609.

Lines 630-631: “Phase and amplitude were corrected by SPIDER, GCTF, and TOMOCTF”. Is that necessary? And how was that performed? All 3 software packages correcting at once or sequentially? Is this part of the NOISE2NOISE? See my related comment above

Reply 2.6: Thanks for reviewer pointing out this typo several times. We have corrected this typo in the manuscript:

“The untitled cryo-EM micrographs were acquired with a Volta phase plate near the focal plane using a Gatan K2 Summit direct electron”

Our previous experiment indicate that image contrast plays a crucial role in the successful reconstruction of individual particles using cryo-electron tomography. As such, we prioritize optimizing contrast even if it means sacrificing some resolution under a defocus of 2.5. For the CTF correction, a detailed step is listed in new version of MS:

“The motion of the cryo-ET frames was corrected by MotionCor2⁶⁷. To reduce the cryo-ET image noise, we followed a machine learning method (NOISE2NOISE method) as described³⁸. The defocus value of both cryo-ET and NS-ET tilt series was measured by using GCTF⁶⁸. During data collection, a carbon area perpendicular to the tilt axis was included to aid in the detection of contrast transfer function (CTF) when particles were scarce. The phase and amplitude of the determined contrast transfer function (CTF) were corrected by TOMOCTF⁶⁹ based on strip-based periodogram averaging method ($\Delta D=1000$, $w1=0.7$, $w2=0.25$).”

Comment 2.7: The reported parameters sometimes do not make sense e.g. reporting the average of a skewed distribution (c.a. Fig. 2e). Shouldn't the mode of the values be presented instead?

Figure 2e: This is not a normal distribution, so please consider reporting the mode, not the average value.

Reply 2.7: Thanks for the feedback provided in the reviews. As per the suggestion, we have incorporated the peaks, i.e. mode values, to effectively describe those skewed distributions in our study

“The angle θ of NCPs are distributed over a wide range from $\sim 0^\circ$ - 140° (with a peak, i.e. mode at $\sim 39^\circ$, and a mean \pm std of $46^\circ \pm 27^\circ$)”

“Its two components θ_{\parallel} and θ_{\perp} showed their peak populations at $\sim 9^\circ$ (with a mean \pm std of $10^\circ \pm 44^\circ$) and $\sim 32^\circ$ ($32^\circ \pm 45^\circ$) respectively”

And etc....

Comment 2.8: The manuscript tries too hard to draw parallels with chromatin function(transcription) and cellular phenotypes (condensation in vivo / in situ). For example: **Lines 85-88:** Implies that the chromatin in this study is transcriptionally active. Is there any experimental evidence to justify this description? The same appears in **lines 91-92**. This statement also implies that somehow the structural changes going from 5mM Na⁺ to 50 mM Na⁺ phenocopies changes that affect “gene expression”. The cell's interior does not, to my knowledge, have a 10-fold variation in sodium concentration, nor do cells change their ionic strength so much through the cell cycle. Also, there are numerous other intracellular factors that affect chromatin condensation that are not tested in this MS. **Line 95:** The Na⁺ concentrations (up to 50 mM) are not physiological, which the text acknowledges in line 286 when referring to the predecessor study. This deviation from physiological-ness is at odds with the implication that this study somehow mimics transcriptionally active chromatin.

Lines 95-98: The mention of “external stimuli” is too vague. Please give more explicit examples of how changes in oligonucleosome structure induced by increasing the Na⁺ concentration or adding histone H1 would inform on a cell's response to external stimuli.

Lines 585-586: 4°C is not physiological for most of the organisms in which chromatin is studied. If the goal is to relate this study's experiments to transcription or to cells, the experiments should be repeated at 30°C or 37°C. This is also not a temperature that in vitro transcription assays would be done at. Alternatively, as I suggested above, remove all the implications about function/cells.

Reply2.8: Thanks for the reviewer's suggestion. These comments are all focus on the specific word used of "transcription active" and "gene expression" in above three places in our introduction section and two places in conclusion. Following reviews suggestion, we have changed the phrase of "loosely packed transcriptionally active state" into a more accurate "loosely-packed array"/"loosely-packed 10-nm fiber" and remove the sentence relate to "gene expression" accordingly.

In the abstract:

"The conformational dynamics of nucleosome arrays generate an ensemble of differently populated microscopic structural states, whose equilibrium shifts modulate the large-scale chromatin morphology. However, this intrinsic flexibility hinders the arrays' structure determination. Here, we used cryogenic electron tomography (cryo-ET) to determine the three-dimensional (3D) structure of each individual particle of mononucleosomes, and di-, tri-, and tetranucleosome arrays. Statistical analysis revealed distinct regulatory modes for intra-array compaction through the influence of ionic strength and linker histone H1 on the angle between DNA linker and nucleosome core particle (NCP). The finding that one-third of the arrays in the presence of H1 contain an NCP invaded by foreign DNA suggests an alternative function of H1 in constructing nucleosomal networks. These experimentally derived irregular-array structures and their conformational changes not only complement the ordered-array structures and phase-separated nucleosomal condensates but also provide valuable insights into the initial step of chromatin structural compaction and condensation."

In the introduction: see reply 2.1

In the conclusion section:

"The nuclear environment of cells facilitates chromatin condensation^{54,55}. However, in certain nucleus domains and nuclear extracts^{15,52}, the loosely-packed "10 nm-fiber" persist due to the interaction of multiple factors such as nuclear proteins, histone acetylation, ionic strength, pH, and temperature^{16,18,56}. To systematically control the experimental variables that induce chromatin compaction in vitro, we studied the structural changes in loosely packed nucleosome arrays under low salt conditions before inter-nucleosome array interactions occurred. Using cryo-ET, the conformation of mononucleosomes and arrays without structural averaging was determined. We found that increasing ionic strength reduced the angle between the DNA linkers and the NCPs, resulting in a smaller and denser asymmetric tetranucleosome unit. H1, on the other hand, compacted the array by both reducing the DNA linker dynamics and enabling foreign DNA to enter NCP, creating alternative nucleosomal networks. Our computer-generated chromatin structures, based on statistics from individual array structures, resemble diverse in-situ chromatin tomographic slices. These new insights into the conformational changes preceding intermolecular interactions of nucleosomal arrays advances our understanding of how nucleosomal arrays are regulated during the initial phase of condensation. "

Comment 2.9: Line 158: The section heading implies that the data will reveal "dynamics". Indeed, in the way the rest of the manuscript is written, the implication is that the cryo-ET data show dynamics. It would be more accurate to call this section "3D structural variability of mononucleosomes". For cryo-ET data to show dynamics, a reaction-initialization device and plunge-freezing like the one from Joachim Frank's lab would be needed.

Lines 200-202 “These results demonstrate the capability of the use of individual particle cryo-ET methods to study nucleosome array dynamics in the following studies.” It’s more accurate to say that cryo-ET initializes a molecular dynamics study. To truly study dynamics by imaging, one would have to image the same unique particle as a function of time, like in liquid-cell EM or timelapse light microscopy.

Reply 2.9: We respectfully disagree with review on this point. The term “dynamics” has been frequently employed in single-particle analysis (SPA) cryo-electron microscopy (cryo-EM), where particles are collected at the same time point but exhibit differences in conformations. The popular SPA software, RELION, states “multi-body refinement can be useful to gain unique insights into the structure and dynamics of large and flexible macromolecular complexes”. Some SPA studies titled with “Structure and dynamics of a 197 base-pair nucleosome in complex with linker histone H1”. These assumptions are rooted in the Ergodic theory, which posits that the average values of physical quantities characterizing a system over time are equivalent to the statistical average values of those quantities. The most prevalent state/conformation possesses lower energy than the infrequent states/conformations, which dictate the motion direction of particles.

Comment 2.10: Lines 235-244: The discussion about the role of electrostatics on DNA arms in the context of oligonucleosomes is confusing. The impact of the DNA arm(s) connecting the dinucleosomes (or nucleosome N with N-1 and N with N+1) is ignored. Consider doing a simulation to back up these claims.

Reply 2.10: Thanks for reviews comment. Given that the 40-bp DNA linker is very rigid (as it significantly less than the DNA persistence length of ~150-bp and have shown in many nucleosome array structural studies, 10.1126/science.1251413, j.molcel.2018.09.027) and also the 40-bp DNA linker serves as a constant parameter across all arrays investigated in our study, we reason that the differences in observed dinucleosome arrays must come from their long flexible distal arms. We have included an explanation of this point in the revised version of our manuscript, in response to the reviewer's suggestion:

“As the 40 bp DNA linker is rigid (considerably shorter than the DNA persistence length of ~150 bp) and remains constant across arrays, observed differences in dinucleosome must arise from their long distal linkers. In fact, dinucleosome is the only array structure where its two negatively charged 200 bp and 100 bp DNA linkers lie in close proximity and hindered each other”

Comment 2.11: Lines 339-340: Looking at the distribution in Extended Data Figure 4, and the standard deviation values, I’m not sure the values of -17 ± 19 -bp versus -14 ± 12 -bp represent any meaningful difference. These values are said to represent a “slightly increased unwrapping level and dynamics”, which is confusing because the unwrapping values are negative, which should be interpreted as rewinding. Lines 340-343: The text here says that -45 bp constitutes unwrapping, but I had thought that only positive values would constitute unwrapping while negative values constitute rewinding.

Figure 2d and lines 166-168: It’s stated that on average, mononucleosomes are unwrapped by 5 bp and 11 bp on the entry and exit side respectively. But in this figure, the values “-5” and “-11” appear instead, alongside grey dotted lines that appear to mark the 0 bp unwrapping. Is the “average unwrapping” calculated based only on the positive unwrapping values? Because the meaning of the sign (positive/negative) is not well defined, it’s hard to know if the distributions imply that most nucleosomes are partially unwrapped or overwrapped.

Reply 2.11: We appreciate the feedback from the reviewer. We would like to note that the unwrapped NCPs in the presence of H1 comprise only a minor population, accounting for 30% of the total population and 1/4 of the NCPs. As suggested by the reviewer that the mean may not a good way to distinguish this difference, we have added peak position label to the figure to Extended Data Figure 4 to indicate the emergence of the second population, which emphasizes our intended point. A new Extended Data Fig. 4c was created to support our claim.

“The presence of H1 did not affect the DNA unwrapping at the entry site (6 ± 8 bp vs. 6 ± 8 bp (without H1)), but it slightly increases DNA dynamics and unwrapping on the exit site (17 ± 19 bp vs. 14 ± 12 bp (without H1)) and shows a second peak at ~ 45 bp (Extended Data Fig. 4a-b, row 6). This subset of exit-site 45 bp-unwrapping events was mainly observed in the second NCP. A longer 40 bp linker can be directly visualized in maps (Extended Data Fig. 4c), suggesting that a segment of DNA completely unwrapped from the H2A-H2B heterodimer, leading to a spontaneous site exposure^{45,46}.”

Regarding the second inquiry, we invert the sign of unwrapping from negative numbers to positive numbers to avoid the confusion. Now, the negative value zone is interpreted as "overwrapping" rather than rewrapping. Due to the considerable variability in arm angles, a minor subset of DNA arms protrudes along the surface of the NCP, i.e. did not exit the histone "zone" immediately beyond the exit point. We have appended the definition and elucidation to the corresponding figure caption.

Figure 1d :“d, Histograms showing DNA unwrapping distributions for the entry (left panel) and exit (right panel) side of the NCP. The positive values represent the events of unwrapping. The negative zone indicates that the fully wrapped arms extend along the surface of the NCP beyond the exit point. “

Minor points:

Comment 2.12: Line 109: Please explain the choice of a 447-bp-long?

Reply 2.11: *“We assembled histone octamers on the 147 bp 601-nucleosome positioning sequence (NPS) flanked by 200 bp and 100 bp of DNA linker on its entry and exit sites, respectively. This allowed us to distinguish the orientation of the nucleosome and facilitated the formation of highly positioned nucleosome arrays.”*

Comment 2.13: Line 125-126: Why would the 2D imaging results become “the basis for the following 3D reconstructions”?

Reply 2.13: Given that the 3D reconstruction results from the back projection of 2D images acquired at various tilt angles (Fourier slice theorem), a better quality of the 2D images constitutes the initial step toward generating an improved 3D map. Based on reviewer’s comment, we have removed this unnecessary sentence.

Comment 2.14: Line 175: The theta angles approach 180° . At these extremes, the two DNA arms could either be crossed or alternatively, the “core” DNA could be half detached, resulting in only one gyre of DNA. To distinguish the two possibilities, the DNA gyres would have to be resolved (~ 2.5 nm resolution). However, the data was low-pass filtered to 4.5 nm (line 661). Also, the resolution Please explain this discrepancy.

Reply 2.14: In Line 175, We wrote “the angle θ of NCPs are distributed over a wide range from $\sim 0^\circ$ - 140° ” instead of 0-180. In line 661, we have now explicitly stated that the low pass filter was

perform by EMAN 'lp' function, which is gaussian function to enhance the low frequency signal, the high frequency information is not completely being removed. Also, these maps were use for the final display in chimera. Following reviewers' suggestion, we have implemented this information into our method section:

"All final density maps were low-pass filtered to 4.5 nm using EMAN⁷¹ 'lp' function with a Gaussian filter and displayed using UCSF Chimera"

Comment 2.15: Line 187 and Fig. 2j: I see a right skew, not a left one.

Reply 2.15: By definition, a distribution is left skewed if it has a "tail" on the left side of the distribution. We consider Fig. 2j is left-skewed comparing to the example (<https://www.statology.org/left-skewed-vs-right-skewed/>).

Comment 2.16: Line 194: It's more accurate to say either "speculate about the dynamics" or "simulate the dynamics" instead of "display the dynamics".

Reply 2.16: Based on reviewer's suggestion, following change has been made: "To present the dynamic behavior of the mononucleosome,....."

Comment 2.17: Line 196-197 It was very hard to find "supplementary figure 3" in the 250-page supplementary figure file. I suggest reordering or giving readers a hint that it's on page 53.

Reply 2.17: We acknowledge the suggestion from the reviewer on this point. We have created a content list at the first page of the supplementary information file, which helps the reader to locate the figures.

Comment 2.18: Line 285: The "previous study" should be ref 28.

Reply 2.18: We acknowledge the suggestion from the reviewer on this point, the error reference has been corrected.

Comment 2.19: Line 398: "a longer linker" – Isn't the linker length constant in the constructs used?

Reply 2.19: We have modified this section base on reviewer 1, "To assess whether the 200 bp DNA linker interacts with the NCP surface or invades it by displacing a segment of the wrapped 601 NPS, we examined the DNA unwrapping of NCPs. If an NCP is invaded, it should exhibit significant unwrapping on one side, resulting in a longer linker length than anticipated."

Comment 2.20: Lines 424-430: The cryolamella referenced is ~160 nm thick and the cryosections referenced are ~70 nm thick, so the argument that too thick of a section (30 nm) would hide 30-nm fibers is incorrect. Also, note that Scheffer's chicken erythrocyte paper was done with 50-nm thick cryosections and the 30-nm fibers are easily visible.

Reply 2.20: We appreciate the feedback from reviewer on this point. Our objective in generating a slender slice from our cube was to draw comparisons with in vivo cryo-experimental outcomes and establish similarities in support of existence of irregular fiber model. Though the cryolamella referenced is ~160 nm thick and the cryosections referenced are ~70 nm thick as mentioned by reviewer, the data was presented in thin slices in those papers.

For example: in "The in-situ structures of mono-, di-, and trinucleosomes in human heterochromatin", FIGURE 1. (A) Tomographic slice (**20 nm**) of the nuclear periphery of a HeLa cell. ... (C, D) Tomographic slices (**10 nm**) of the (C).

In the "Evidence for short-range helical order in the 30-nm chromatin fibers of erythrocyte nuclei" Fig. 1. (A) A 2D overview image of an approximately **50-nm-thick** cryosection of a chicken erythrocyte nucleus. (B) A **10-nm-thick slice** through a 3D reconstruction of a chicken erythrocyte nucleus.

To establish comparison to those data, which present in 10-50 nm slice, we specifically selected a 25 nm thickness slice from our cubic volume. In response to the reviewers' suggestion, we have removed the sentence discussing the challenges of finding the 30-nm fiber in situ and made the following modifications in our manuscript:

"These values are consistent with the experimentally determined range of 80-520 μM in vivo ⁵¹ and ~340 μM in vitro ^{16,18}. The distribution of NCPs in a thin volume slab (25 nm thickness) (Fig. 5c-d left panel) is also similar to those observed in tomographic slices of HeLa cells (10-20 nm thickness)³⁸, chicken erythrocyte nucleus (10-50 nm thickness)⁴⁸, picoplankton cell lysates (30 nm thickness)⁵², and in 2D views of swollen HeLa S3 chromosomes at low Mg^{2+} concentration¹².

Comment 2.21: "mono-, di-, tri-, and tetranucleosome arrays" Mononucleosomes are not arrays

Reply 2.21: We acknowledge the suggestion from the reviewer. The word arrays have been deleted accordingly

Comment 2.22: "Intramolecular" and "intermolecular" seem to be used interchangeably throughout.

Reply 2.22: We identified a single instance of word misuse and rectified it accordingly "responds to external stimuli ²⁹ prior to the intermolecular interaction and phase transition"

Comment 2.23: Line 50: "reorganization of nucleosome arrays" – This line refers to the state of chromatin inside of cells. I'm afraid it's confusing to use "arrays" to describe cellular chromatin.

Reply 2.23: "The switching between these two states is achieved by the reorganization of nucleosomal architecture"

Comment 2.24: "in response to external stimuli": This term is appropriate for cell and animal studies. I think "changes in buffer conditions" is more correct here.

Reply 2.24: Since we are talking about more general activation of this transition, we have replaced all the word of “external stimuli” into “**regulatory signal**”

Comment 2.25: Because the nucleosome has pseudo-twofold symmetry, the terminology used to describe entry and exit DNA needs to be clearer. The mononucleosome used in this MS is asymmetric because one DNA arm is much longer than the other. Instead of calling it the entry and exit DNA arms, it would be a lot clearer to say long and short DNA arms so readers don't have to remember that entry = 200 bp and exit = 100 bp.

Reply 2.25: Respectfully, we disagree with the reviewer on this matter. The terminology of "entry-exit" convention is specifically employed to describe the directionality of the non-palindromic Widom 601 sequence(10.1016/j.cell.2015.02.001, pnas.2206513119). The 40-147-40 intermediate NCP contains two linkers of equal length but differing in the order of the 601 sequence. Hence, we favor the usage of the entry-exit convention to describe the side of the NCP.

Comment 2.26: Line 78-79 “Histone H1 catalyzes more than 10-fold the spinodal-to-spherical conversion”

Similarly, in lines 332-333: “Considering the Histone H1 catalyzes the spinodal-to-spherical conversion for more than 10-time faster than that absents H1” → “Considering that histone H1 catalyzes the spinodal-spherical conversion rate more than 10-times”

Lines 332-333: “Considering the Histone H1 catalyzes the spinodal-to-spherical conversion for more than 10-time faster than that absents H1” > “Considering that histone H1 catalyzes the spinodal-spherical conversion rate more than 10-times”

Reply2.26: We have rewritten the intro section in clarify the scope of our study. The 332-333 has been modified into “*Since linker histone H1 plays a catalytic role in the nucleosome array phase separation¹⁸ and its binding depend on the linker trajectory of the array⁴⁴, we determine if H1 can condense the loosely-packed nucleosome arrays and how this process is achieved.*”

Comment 2.27: Line 84: “large portion of chromatins” > “large portions of the chromatin”

Reply2.27: Changes has been made following reviewer’s suggestion.

Comment 2.28: Line 129-130 “Reference-free 3D reconstruction”: This is a bit excessive. I am unaware of people in the tomography field doing non-reference-free 3D reconstruction.

Reply2.28: Changes has been made following reviewer’s suggestion

Comment 2.29: Line 132 “missing-wedge correction”: I suggest “missing wedge compensation” instead because “correct” means there are no errors. While tomography experts know better, the broader audience will think that we’ve eliminated all matters with the missing wedge, which is not true.

Reply2.29: Changes has been made following reviewer’s suggestion

Comment 2.30: Lines 115-118: The structural features visible are described in a sentence that includes both cryo-EM and NS-EM. Most notably, this sentence mentions “major groove” in the

same phrase as cryo-EM. While the figure callout that follows (Extended Data) is for the NS-EM experiment, a lot of readers are going to think that major grooves were visible in the cryo-EM data. It's better to describe what's seen in cryo-EM vs. what's seen in NS-EM in separate sentences.

Reply2.30: To address the concerns of reviewers, we have added the corresponding figure label at the end of each specific claim.

“A survey of cryo-EM (Fig. 1b) and NS-EM (Extended Data Fig. 1c) micrographs revealed that in a low ionic strength buffer (5 mM Na⁺), mononucleosomes showed: i) a discoidal-shaped NCP with dimensions of ~10 × 10 × 4 nm (Fig. 1d); ii) DNA linkers that are ~2 nm in width, (Fig. 1c) with a helical pitch of ~ 2 nm observed in NS images (Extended Data Fig. 1d-f) which corresponds to its major groove;”

Comment 2.31: Lines 124-125: This sentence conflates what's visible in cryo-EM vs NS-EM. It hurts the credibility of the cryo-EM/ET data (the bulk of the experiments in this MS), which don't have any DNA groove features.

Reply2.31: Respectfully, we disagree with the reviewer on this matter. 1. Based on the aforementioned comment, we have now segregated the NS and cryo-EM 2D image results. 2. Throughout the manuscript, we have not made any assertion regarding the observation of DNA groove features in 3D. It is reasonable to assume that the features in 2D appear lower resolution after 3D reconstruction due to low dose imaging and the missing wedge artifact. Hence, it is not necessary for the same feature to be observed in both 2D and 3D, as assumed by the reviewer.

Comment 2.32: Line 257: What does it mean for DNA arms to be “lifting off from the NCP discoidal plane”?

Reply2.32: We have removed this sentence from our manuscript

Comment 2.33: Line 291 “Tetranucleosome has been used as a phenotype with its minimal structural unit and its capability of forming higher-order chromatin structures...” I don't understand this sentence.

Reply2.33: We have replaced the sentence with: *“The tetranucleosome is considered the fundamental building block of chromatin⁴² due to its ability to form higher-order structures, such as helical fibers^{6,7} and phase separated condensates¹⁸. To study how it initiates intramolecular compaction, we performed the cryo-ET reconstruction with increased ionic strength (Extended Data Fig. 8a).”*

Comment 2.34: Line 317: What kind of “electrostatic effect”?

Reply2.34: It has been rephrased to “electrostatic screening”.

Comment 2.35: Lines 590-591 “optimized negative-staining protocol (OpNS)” To my knowledge, everyone optimizes their negative staining experiments (and all other experiments) prior to publication, but they don't call it optimized negative-staining.

Reply2.35: Following reviewers' suggestion, the sentence has been changed to: “The NS-EM specimens of nucleosome array sample were prepared using the protocol as described⁶²”

Comment 2.36: Lines 648-649: “which can eliminate the effects from the large image distortion”. Local tilt series image alignment cannot eliminate any distortion effects, it merely reduces the impact of large image distortions (?).

Does “large image distortion” mean distortion of a large image (like the original uncropped tilt series) or an image distortion that’s large?

Reply 2.36: Following reviewers’ suggestion, the sentence has been changed to:

“which can **increase the tilt series alignment accuracy by reducing** the effects from the large image distortion³⁷”

FIGURES:

Comment 2.37: Figure 1a, What Na⁺ concentration was used in this experiment? What kind of nucleosome arrays were used in this experiment?

Reply 2.37: The figure 1a has been replaced with a new schematic to show our scope of study.

Comment 2.38: Figure 1c, The density features indicated in orange are not turns of DNA (~90 bp). I think “gyres” is more often used in the field.

Reply 2.38: Following reviewer’s suggestion, these words have been rephrased into: “**the two DNA gyres of an NCP**”.

Comment 2.39: Figure 1d, I am not sure what the advantage of showing tilt series images is. The particles can barely be seen in these projection images.

Reply 2.39: Following reviewers' suggestion, we have deleted the Figure 1d.

Comment 2.40: Figure 1 f/g, Do the cyan and grey isosurfaces correspond to the different contour levels?

Reply 2.40: We acknowledge the comment from the reviewer. The sentence has been modified into "with (gray) and without (cyan) 45 Å Gaussian lowpass superimposed with its flexibly fitted model."

Comment 2.41: Figure 1f/h, Images are cut off.

Reply 2.41: We acknowledge the comment from the reviewer. The image has been cropped into a bigger size.

Comment 2.42: Figure 1g-h: Please give a more detailed description of the difference between theta and theta-parallel? And why are the values for the beta-angles negative?

Reply 2.42: We have included an updated figure g and h, and a more detailed description of angle definition as following:

"g, θ corresponds to the angle between the two DNA arms (red dashed lines) measured in 3-dimensional space. h, Schematic showing the projections of the θ angle (θ_{\parallel} and θ_{\perp}) onto the x-y or y-z planes that are parallel or perpendicular, respectively, to the discoidal plane of the NCP. The orientation of the linker DNA relative to the NCP is defined as the wrapping angle α and the bending angle β , for both entry (cyan) and exit (yellow) side of the NCP. The convention of defining the negative direction as bending away from the NCP is applicable to α_{en} , α_{ex} , β_{en} , β_{ex} , θ_{\parallel} , and θ_{\perp} ."

Comment 2.43: Figure 1h, alpha en and alpha ex are introduced, but the rest of the text and plots just have alpha without subscripts. Please clarify.

Reply 2.43: In the updated manuscript, we explained it as the following:

"The weak correlation between the α_{en} and α_{ex} (Fig. 2j, green points), and between the β_{en} and β_{ex} (Fig. 2j, orange points) suggests that the two DNA arms move independently of one another. Thus, we merge entry- and exit-side measurements into a single distribution."

Comment 2.44: Figure 3c, For multimodal distributions, please show the individual Gaussian curves' fits that allow the extraction of the different peak values. While it is obvious that there are two peaks for the distribution of dinucleosome alpha-angles, it is not for the trinucleosome and tetranucleosome alpha-angle distributions. Could there be more than 2 peaks for those? This issue also appears multiple times in graphs of other figures e.g. Extended Data Figure 7b.

Reply 2.44: We acknowledge the comment from the reviewer. We have calculated the statistics with multimodal gaussian distribution fitting for alpha, beta and Extend Data Figure 7b. The results were appended to the Extend Data Figure 10c.

C

type	Mono-Nuc	Di-Nuc	Tri-Nuc	Tetra-Nuc	Tetra-Nuc-50	Tetra-Nuc-H1
θ	46.5 ± 27.0°	55.0 ± 25.0°	51.9 ± 27.6°	57.9 ± 35.2°	48.2 ± 19.6°	69.7 ± 39.8°
$\theta \parallel$	10.4 ± 44.1°	-28.7 ± 45.3°	-0.9 ± 49.1°	7.4 ± 58.3°	-31.3 ± 35.2°	-2.8 ± 81.3°
$\theta \perp$	32.3 ± 44.7°	33.2 ± 38.2°	36.6 ± 48.5°	31.2 ± 67.8°	22.0 ± 33.8°	18.3 ± 76.6°

type	position	Mono-Nuc	Di-Nuc	Tri-Nuc	Tetra-Nuc	Tetra-Nuc-50	Tetra-Nuc-H1
θ	Distal	46.5 ± 27.0°	55.0 ± 25.0°	51.0 ± 28.9°	59.4 ± 37.5°	49.3 ± 22.4°	60.0 ± 39.1°
	Interior			53.7 ± 25.0°	57.5 ± 33.5°	47.4 ± 17.2°	79.4 ± 38.5°
$\theta \parallel$	Distal	10.4 ± 44.1°	-28.7 ± 45.3°	-3.9 ± 49.2°	3.9 ± 63.3°	-27.4 ± 42.9°	-24.9 ± 68.4°
	Interior			5.0 ± 49.0°	9.7 ± 55.1°	-34.4 ± 27.7°	19.4 ± 87.4°
$\theta \perp$	Distal	32.3 ± 44.7°	33.2 ± 38.2°	34.1 ± 50.6°	28.5 ± 68.5°	20.7 ± 40.4°	12.2 ± 65.2°
	Interior			41.6 ± 44.0°	33.0 ± 67.6°	23.0 ± 27.7°	24.3 ± 86.6°

side	Mono-Nuc	Di-Nuc	Tri-Nuc	Tetra-Nuc	Tetra-Nuc-50	Tetra-Nuc-H1
Entry	-5 ± 5 bp	-4 ± 5 bp	-7 ± 7 bp	-4 ± 8 bp	-4 ± 9 bp	-6 ± 8 bp
Exit	-11 ± 13 bp	-8 ± 10 bp	-10 ± 11 bp	-14 ± 12 bp	-9 ± 8 bp	-17 ± 19 bp

Group	twoGau.Fitting	Mono-Nuc	Di-Nuc	Tri-Nuc	Tetra-Nuc	Tetra-Nuc-50	Tetra-Nuc-H1
Alpha	1 st	39.0 ± 5.4° (54%)	31.8 ± 11.4° (43%)	32.2 ± 13.9° (36%)	31.9 ± 13.5° (39%)	29.7 ± 14.1° (38%)	26.6 ± 13.6° (33%)
Alpha	2 nd	14.8 ± 15.8° (46%)	-5.0 ± 14.0° (57%)	-0.1 ± 16.2° (64%)	-8.4 ± 16.9° (61%)	-13.6 ± 14.4° (62%)	-15.7 ± 13.6° (67%)
Beta	N/A	-9.0 ± 11.2°	-7.6 ± 11.3°	-8.8 ± 12.3°	-9.4 ± 14.1°	-4.4 ± 11.6°	-0.8 ± 13.0°

Group	twoGau.Fitting	Di-Nuc	Tri-Nuc-Nuc	Tetra-Nuc	Tetra-Nuc 50	Tetra-Nuc- H1
$i, i+1$	1 st	224.9 ± 34.6 Å	237.3 ± 27.5 Å	240.9 ± 37.9 Å	208.7 ± 23.7 Å	203.8 ± 19.9 Å
$i, i+1$	2 nd	N/A	N/A	N/A	N/A	293.8 ± 41.2 Å
$i, i+2$	1 st		274.2 ± 89.0 Å	301.7 ± 89.6 Å	155.1 ± 54.1 Å	204.9 ± 40.8 Å
$i, i+2$	2 nd		N/A	N/A	N/A	416.4 ± 90.1 Å
$i, i+3$	N/A			372.3 ± 121.3 Å	273.4 ± 31.0 Å	394.0 ± 106.7 Å

Comment 2.45: Extended Data Figure 1a, What fluorescence is being detected?

Reply 2.45: We acknowledge the comment from the reviewer. Following information has been appended to the MS:

“Development of nucleosomal compaction and condensation in physiological salt by fluorescence microscopy (a) and NS-EM (b) at 0-min, 10-min, and 30-min. **Tetranucleosomes were labeled with Cy3 fluorophore.**”

Comment 2.46: Extended data 1b, lower panel. The condensate is referred to as “Spherical”. Being that this is a negatively stained sample, the material has most likely flattened against the carbon support and indeed in the results, they are described as “relatively flattened”. Please use a more accurate description of the shape of this condensate.

Reply 2.46: We acknowledge the comment from the reviewer. We have added the label “**A flatten spherical condensate**” in Extended data 1b, lower panel.

Comment 2.47: Extended Data Figure 2. Is this data showing 3D reconstructions of cryo-ET or NS-ET? If it is cryo-ET, panel A should show a representative tilt series of cryo-ET, not NS-ET.

Minor note: Or should readers understand that NS-ET data was acquired while the sample holder is operated in cryogenic conditions? Not mentioned in methods.

Reply 2.47: We would like to thank the reviewer for bringing up this point and helping us improve the clarity of our work. We have reorganized the figure so that the NS data only present in Extended Data Fig. 1 to avoid the confusion. The Extended Data Fig. 2 is analysis based on cryo-ET data. **Extended Data Fig. 2: Analyses of cryo-ET 3D reconstruction resolution and nucleosome conformational variety.**

Comment 2.48: Extended Data Figure 5a. There appears to be two peaks for the theta-angle distribution of tetranucleosomes with H1, but only one peak value is reported. This issue also appears multiple times in graphs of other figures e.g. Extended Data Figure 7b.

Reply 2.48: We appreciate the comment from the reviewer. However, our aim is to assist readers in identifying the prominent peak of each distribution, which consistent with the descriptions in the manuscript. As a compromise, we have performed multi-model Gaussian fitting and have included these results in Extended Data Figure 10c.

Comment 2.49: Extended Data Figure 9. “causing intra-array NCP units’ differentiation” – I do not understand what this means. I cannot distinguish individual nucleosomes. There needs to be a 3-D isosurface rendering like Fig. 1f to show that the theta angles can be reasonably measured. For the distribution plots (like Fig 3c), it would be clearer to use a schematic to show the range of angles as two lines with an double-headed arced arrow like Fig. 5b. Also, it would be helpful to have such a schematic for the other distributions like Fig. 3d.

For the tilt series images throughout the MS, why do some show -45, 0 , 45 and others -30, 0, 30 degrees?

Reply 2.49: We appreciate the comment from the reviewer. The title has been rephrased into:” **Extended Data Fig. 9: Linker histone H1 inducing local array networks and conformational change of 2nd NCP.**” The purpose of the figure A is to illustrate the interactions among tetranucleosomes in the presence of H1, leading to the formation of larger structures. We did not measure any statistics from these structures. To make this point clear, we have rewritten the section as “**a, *The Cryo-EM micrograph reveals the presence of large-scale structures in a sample containing tetranucleosome and H1, prepared in a 20 mM HEPES buffer with 5 mM Na⁺. On the right side, a zoomed-in view of the area highlighted by a white rectangle box is displayed.***”

The chosen range was based on the consideration of the figure size and shape. To maintain consistency, we have displayed all tilt series within a range of -30 to 30.

Reviewers' Comments:

Reviewer #1:

Remarks to the Author:

I still have some important concerns with this manuscript. The text is still very hard to follow. Most importantly, the authors have not addressed my technical concerns satisfactorily.

One of the main points I raised was that the single-molecule structural behaviour of chromatin cannot be unequivocally linked to the phase-separation properties of chromatin. Yet, the authors argue that array self-compactation is a preliminary step in the phase separation of chromatin arrays, and therefore, that describing changes to single-molecule behaviour of chromatin arrays is informative of their phase behaviour. Such a connection is hypothetical, so by extrapolating their single-molecule findings to chromatin phase separation, the authors are overstating the impact of their findings.

A second crucial point that I raised was that the results from TMD simulations do not provide dynamic information. My precise concern was "However, targeted MD is a method that biases a conventional MD simulation to enforce a transition between two equilibrium states (e.g. state I and II) by using moving harmonic restraints. Hence, the trajectories stemming from targeted MD are nonequilibrium trajectories that do not reflect the true unbiased dynamics of the system. In other words, the dynamic information that is extracted from TMD is meaningless." I did not say that TMD simulations are meaningless; what I wanted to explain is that nonequilibrium trajectories cannot be used directly to extract dynamical information. Changing "dynamic" to "pseudo-dynamic" is still not accurate. Using TMD to increase the resolution of individual experimental structures at a fixed point in time is ok, what is incorrect is assuming that the nonequilibrium trajectories describe the time-dependent behaviour of chromatin or can reveal how the various experimental structures are linked to one another in time.

The details of the simulations are still very brief, and it is impossible to determine if they were correctly performed.

Reviewer #3:

Remarks to the Author:

Most issues have been addressed sufficiently. There are a few remaining points the authors should address prior to publication and primarily involve wording.

One persistent issue is that the tetranucleosome arrays with H1 have only been analysed in the low ionic strength buffer and are being compared to the 5/50 mM Na⁺ tetranucleosome structures. It would be helpful to have the H1 sample at both the 5 mM and 50 mM Na⁺ concentrations to more carefully characterise how ionic strength and the addition of the linker histone alter tetranucleosome packing and how they contribute to each other. If the authors cannot provide this data, they should state in their conclusions or elsewhere that this a current limitation in this study that will require further work.

Comment 2.5

The title is super vague now. There have been multiple studies that have looked at chromatin conformation using cryo-ET (typically with averaging, which is different from the work presented here). It would be good if the title had some way of distinguishing this manuscript from other that have been recently published. The title suggested by the previous reviewer sufficiently describes the findings in this manuscript and puts them in context of other studies.

Comment 2.14 is not really addressed. The issue is that with low pass filtering it will not be possible to distinguish the gyres of DNA. How did the authors actually distinguish between crossovers or detachment of the DNA ends when Theta approaches 180° with their current applied 4.2 nm lowpass filtering scheme?

Line 597-598: The authors state that use magnifications of 80/50 kx and then state that the pixel

size here is 1.46/1.47 Å/pixel. This does not make sense. The pixel size between 80kx and 50kx should be substantially different.

Minor comments

Line 312- Catalytic to most biochemists suggests that an enzyme is involved, and H1 is not an enzyme. A better term to use here is that H1 stimulates in nucleosome array phase separation rather than catalyses this activity.

Line 505- probably missing the right reference (currently Wittmeyer)

Line 572-573- 15s of glow discharging is mentioned twice

Line 578-579- it would be better to state "as described in a previously published protocol".

Reviewer #1 (Remarks to the Author):

Comment #1.0: *I still have some important concerns with this manuscript. The text is still very hard to follow. Most importantly, the authors have not addressed my technical concerns satisfactorily.*

Reply #1.0: We appreciate the referee's concern, and in response to his/her comments, we have extensively revised several sections and paragraphs of the manuscript to make its reading clearer, we have rearranged the figures. Moreover, in response to the request of the reviewers, we have performed a whole set of experiments and analyses as follows:

1. We have performed new NS-EM experiments with tetranucleosome arrays at 5 mM Na⁺ after incubation for 2 minutes, 20 minutes, 60 minutes, 2.5 hours, 5 hours, and 10 hours, respectively, at room temperature (**The last row in the Fig. 1b**). These experiments have allowed us to observe changes in tetranucleosome morphological development toward condensation and phase separation, under low salt concentration.
2. In parallel, we used Atomic Force Microscopy (AFM) to examine the tetranucleosome arrays at 5 mM Na⁺ after incubation for 75 minutes, 4 hours, and 12.5 hours, respectively, to track the morphology changing by time (**Supplementary Fig. 1d**).
3. We repeated the NS-EM experiments at 50 mM Na⁺ and 150 mM Na⁺ after incubation for 2 minutes, 20 minutes, 60 minutes, 2.5 hours, 5 hours, and 10 hours, respectively, at room temperature, to track the morphology and phase separation process under a higher salt concentration (**The second row in the Fig. 1b and first row in the Fig. 1b**).
4. We used Cryo-ET and 3D reconstruction to examine the detailed structure of the 30-500 nm spherical condensate particles observed in tetranucleosome arrays under physiological salt concentration. These particles have a similar size, shape and density to those previously described (Zhang, *et al.*, *Mol Cell*, 2022; 82(16):3000-3014.e9) (**Fig. 1c**).
5. We used Cryo-ET and 3D reconstruction to examine tetranucleosome arrays with and without H1 under 5- and 50-mM Na⁺ and physiological salt concentration, after ~20 minutes of incubation. This experiment aims to compare how the presence of H1 alters the morphology of tetranucleosome arrays (**Fig. 4a,b,c**).

Comment #1.1: *One of the main points I raised was that the single-molecule structural behaviour of chromatin cannot be unequivocally linked to the phase-separation properties of chromatin. Yet, the authors argue that array self-compaction is a preliminary step in the phase separation of chromatin arrays, and therefore, that describing changes to single-molecule behaviour of chromatin arrays is informative of their phase behaviour. Such a connection is hypothetical, so by extrapolating their single-molecule findings to chromatin phase separation, the authors are overstating the impact of their findings.*

Reply #1.1: We thank this referee for these comments. In response to these remarks, we have added **Figure 1a**, which details the multi-stage process of phase separation. This process encompasses array self-compaction, spinodal condensate formation, nucleation, and the growth of large spherical liquid droplets, as described in our previous publication (Zhang, *et al.*, *Mol Cell*, 2022; 82(16):3000-3014.e9). It is noted that array self-compaction is a rapid reaction (occurring within 2 minutes) under physiological salt concentrations and room temperature, leaving insufficient time for cryo-ET to capture intermediates before the process concludes. Consequently, we slowed down this process by using low salt concentrations, such as 5- and 50-mM Na⁺, respectively, and incubating at room temperature for 20 minutes. To demonstrate that

tetranucleosome samples in low salt concentration did unequivocally link to the phase transition, we conducted additional experiments using NS-EM and AFM to examine the samples under various salt concentrations (5-, 50- and 150-mM Na⁺) after incubation for up to 12 hours (**Fig. 1b and Supplementary Fig. 1c**). The new results confirm that under the low salt conditions, the sample undergoes the same type of phase separation described previously (Zhang, *et al.*, *Mol Cell*, 2022; 82(16):3000-3014.e9) but at significantly lower rates. In this way, we confirm that the intra-array changes described in our work are on-pathway to the formation of inter-array interactions that eventually lead to the generation of liquid droplets.

We agree with the reviewer that although the changes we observe on the isolated arrays are on pathway to the formation of condensates, the arrays may adopt new conformations, different from those we observed in the isolated arrays once inter-array interactions takes place. Nonetheless, we have now shown, that the inter-array interactions that lead to the formation of liquid-liquid phase separation, occur among individual arrays that have undergone the conformational intra-array changes that we have described, whether these changes are retained in the resulting condensates or not. Throughout the revised manuscript, we have refrained from making statements that imply a focus on liquid droplets. We also realized that the simulation of confining/filling in-silico fibers within a box could be misleading, as it artificially brings NCPs together within a small volume which resemble a condensate. Our primary objective in conducting this simulation was to determine the tertiary structure of these loosely packed (not condensed) fibers and compare them to in-situ tomographic slices of similar thickness. To eliminate this potential source of confusion, we have removed this part of the results and designed a new simulation protocol. In this revised simulation, although computationally intensive, we successfully allow the fibers to grow freely, forming a mini-chromosome (containing 250,000 NCPs) without any box confinement. This structure encompasses both high-density and low-density regions that has no structural similarity to the spherical condensates.

The added sections, revised paragraphs and figures include the following:

The first section in result:

“Slowing down the kinetics of phase separation by reducing Na⁺ concentration

Prior investigations utilizing fluorescence microscopy and negative stain electron microscopy (NS-EM) have revealed that spherical condensates of 30 nM tetranucleosome arrays can form within 10 minutes of incubation under physiological salt concentrations (150 mM Na⁺ and 5 mM Mg²⁺) and room temperature, (**Supplementary Fig. 1a-b**). This process occurs following the almost immediate formation of spinodal condensates. In the presence of H1, the spherical condensates are observed within 2 minutes of incubation¹. This rapid reaction poses a challenge to characterize the initial intra-array conformational changes that take place before inter-array interactions commence.

After 20-60 minutes of incubation at room temperature under a 150 mM Na⁺ concentration, NS-EM imaging revealed the presence of irregular aggregates previously characterized as spinodal condensates, and small isolated spherical condensate particles ranging from 30-50 nm in diameter¹. These small spherical condensate particles could grow to micrometer size with prolonged incubation (**Fig. 1b**). However, reducing the salt concentration to 50 mM Na⁺ delayed the appearance of similar condensates to longer than 2.5 hours of incubation (**Fig. 1b**). Further reduction to 5 mM Na⁺ extended the generation of similar condensate particles to approximately 10 hours (**Fig. 1b**). Similar results were observed by AFM (**Supplementary Fig. 1c**). Under Cryo-

ET analysis, these 30-50 nm spherical condensates, displayed the same features observed under physiological salt concentrations (**Fig. 1c**), corroborating previous cryo-EM study¹. These experiments indicate that decreasing Na⁺ concentration can be used to capture the changes in intra-nucleosomal array structure during the rapid process of intra-array compaction before the emergence of spherical condensates. Therefore, we analyzed structures obtained at 5 mM Na⁺ and 50 mM Na⁺ after 20 minutes of incubation.”

“Fig. 1: Structure and morphology of phase separation. a, A schematic model depicting the structural transition of nucleosomes, from a 10-nm “beads on a chain” structure or loosely-packed polymer state to a fiber-like structure, spinodal condensate, and finally to a spherical condensate through the state of intra-array compaction, inter-array interaction and condensation processes. **b**, NS-EM images of the morphology of tetranucleosome incubated with 150 mM, 50 mM and 5

mM Na⁺, respectively, under room temperature for up to 10 hours. The spinodal condensate particles are marked by orange circles, while the spherical condensates are indicated by cyan arrows. The boundary for generating spinodal condensate is described by orange dash line, while the boundary to generating the spherical condensate is described by cyan dash line. **c**, Three representative area of cryo-ET 3D reconstruction of spherical condensate generated from tetranucleosome in physiological salt concentration as described¹. Each 3D map showed by its central slide (top left) and zoomed-in portion of the small spherical condensates (top right), which is compared to the 3D density map superimposed with models (the NCP portions are colored by cyan, and DNA portion are colored by yellow).”

“Supplementary Fig. 1: Development of nucleosomal compaction and condensation. **a**, Phase separation process of tetranucleosome in physiological salt conditions observed by

fluorescence microscopy, with tetranucleosomes labeled with Cy3 fluorophore. **b**, NS-EM images show loosely-packed arrays at low-salt (1.5 mM Na⁺) conditions, and **c**, inter-array interaction indicating spinodal and spherical condensates at 10 and 30 minutes, respectively, under the physiological salt concentration (150 mM Na⁺, and 5 mM Mg²⁺). **d**, AFM observation of spinodal condensate conformation. The control image at the top panel is from a sample without tetranucleosome, containing only HEPES buffer with 5 mM Na⁺ after 5 minutes incubation. **e**, The remaining three images are from the samples containing the 30 nM tetranucleosome in 20 mM HEPES buffer with 5 mM Na⁺ after 75 minutes, 4 hours, and 12 hours of incubation, respectively. “

Simulation sections now reads as follows:

Last paragraph in result: “Additionally, by extending our simulations to encompass a larger structure (250,000 NCPs, analogous to a mini chromosome with 50 million base pairs of DNA), we explored the tertiary structure of these simulated fibers. This larger chromatin structure display a relatively dispersed spatial organization, featuring both low-density and high-density regions (Supplementary Fig. 13b-c), closely resembling the chromatin domain morphology observed in studies using fluorescence 3D imaging² and immuno-gold labeling³ of interphase nuclei.”

Discussion (in the middle of third paragraph): “... A further test was conducted by simulating a mini-chromosome threading 250,000 NCPs using the same angle parameters. The distribution of NCPs within a thin slab (25 nm thickness) in the dense region (Supplementary Fig. 13b-c, right panel) displays a morphology and density similar to those observed in in-situ cryo-ETs, such as the tomographic slices of HeLa cells (10-20 nm thickness)⁴, chicken erythrocyte nuclei (10-50 nm thickness)⁵, picoplankton cell lysates (30 nm thickness)⁶, and in 2D views of swollen HeLa S3 chromosomes at low Mg²⁺ concentration⁷. Other parameters, such as physiological salt, temperature, involvement of binding proteins, and nucleosomal phasing, have not yet been incorporated into the simulation. However, the likeness between the structures observed experimentally and those generated here, highlights the importance of the entry and exit linker angles in regulating the overall morphology of chromatin prior to phase transition.”

“Supplementary Fig. 13: Simulation of the chromatin morphology regulated by the experimental distribution. a, Distribution of in-silico hectanucleosome array fiber length, width, and NCP density and the statistics of the experimentally measured array parameters. Histograms of the measured contour length (left columns) and width (right column) of the simulated nucleosome array fiber with 100 NCP units. 100 simulated array fibers are generated for each category (row 1 through 4) to calculate the statistics. **b,c,** Simulation of a mini-chromosome by threading 250,000 NCPs using the parameters measured from the tetranucleosome arrays in 5 mM and 50 mM Na⁺, respectively. Representative 200-nm cubic volumes (middle panel) and their central slices with a thickness of 25 nm (right panel) were cropped from the high-density regions of each structure.”

Comment #1.2: *A second crucial point that I raised was that the results from TMD simulations do not provide dynamic information. My precise concern was “However, targeted MD is a method that biases a conventional MD simulation to enforce a transition between two equilibrium states (e.g. state I and II) by using moving harmonic restraints. Hence, the trajectories stemming from targeted MD are nonequilibrium trajectories that do not reflect the true unbiased dynamics of the system. In other words, the dynamic information that is extracted from TMD is meaningless.” I did not say that TMD simulations are meaningless; what I wanted to explain is that nonequilibrium trajectories cannot be used directly to extract dynamical information. Changing “dynamic” to “pseudo-dynamic” is still not accurate. Using TMD to increase the resolution of individual experimental structures at a fixed point in time is ok, what is incorrect is assuming that the nonequilibrium trajectories describe the time-dependent behaviour of chromatin or can reveal how the various experimental structures are linked to one another in time. The details of the simulations are still very brief, and it is impossible to determine if they were correctly performed.*

Reply #1.2: We appreciate the referee's comment. We acknowledge that we are not experts in the field of Targeted Molecular Dynamics (TMD) to debate its applicability range in modeling large-scale conformational changes in equilibrium or nonequilibrium systems. However, based on the literature, TMD is described as "a method to induce a conformational change to a known target structure at ordinary temperature by applying a time-dependent, purely geometrical constraint. The transition is enforced regardless of the energy barriers' height, while the dynamics of the molecule is minimally influenced by the constraint" (quoted from Schlitter et al., *J Mol Graph.* 1994;12(2):84-9). Over the past decade, TMD has been "applied generally to model large conformational changes in any protein for which the initial and final three-dimensional structures are known" (quoted from Krüger et al., *Protein Sci.* 2001;10(4):798-808), irrespective of whether the structures were derived from nonequilibrium (Wolf et al., *Nat Comm.*, 2020; 11, 2918; Moradi et al., *PNAS*, 2013; 110 (47) 18916-18921; Wolf et al., *J. Chem. Theory Comput.* 2018, 14, 12, 6175–6182) or equilibrium systems (Zhang et al., *Nat Comm.*, 2016; 7, 11083; Yoon et al., *Sci Rep*, 2018; 8, 5673; Mori et al., *Biophys J.* 2021;120(6):1060-1071; Li et al., *Sci Rep*, 2015, 5, 11289; Wu et al., *J Biomol Struct Dyn.* 2017;35(1):119-127; Zhang et al., *Sci Rep*, 2015; 5, 9803).

In this study, we wish to clarify that we did not employ TMD to link structures observed at different time points. Instead, we used TMD solely to illustrate the various conformational states of structures observed at the same time point (after 20 minutes of incubation, an operation similar to the morphing function in UCSF Chimera (Krebs et al., *Nucleic Acids Res.* 2000; 28(8):1665-

75). The morphing movies between two structures are a common practice in cryo-EM structural studies (Abid Ali et al., *Nat Comm.*, 2017; 8, 2241; Hirschi et al., *Nat Comm.*, 2021; 12, 4107; James et al., *PNAS*, 2017; 114(17): 4430–4435), aiming to facilitate the visualization of structures changing via motion rather than a static snapshot. We encountered an issue with the ridge morphing function in software that caused the DNA chain to split or become distorted. Consequently, we opted for a more sophisticated TMD morphing approach to induce a conformational change by applying a purely geometrical constraint. In agreement with the reviewer, we acknowledge that these morphing animations serve only a visualization purpose, and we should not interpret the dynamic trajectories as data or results. Considering that '*pseudo-dynamic*' is not an accurate term to describe the movie, we have revised our terminology to use 'morph' and 'conformational variety' to more precisely describe this morphing animation as following.

Changes made in main text:

The last paragraph in the section of “**3D structural dynamics of mononucleosomes under 5 mM Na⁺**” now reads as follows:

“To characterize the structural diversity of the mononucleosomes, the 47 structures were pairwise aligned by using the minimization of their root-mean-square deviations (RMSD) and then ordered according to a hierarchical classification⁸ (**Supplementary Fig. 4a**). Mononucleosome structures were morphed from the most populated state to the rarest state under a purely geometrical constraint by target molecular dynamic (TMD) simulation. This morphing illustrates the possible interconversion among conformational states adopted by the arrays under non-equilibrium conditions after 20 minutes incubation and under 5 mM Na⁺ (**Supplementary Video 1**). ...”

The last paragraph in the section of “**3D structural dynamics of nucleosome arrays under 5 mM Na⁺**” now reads:

“Movies of di- and trinucleosomes morphing from the most populated to the least or rare state were created under a purely geometrical constraint by TMD simulation, as used for mononucleosomes (**Supplementary Fig. 4b,c** and **Supplementary Video 2**). These movies illustrate possible conformational fluctuations of the arrays after 20 minutes incubation and under 5 mM Na⁺. ...”

And the materials and Methods section has now been expanded to provide more information about the morphing process as following,

Visualization of the nucleosome array structural diversity

“To facilitate the visualization of the conformational variety of nucleosome array structures via motion rather than a static snapshot, we morphed the conformations of nucleosome array structures from one to another. This was done by following a hierarchical clustering order, calculated based on the pairwise RMSD values among all structures. The morphing animations began with the structures located at the lowest branch of the dendrogram (representing consensus conformations) and terminated at the highest branch, corresponding to rare conformations. Because the chimera morphing function introduces artificial DNA kinks for nucleosome arrays, the morphing was carried out using Targeted Molecular Dynamics (TMD) simulations. The TMD was performed in NAMD2 engine⁹, where the SIRAH coarse-grained structure¹⁰ was steered from one conformation toward to another under Amber force field (using the same parameters as SIRAH forcefield in AMBER Tutorial2¹⁰) within implicit solvent (5 mM ion

concentration). The System was maintained at 298 K temperature using the Langevin dynamics with a damping coefficient of 50/ps. The elastic constants for TMD forces used were in a range of 200-600 kcal/mol/Å scaled based on the distances of the corresponding CG model atom pairs. After 50,000 steps of energy minimization (corresponding to a total time of 1 ns, with each step of 20 fs) and 1,000,000 to 4,000,000 steps of steering (corresponding to a total time of 20 ns to 80 ns) within the implicit solvent, the simulation was terminated when the real-time RMSD fell below 3 Å. Among these morphing structure pairs, ~10% of them were eliminated due to the DNA intertwining with each other. The morphing movie was also displayed with the same order as above from the most populated structure states to the rarest states. We noted that the trajectories stemming from TMD are nonequilibrium trajectories that do not reflect the true unbiased dynamics of the system.”

Reviewer #3 (Remarks to the Author):

Comment #3.0: *Most issues have been addressed sufficiently. There are a few remaining points the authors should address prior to publication and primarily involve wording.*

Reply #3.0: We appreciate the reviewer's acceptance of most of our clarifications and modifications, which significantly reduced the number of concerns in the first round. In the current revised manuscript, we have carefully addressed the remaining questions.

Comment #3.1: *One persistent issue is that the tetranucleosome arrays with H1 have only been analysed in the low ionic strength buffer and are being compared to the 5/50 mM Na⁺ tetranucleosome structures. It would be helpful to have the H1 sample at both the 5 mM and 50 mM Na⁺ concentrations to more carefully characterise how ionic strength and the addition of the linker histone alter tetranucleosome packing and how they contribute to each other. If the authors cannot provide this data, they should state in their conclusions or elsewhere that this a current limitation in this study that will require further work.*

Reply #3.1: We thank the referee for this discussion. In response to this comment, we conducted cryo-ET and 3D reconstruction to examine the tetranucleosome arrays with and without H1, after ~20 minutes of incubation under 5- and 50-mM Na⁺ and physiological salt concentration. This experiment is to compare how H1 changes the morphology of tetranucleosome arrays in response to different salt conditions. The result demonstrates that at higher ionic condition (**Fig. 4b and c**, and **Supplementary Fig. 11d and f**), H1 not only causes NCPs to stack tightly together but also prompts inter-array interactions, leading to the entanglement of long DNA arms. It becomes challenging to distinguish the NCP orientation and the origin of the long distal DNA arms. Thus, we have included the figures from the new experiments and updated the related paragraph as following,

“Fig. 4: Morphology and 3D structures of tetranucleosome arrays in the presence of H1. a-c, Cryo-EM images show the morphology of tetranucleosome arrays incubated with 5 mM Na⁺, 50 mM Na⁺ and physiological salt, respectively, for 20 minutes, in the presence of linker histone H1. ...”

“Supplementary Fig. 11: Morphology induced by the present of H1 under various Na⁺ concentrations. a, The Cryo-ET image (left panel) of tetranucleosome in 5 mM Na⁺ after 20 min incubation at room temperature. The high-density portions are highlighted in blue (left panel) and superimposed on the micrograph (middle panel). b, The same sample in present of H1 is compared. c, Cryo-ET image of the tetranucleosome in 50 mM Na⁺ after 20 min incubation at room temperature, d, which is compared to same sample in the present of H1. e, The Cryo-ET image of the same sample in physiological salt concentration after 20 min of incubation at room temperature, f, which is compared to the same sample in the present of H1. ”

Changes made in main text “H1-induced DNA invasion into neighboring NCPs” section

“Given that the linker histone H1 accelerates tetranucleosomes phase separation¹ and that its binding affects the linker trajectory of the array¹¹, we aimed to determine whether H1 could induce compaction in loosely-packed nucleosome arrays under 5 mM Na⁺, akin to the effect observed at higher salt concentrations (Fig. 3m). Tetranucleosomes were incubated with H1 in 5 mM Na⁺

using a 1:4 molar ratio for 20 mins at room temperature (Fig. 4a). Interestingly, visual inspection indicated that arrays in the presence of H1 do not attain the same level of compaction at 5 mM Na⁺ as observed in its absence at higher salt concentration (50 mM Na⁺) (Fig. 4a, Supplementary Fig. 10). However, the simultaneous presence of both factors, H1 and higher salt conditions, under the same incubation time and temperature, induces array condensation (Fig. 4b,c, and Supplementary Fig. 11), preventing the structural determination of each individual array. Thus, we can only compare structural changes observed with and without H1 at 5 mM Na⁺.”

Comment #3.2: *The title is super vague now. There have been multiple studies that have looked at chromatin conformation using cryo-ET (typically with averaging, which is different from the work presented here). It would be good if the title had some way of distinguishing this manuscript from other that have been recently published. The title suggested by the previous reviewer sufficiently describes the findings in this manuscript and puts them in context of other studies.*

Reply #3.2: We appreciate the reviewer's feedback regarding the title of our manuscript. We agree with the reviewer's suggestion that our title should distinguish the current study from structural averaging. To encompass all these aspects, we have revised our title as following,

“Angle between DNA linker and nucleosome core particle regulates nucleosome array compaction revealed by individual-particle cryo-electron tomography”

Comment #3.3: *Comment 2.14 is not really addressed. The issue is that with low pass filtering it will not be possible to distinguish the gyres of DNA. How did the authors actually distinguish between crossovers or detachment of the DNA ends when Theta approaches 180° with their current applied 4.2 nm lowpass filtering scheme?*

Reply #3.3: We appreciate the comment from this referee. In response, we added additional figures regarding the zoomed-in view of the NCP and DNA linker portions in the 3D reconstruction (Fig.2 and Supplementary Fig. 3). As can be seen, the projection of the densities showed the detailed connection between the linker DNA and NCP, which provide sufficient information to define the gyres of DNA. Regarding two different cases, i.e. the crossovers and detachment of the DNA ends when Theta approaches 180°, it would result in a 27 nm length (corresponding to 80 base pairs) increase of the DNA linker at either entry or exit ends of the NCP as showed in Supplementary Fig. 8e-g as shown below. This significant feature can be easily observed during the model fitting.

“**Fig. 2: Cryo-EM 3D structure and dynamics of mononucleosome.** ... **h,i,j,** Zoom-in view of the NCP regions of three mononucleosomes, shown with the projection, the final map, and fitted model, respectively. DNA on the histone is highlighted by yellow arrows. “

“**Supplementary Fig. 3: Validation of the cryo-ET 3D reconstruction resolution.** ... **d,e,f,** Three representative particle showed from two directions (45°), displayed by the projection of the final 3D density map, two isosurface contour maps, and fitting model. The observed spatial orientation of 2-nm dsNDA linker validates the reported 3D map resolution. “

“**Supplementary Fig. 8: Distributions of the wrapping angle α and the bending angle β .** ...**e-g,** Evaluate nucleosome array model fitting by correlating unwrapping angle with linker length. **e,** Schematic of two possible NCP fitting orientations for an unwrapped NCP map. **f,** NCP unwrapping leads to longer linker length, proportional to degree of unwrapping (measured as φ angle compared to fully wrapped crystal structure). **g,** Correlation analysis between DNA linker length and NCP unwrapping angle from all NCP pairs, with orange points as fitting outliers.”

We also revised the second paragraph in the method section of “**Modeling the structure of nucleosome arrays**” as follows,

“In the meantime, the DNA linker length serves as a mean to differentiate between crossovers (octasome) and detachment of DNA ends (hexasome) when θ angle approaches 180° . An evident contrast between the two scenarios is that ~ 80 bp of DNA unwrapping in the hexasome leads to an additional 27 nm of DNA extending from either distal end of the NCP arms. ...”

Minor comment #3.1: Line 597-598: The authors state that use magnifications of 80/50 kx and then state that the pixel size here is 1.46/1.47 A/pixel. This does not make sense. The pixel size between 80kx and 50kx should be substantially different.

Reply: Typo has been corrected, 1.47 has been changed to 1.67

Minor comment #3.2: Line 312- Catalytic to most biochemists suggests that an enzyme is involved, and H1 is not an enzyme. A better term to use here is that H1 stimulates in nucleosome array phase separation rather than catalyses this activity.

Reply: We have changed the word of catalysis to accelerate

Minor comment #3.3: Line 505- probably missing the right reference (currently Wittmeyer)

Reply: Change has been made

Minor comment #3.4: Line 572-573- 15s of glow discharging is mentioned twice

Reply: Duplication has been removed

Minor comment #3.5: Line 578-579- it would be better to state “as described in a previously published protocol”.

Reply: Change has been made

References:

- 1 Zhang, M. *et al.* Molecular organization of the early stages of nucleosome phase separation visualized by cryo-electron tomography. *Mol Cell* **82**, 3000-3014 e3009, doi:10.1016/j.molcel.2022.06.032 (2022).
- 2 Hao, X. *et al.* Super-resolution visualization and modeling of human chromosomal regions reveals cohesin-dependent loop structures. *Genome Biol* **22**, 150, doi:10.1186/s13059-021-02343-w (2021).
- 3 Visser, A. E., Jaunin, F., Fakan, S. & Aten, J. A. High resolution analysis of interphase chromosome domains. *J Cell Sci* **113 (Pt 14)**, 2585-2593, doi:10.1242/jcs.113.14.2585 (2000).
- 4 Cai, S., Bock, D., Pilhofer, M. & Gan, L. The in situ structures of mono-, di-, and trinucleosomes in human heterochromatin. *Mol Biol Cell* **29**, 2450-2457, doi:10.1091/mbc.E18-05-0331 (2018).
- 5 Scheffer, M. P., Eltsov, M. & Frangakis, A. S. Evidence for short-range helical order in the 30-nm chromatin fibers of erythrocyte nuclei. *Proc Natl Acad Sci U S A* **108**, 16992-16997, doi:10.1073/pnas.1108268108 (2011).
- 6 Cai, S., Song, Y., Chen, C., Shi, J. & Gan, L. Natural chromatin is heterogeneous and self-associates in vitro. *Mol Biol Cell* **29**, 1652-1663, doi:10.1091/mbc.E17-07-0449 (2018).
- 7 Eltsov, M., Maclellan, K. M., Maeshima, K., Frangakis, A. S. & Dubochet, J. Analysis of cryo-electron microscopy images does not support the existence of 30-nm chromatin fibers in mitotic

- chromosomes in situ. *Proc Natl Acad Sci U S A* **105**, 19732-19737, doi:10.1073/pnas.0810057105 (2008).
- 8 Pedregosa, F. *et al.* Scikit-learn: Machine learning in Python. *the Journal of machine Learning research* **12**, 2825-2830 (2011).
- 9 Phillips, J. C. *et al.* Scalable molecular dynamics on CPU and GPU architectures with NAMD. *J Chem Phys* **153**, 044130, doi:10.1063/5.0014475 (2020).
- 10 Dans, P. D., Zeida, A., Machado, M. R. & Pantano, S. A Coarse Grained Model for Atomic-Detailed DNA Simulations with Explicit Electrostatics. *J Chem Theory Comput* **6**, 1711-1725, doi:10.1021/ct900653p (2010).
- 11 Dombrowski, M., Engeholm, M., Dienemann, C., Dodonova, S. & Cramer, P. Histone H1 binding to nucleosome arrays depends on linker DNA length and trajectory. *Nat Struct Mol Biol* **29**, 493-501, doi:10.1038/s41594-022-00768-w (2022).

Reviewers' Comments:

Reviewer #1:

Remarks to the Author:

The authors have now satisfactorily addressed my concerns.

Minor comment:

"under Amber force field (using the same parameters as SIRAH forcefield in AMBER Tutorial)" should read instead: "with the SIRAH forcefield as implemented in AMBER".

Reviewer #3:

Remarks to the Author:

The authors have addressed my concerns.

Reviewer #1 (Remarks to the Author):

Comment #1.0: *The authors have now satisfactorily addressed my concerns.*

Reply #1.0: We greatly appreciate the referee's latest comment. We have addressed these comments in the revised manuscript.

Comment #1.1: *Minor comment: "under Amber force field (using the same parameters as SIRAH forcefield in AMBER Tutorial)" should read instead: "with the SIRAH forcefield as implemented in AMBER".*

Reply #1.1: We thank the referee for this comment. In response, we have revised the related sentence as follows:

Line 891:

“...The TMD was performed in NAMD2 engine³⁴, where the SIRAH coarse-grained structure⁷⁶ was steered from one conformation toward to another **with the SIRAH forcefield as implemented in AMBER**⁷⁶ within implicit solvent (5 mM ion concentration)...”

Reviewer #3 (Remarks to the Author):

Comment #3.0: *The authors have addressed my concerns.*

Reply #3.0: We thank the referee again for the previous comments.